JCB Journal of Cell Biology

# Loss of ErbB3 redirects Integrin β1 from early endosomal recycling to secretion in extracellular vesicles

Dorival Mendes Rodrigues-Junior[1]* , Ana Rosa Sáez-Ibáñez[2]* , Takeshi Terabayashi[3] , Nina Daubel[4] , Taija Mäkinen[4] , Olof Idevall-Hagren[5] , Aristidis Moustakas[1] , and Ingvar Ferby[2,4]

Receptor tyrosine kinases (RTKs) are important cargo in endocytic trafficking, yet their role in endosomal sorting and maturation of multivesicular bodies remains unclear. Here, we show that the ErbB3 (HER3) receptor sorts internalized Integrin β1 and the transferrin receptor, for endocytic recycling, in a manner that does not require ligand-induced ErbB3 signaling in breast epithelial cells. Loss of ErbB3 abrogates recycling of Integrin β1, likely from a Rab4-positive compartment, and redirects it toward lysosomal degradation or secretion as an extracellular vesicle (EV) cargo. ErbB3 depletion impairs the collective migration of breast epithelial cell sheets, coinciding with reduced cell-surface levels of Integrin β1 and increased release of Integrin β1–containing EVs. In contrast, EVs secreted from ErbB3-depleted cells enhance the motility of wild-type cells. Mechanistically, ErbB3 promotes assembly of the Arf6–GGA3–Rabaptin5 endosomal sorting complex to facilitate early recycling and suppress EV release. These findings provoke the notion that pseudo-RTKs play an active role in vesicular trafficking.

## Introduction

The epidermal growth factor receptor (EGFR) family of receptor tyrosine kinases (RTKs), including EGFR, ErbB2, ErbB3, and ErbB4, form homo- or heterodimers and play key roles in epithelial development and homeostasis, while their deregulation contributes to most epithelial cancers (Schlessinger, 2002). RTK dimers are activated upon ligand binding, leading to recruitment of molecules that initiate intracellular signaling cascades. ErbB3 has impaired kinase activity, but signals by forming heterodimers with other members of the EGFR family, and accumulating evidence highlights its potent oncogenic activity. In particular, ErbB3 has been linked to the migratory behavior and metastasis of cancer cells and is responsible for acquired resistance to anti-EGFR therapies (Gaborit et al., 2016; Pradeep et al., 2014; Smirnova et al., 2012; Yoshioka et al., 2010; Tiwary et al., 2014).

Ligand binding to RTKs also triggers receptor internalization and trafficking within the cells, which determines strength, duration, and spatial distribution of EGFR signals (Al-Akhrass et al., 2017; Avraham and Yarden, 2011; Miaczynska and Bar-Sagi, 2010; Sorkin and Goh, 2009; Wiley, 2003). RTKs, including EGFRs, are internalized through clathrin-dependent and clathrin-independent mechanisms, converging on the delivery of the receptors to early endosomes (Sorkin and Goh, 2009). Notably, although the mechanisms regulating ErbB3 turnover are still not fully understood, protein kinase C has been linked promoting ErbB3 endosomal sorting (Dietrich et al., 2019). Thereafter, RTKs are diverted toward either (1) the plasma membrane via recycling; (2) protein degradation through lysosomes; or (3) secretion in the extracellular milieu as cargo of small extracellular vesicles (EVs) (Fosdahl et al., 2017; van Niel et al., 2022). The collective term EVs refers to a variety of secreted nanovesicles confined by a lipid bilayer, such as (1) exosomes, which have endosomal origin, are formed upon the maturation of MVBs, and are released into the extracellular milieu; or (2) microvesicles originating from the plasma membrane, both mediating intercellular communication from secreting to recipient cells by carrying bioactive molecules (Raposo and Stoorvogel, 2013; van Niel et al., 2022).

Rab GTPases Rab4 ("short-loop") and Rab11 ("long-loop") are major regulators driving endosomal trafficking and recycling of

[1]Department of Medical Biochemistry and Microbiology, Science for Life Laboratory, Biomedical Center, Uppsala University, Uppsala, Sweden;   [2]Department of Medical Biochemistry and Microbiology, Biomedical Center, Uppsala University, Uppsala, Sweden;   [3]Department of Pharmacology, Faculty of Medicine, Oita University, Oita, Japan;   [4]Department of Immunology, Genetics and Pathology, Uppsala University, Uppsala, Sweden;   [5]Department of Medical Cell Biology, Uppsala University, Uppsala, Sweden.

*D.M. Rodrigues-Junior and A.R. Sáez-Ibáñez contributed equally to this paper.   Correspondence to Dorival Mendes Rodrigues-Junior: dorival.mrj@imbim.uu.se;   Ingvar Ferby: ingvar.ferby@igp.uu.se.

RTKs and other transmembrane proteins, including integrins (Bridgewater et al., 2012; Lolo et al., 2022; Tomas et al., 2014). Of note, the ubiquitously expressed GGA1-3 proteins, which are clathrin adaptors dependent on small GTPases of the Arf family, have been found to regulate Rab4- or Rab11-dependent endosomal recycling of Integrin β1 (GGA2, GGA3), the Met and Ret RTKs (GGA3), and the transferrin receptor (TfR) (Arjonen et al., 2012; Crupi et al., 2020; Parachoniak et al., 2011; Puertollano and Bonifacino, 2004; Ratcliffe et al., 2016; Sahgal et al., 2019; Zhao and Keen, 2008). GGA proteins contain several functional domains that bind to accessory proteins modulating membrane trafficking, including Rabaptin5 (Puertollano and Bonifacino, 2004; Mattera et al., 2003; Miller et al., 2003; Zhai et al., 2003; Zhu et al., 2004). Rabaptin5 cooperatively promotes a linkage among Rab5- and Rab4-containing microdomains on early endosomes to coordinate the coupling of these domains, thereby regulating sorting of cargo including Integrin β3 and TfR into recycling endosomes (Christodorides et al., 2012; de Renzis et al., 2002; Deneka et al., 2003; Pagano et al., 2004).

We found a novel role of ErbB3 in vesicular sorting whereby it stabilizes the Arf6–GGA3–Rabaptin5 endosomal sorting complex, to promote early recycling of TfR and Integrin β1. Depletion of ErbB3 or its effectors Rabaptin5 or GGA3 reroutes Integrin β1 from endocytic recycling toward lysosomal degradation or secretion as EV cargo. Altogether, these findings depict a novel mechanism by which a pseudo-RTK regulates endosomal events with potential impact on endocytic pathway–related biomedicine.

## Results

### ErbB3 guides Integrin β1 to the leading edge of motile epithelial cell sheets

ErbB3 is commonly overexpressed in several types of cancer, contributing to tumor progression and dissemination proposedly by promoting cell migration, yet the underlying molecular mechanism and cellular function remain poorly understood likely due to its lack of intrinsic kinase activity. To better elucidate the role of ErbB3 in cell migration, we assessed the impact of ErbB3 silencing on migrating sheets of nonmalignant breast epithelial MCF10A cells. Depletion of ErbB3 in MCF10A cells (Fig. 1 A) reduced the rate of wound closure compared with control (Fig. 1, B and C), with the area under the curve (AUC) increasing by 50.2% and 48.2% in DMSO vs. lapatinib-treated samples, respectively (Fig. 1 D). Lapatinib is a dual-specificity EGFR/ErbB2 inhibitor that was administered to eliminate potential influence of epidermal growth factor (EGF)–induced cell migration (Lauand et al., 2013; Maretzky et al., 2011). At the applied concentration of 1 µM, no cytotoxic effects were observed, while activation-linked phosphorylation of EGFR and downstream ERK1/2 signaling was effectively blocked (Fig. S1, A and B). Additionally, the cell proliferation determined by EdU incorporation did not differ between control or ErbB3 siRNA-transfected cells (Fig. 1 E). Thus, in the absence of any impaired proliferation, the observed sheet migration defect caused by ErbB3 loss cannot be attributed to reduced cell proliferation. Our data show that ErbB3 promotes migration of epithelial sheets in

a manner that does not require canonical transphosphorylation of ErbB3 by EGFR or ErbB2 or the kinase activity of these receptors.

Integrins are pivotal molecules regulating cell–cell adhesion and cell–extracellular matrix (ECM) interactions, and the β1-subunit is the most common among the 18 α- and 8 β-subunits, forming the basis for cell survival and motility of epithelial cells (Sun et al., 2023; Weaver et al., 1997). Endocytic recycling is important for the polarized and dynamic distribution of integrins in migrating cells (Haskins et al., 2014; Nader et al., 2016; Qu et al., 2016; Wali et al., 2014). Integrin β1 localizes to the leading edge of migrating epithelial cells in a Rab4-dependent manner (Arjonen et al., 2012; Ratcliffe et al., 2016). To evaluate whether ErbB3 regulates the distribution of recycled Integrin β1 in sheets of migrating breast epithelia, a scratch wound was inflicted on confluent monolayers of MCF10A cells, followed by their incubation for surface labeling with Alexa 488–conjugated Integrin β1 antibody for 1 h on ice, prior to removal of the antibody and subsequent incubation at 37°C for 1 h (Fig. 1 F). Chased Integrin β1 and F-actin were visualized in cells at the leading front of the closing cell sheets by immunofluorescence imaging (Fig. 1 G). Integrin β1 enrichment at the leading edge or along cell–cell contacts relative to adjacent cytoplasm was quantified in cells bordering the wound (boxed regions in Fig. 1 G). RNAi-mediated depletion of ErbB3 led to a 59.6% decrease in Integrin β1 enrichment at the leading edge, as compared to control siRNA-treated cells after 1 h of tracing at 37°C (Fig. 1 H). At cell–cell contacts, Integrin β1 enrichment decreased by 42.5% (Fig. 1 I). These results are consistent with a ligand-independent role of ErbB3 in recycling Integrin β1 to the leading edge of migrating cells and to a lesser extent to cell–cell contacts.

### ErbB3 promotes endocytic recycling of Integrin β1

Notably, ErbB3, like Integrin β1, has been found to continuously endocytose and recycle back to the plasma membrane in a manner that does not require ligand stimulation or other members of the EGFR family (Fosdahl et al., 2017; Sak et al., 2012). Thus, we wondered whether endosomal trafficking of ErbB3 might be coordinated with, or even regulate, endocytic sorting and trafficking of Integrin β1. As an initial step, we asked whether ErbB3 colocalizes with internalized Integrin β1. Briefly, the surface pool of Integrin β1 on the luminal breast cancer (BRCA) cell line, MCF7, was labeled on ice with an Alexa 488–conjugated antibody, and allowed to internalize for 15 min at 37°C, prior to fixation and immunolabeling of endogenous ErbB3. MCF7 cells were used since they overexpress ErbB3 facilitating detection of endogenous protein by immunofluorescence (Fig. S1 C). Confocal microscopy showed that ErbB3 and traced Integrin β1 colocalized on intracellular structures likely to be endosomes based on their estimated size (0.5–2 µm), as well as in filopodia (Fig. 2, A–C). The degree of colocalization on putative endosomes was determined from deconvoluted confocal images by measuring enrichment of Integrin β1 on ErbB3-positive putative endosomes, relative to their adjacent surrounding cytoplasm (Fig. 2 D). A significant proportion, 32%, of the ErbB3-positive endosomal structures showed prominent Integrin β1 enrichment (>100%) (Fig. 2 D). Furthermore, traced

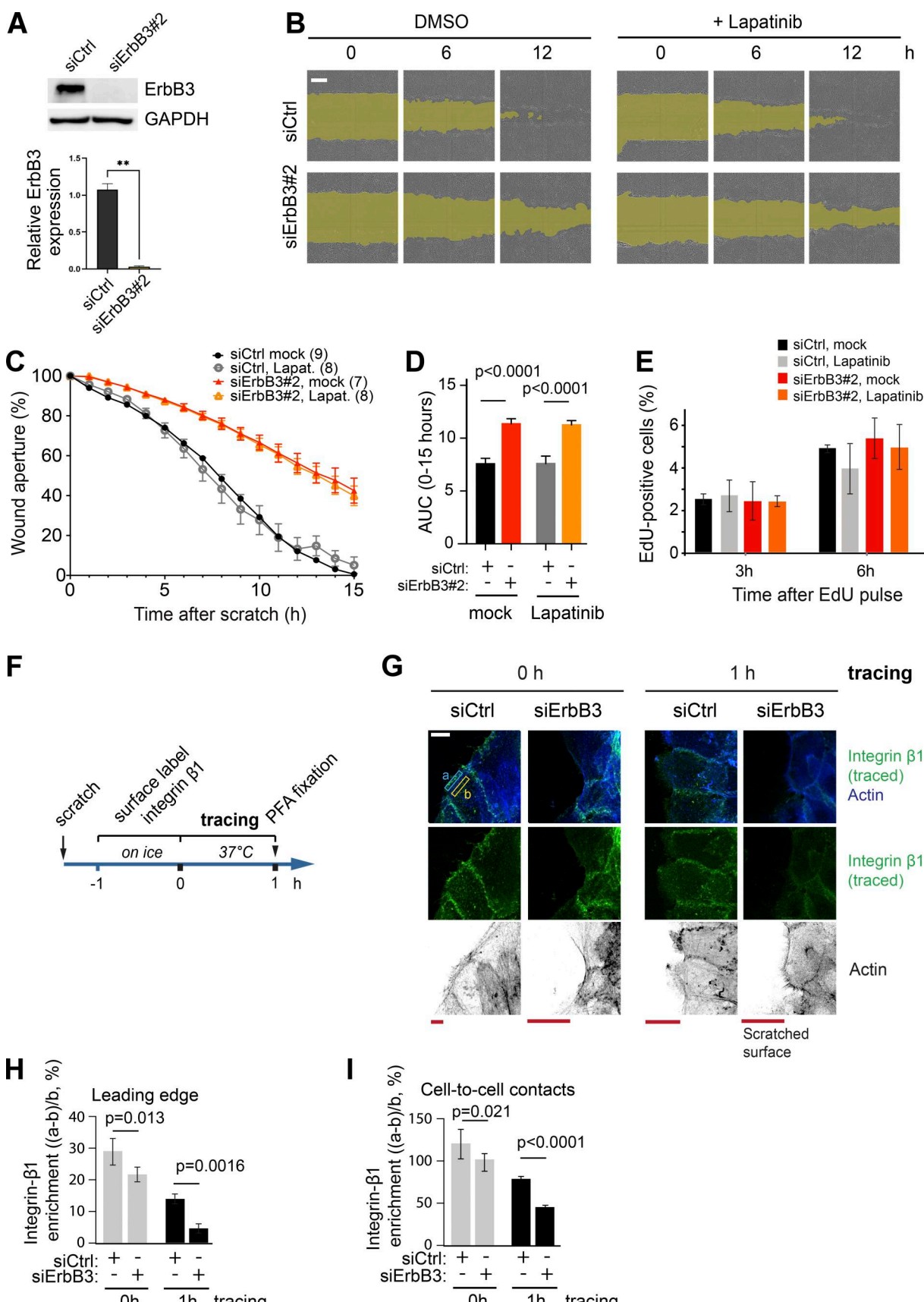

Figure 1. **Ligand-independent role of ErbB3 in epithelial sheet motility. (A)** Protein expression of ErbB3 and GAPDH (as a loading control) in the cell extract of MCF10A transiently transfected with control siRNA (siCtrl) or siRNA targeting ErbB3 (#2). Densitometric values of ErbB3 protein expression in three

biological replicates ± SEM of MCF10A cells are shown normalized to the respective siCtrl. Data are presented as mean values ± SEM. P values were determined by unpaired Student's *t* test. **P ≤ 0.01. **(B)** Scratch closure assay of MCF10A cells transiently transfected with siCtrl or siErbB3#2, cultured in serum-containing but growth factor–deprived media in the presence of DMSO (vehicle) or 1 µM of the EGFR/ErbB2 inhibitor lapatinib. The wound area is highlighted in yellow. **(C and D)** Quantification of the scratch aperture (C) or AUC (D) of samples treated as in B. Data are presented as mean values ± SEM; *n* values are indicated in the parenthesis. **(E)** Quantification of cell proliferation as incorporation of EdU for the indicated times in control or ErbB3 siRNA#2-transfected cells in the presence of DMSO (vehicle) or 1 µM lapatinib. Data are presented as mean values ± SEM; *n* = 3 independent experiments. **(F–I)** Confocal immuno-fluorescence imaging of surface-labeled Integrin β1 (green) or Actin (blue/black) on confluent sheets of MCF10A cells transfected with control siRNA (siCtrl) or siErbB3#2 at 0 or 1 h after labeling (G), as outlined in F. The boxed regions in G indicate the leading edge of the wound within a closing cell sheet. **(H)** Integrin β1 enrichment was determined as ((a−b)/b), where a = mean fluorescence intensity (Integrin β1) at a defined area of the leading edge (H) or cell–cell contact (I) and b = mean intensity of adjacent cytoplasm of the same area. Data are presented as mean values (>74 cells per data point) ± SEM; *n* = 3 independent experiments. Scale bars, 1 mm (B) and 10 µm (G).

surface-labeled ErbB3 and Integrin β1 partially colocalized with EHD1, a marker of recycling endosomes, at the endogenous level (Fig. S1 D), consistent with coordinated trafficking of the two.

We next investigated the putative role of ErbB3 in endocytic trafficking of Integrin β1 by conducting a recycling assay in MCF10A and in primary breast epithelial cells (prHMEC) cells, as outlined in Fig. 2 E. Briefly, the cell-surface pools of Integrin β1 were labeled with Alexa 488–conjugated antibody on ice, prior to 15-min incubation at 37°C, time during which the internalized integrins accumulate primarily in early endosomes (Roberts et al., 2001). The proportion of surface-labeling integrin antibody that remained on the cell surface was then quenched with an anti-Alexa 488 antibody on ice, as previously described (Arjonen et al., 2012). Cells were subsequently incubated at 37°C, and reemergence of internalized Integrin β1 at the basal membrane was followed by live-cell TIRF microscopy imaging. About 70% the chased Integrin β1 was detected at the plasma membrane already after 10 min, and ErbB3 silencing, using two independent and validated siRNAs, reduced recycling of Integrin β1 by 40–50% (Fig. 2, F and G; and Fig. S1, E and F). Similar results were obtained on prHMEC cells (Fig. 2, H and I; and Fig. S1, G and H). The initial amount of fluorophore-labeled Integrin β1 detected on the cell surface, prior to tracing, was 18–19% lower upon ErbB3 knockdown when compared to MCF10A control (Fig. S1, I and J), but unchanged in prHMEC (Fig. S1, K and L), as determined by confocal imaging. Notably, Integrin β1 protein levels did not change significantly upon ErbB3 depletion in MCF10A (Fig. S2 A) and prHMEC cells (Fig. S2 B), yet ErbB3 ablation significantly increased *ITGB1* mRNA levels in MCF10A cells (Fig. S2 C).

The ErbB3 pseudoreceptor is currently thought to exclusively act in a dimer configuration with other members of the ErbB family. Thus, in order to determine whether its preferred dimerization partners EGFR or ErbB2 play a part in endocytic recycling of Integrin β1 alongside ErbB3, we subjected MCF10A cells to RNAi-mediated depletion of EGFR or ErbB2 prior to assaying endocytic recycling of Integrin β1. Depletion of EGFR or ErbB2 did not impair Integrin β1 recycling (Fig. 2, J and K; and Fig. S2, A and E–H), suggesting that ErbB3 acts independently of its canonical heterodimer partners to promote endocytic recycling of Integrin β1. Of note, the ectopic expression of siRNA-resistant ErbB3 restored the surface pool of traced surface-labeled Integrin β1 in ErbB3-depleted MCF10A cells, as visualized by confocal imaging (Fig. S2, I and J), indicating that off-target effects of the ErbB3 siRNA do not underlie the

observed recycling defect. ErbB4 was excluded from these experiments due to the lack of its expression in these cells, as previously reported (Haskins et al., 2014; Wali et al., 2014). Of importance, the recycling assays were conducted in a cell culture medium devoid of growth factors. Under these conditions, we did not detect tyrosine phosphorylation of immunoprecipitated ErbB3 by immunoblotting with an antibody that recognizes global phosphotyrosine (Fig. S2 K), nor did we detect phosphorylation of ErbB3 on Tyr1289, or significant activation of the AKT and ERK1/2 kinases, typically observed downstream of ligand-stimulated ErbB3 (Fig. S2 A). These results indicate that ErbB3 promotes Integrin β1 recycling in a manner that does not require ligand-induced receptor signaling.

The finding that ErbB3 regulates endocytic recycling of Integrin β1 in nonmalignant breast epithelial cells raises the question of whether this mechanism is also relevant in cancer cells overexpressing ErbB3. *ErbB3* mRNA is highly expressed in BRCA patients in comparison with healthy controls (extracted from TCGA datasets; Fig. S3 A), which correlates significantly with a poorer overall survival of BRCA patients (Fig. S3 B). Moreover, higher mRNA levels of *ErbB3* were associated with the lack of response of BRCA patients to chemotherapies (Fig. S3 C), which is consistent with the known protumorigenic role of ErbB3 in BRCA (Pradeep et al., 2014; Smirnova et al., 2012; Tiwary et al., 2014; Yoshioka et al., 2010; Gaborit et al., 2016). Notably, a meta-analysis of 12 studies found that elevated ErbB3 protein expression was associated with worse overall survival not only in BRCA patients, but also in patients with colorectal, gastric, melanoma, ovarian, head and neck, pancreatic, and cervical cancers (Ocana et al., 2013). Accordingly, we found *ErbB3* mRNA levels were significantly higher in the luminal BRCA cell line, MCF7, when compared to the nonmalignant breast epithelial cells MCF10A and the prHMEC (Fig. S1 C). To assess the putative role of ErbB3 in Integrin β1 recycling in malignant MCF7 cells, we surface-labeled Integrin β1 on ice and allowed internalization at 37°C, followed by quenching of the remaining surface labeling and subsequent tracing of internalized Integrin β1 in the cells transfected with control siRNA (siCtrl) or siErbB3 cells, in line with our prior recycling assays. We found that Integrin β1 internalization was less effective in MCF7 cells as compared to MCF10A and prHMEC with limited detectable return of internalized Integrin β1 to the cell surface of both control and ErbB3-depleted cells, precluding reliable assessment of Integrin β1 recycling in these cells (Fig. S3, D and E). This observation may reflect differing Integrin β1 trafficking

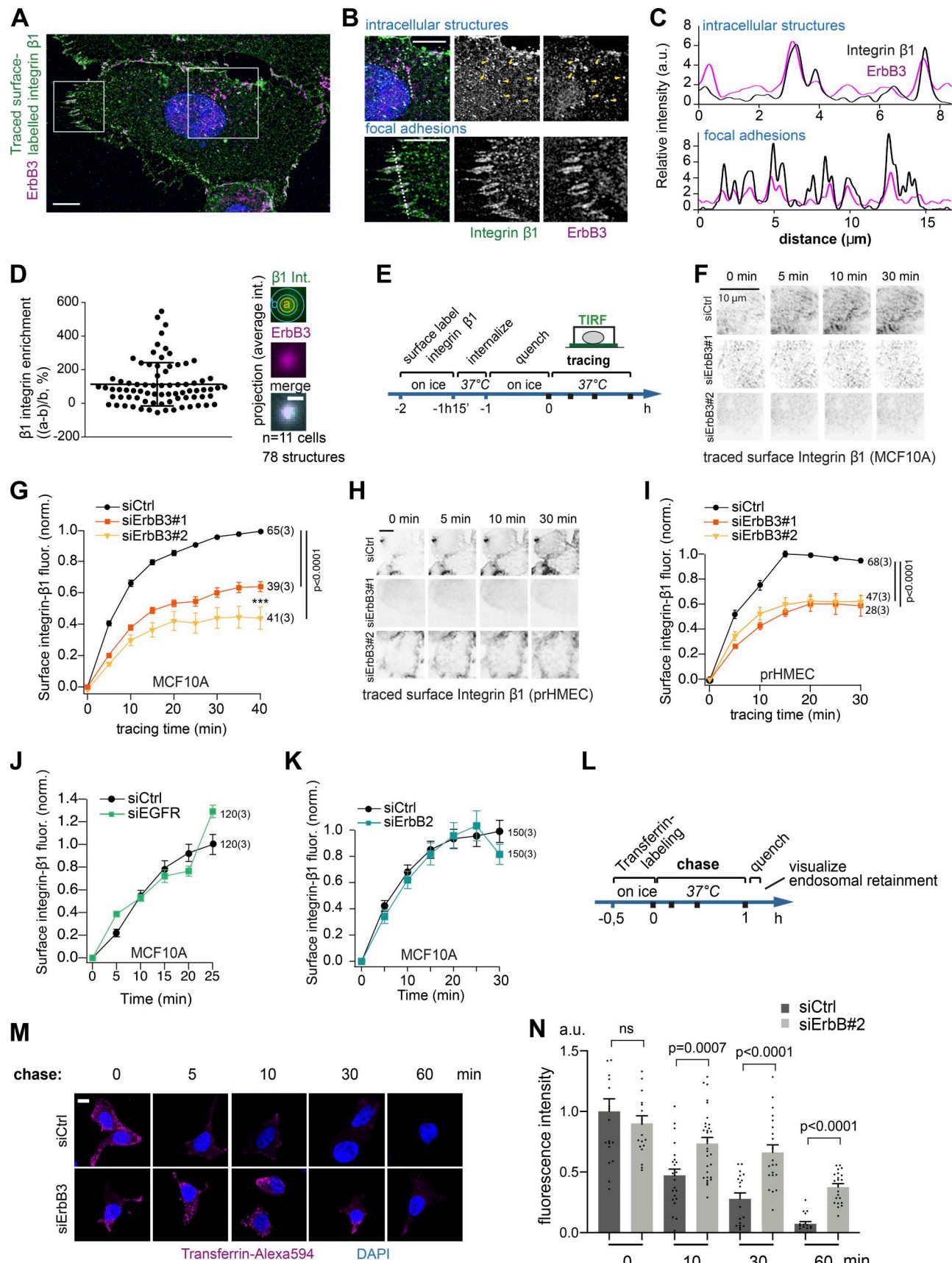

Figure 2. **ErbB3 promotes Integrin β1 and TfR endocytic recycling. (A–D)** Confocal immunofluorescence imaging of traced surface-labeled Integrin β1 and ErbB3: MCF7 cells were labeled on ice with an Alexa 488–conjugated anti-Integrin β1 antibody prior to incubation for 15 min at 37°C to allow Integrin

β1 internalization, and subsequent cell fixation and immunolabeling of ErbB3 (red) and counterstaining with DAPI (blue). Note that B represents the magnified (squared) regions of A. **(C)** Histogram of fluorescence intensities along dotted lines indicated in B. **(D)** Analysis of Integrin β1 and ErbB3 colocalization. Integrin β1 enrichment in ErbB3-positive intracellular structures (0.5–2 μm diameter) was determined by the formula (a–b)/b, where a is the Integrin β1 intensity at ErbB3-positive structures, and b is the adjacent intensity (background) for each structure. Average intensity projections of all analyzed structures are shown on the right-hand side. **(E)** Schematic outline of Integrin β1 recycling assays conducted in F–K after transfection with siRNA against the indicated targets. Integrin β1 surface pool was labeled with an Alexa 488–conjugated antibody and allowed to endocytose. Fluorophore label remaining on the cell surface was quenched with an anti-Alexa 488 antibody, prior to visualization of traced Integrin β1 reemerging on the cell surface by live-cell TIRF microscopy. **(F)** Representative TIRF microscopy images of Integrin β1 from peripheral areas of MCF10A cells transiently transfected with control siRNA (siCtrl), siErbB3#1, or siErbB3#2. **(G)** Quantifications of recycled Integrin β1 performed on indicated number of cells (outside of brackets on the right-hand side of graphs), from three independent experiments and shown as Alexa 488 intensity normalized between 0 and 1, with the control as reference where $F_{norm}=((F_{max}-F_{min})/(F-F_{min}))$. **(H)** Representative TIRF microscopy images of Integrin β1 from prHMEC cells transiently transfected with siCtrl, siErbB3#1, or siErbB3#2. **(I)** Quantifications of recycled Integrin β1 in prHMEC performed as described in E. **(J and K)** Quantified Integrin β1 recycling, after siRNA-mediated depletion of either EGFR (J) or ErbB2 (K). Data are presented as mean values ± SEM, and P values were determined by two-tailed paired Student's t test. ns, nonsignificant. **(L)** Schematic outline of transferrin recycling assays. **(M)** Confocal imaging of Alexa 594–conjugated transferrin chased with unlabeled holo-transferrin for the indicated times in MCF7 cells. **(N)** Quantification of Alexa 594 fluorescence intensity in cells treated as in M (n > 17 cells for each data point from three experiments) normalized against the control siRNA-treated samples, 0-h time point of each independent experiment. Scale bars, 7 μm (A), 4 μm (B), 10 μm (F, H, and M), and 0.5 μm (D).

dynamics and underlying mechanisms between malignant and normal breast epithelial cells.

Notwithstanding, we asked whether ErbB3 plays a more general role in regulating endosomal recycling beyond Integrin β1, by assessing the effect of ErbB3 depletion on recycling of the TfR that has been extensively studied as a generic recycling cargo that undergoes continuous internalization and recycling with negligible sorting to late endosomes and lysosomes (Sönnichsen et al., 2000). TfR on the cell surface of MCF7 transfected with siCtrl or siErbB3 (Fig. S3 F) was tagged with Alexa 594–conjugated transferrin on ice for 30 min. Unbound, labeled transferrin was washed away, and cells were incubated at 37°C in the presence of unlabeled transferrin for up to 1 h. Cells were brought back on ice and subjected to low pH treatment to quench Alexa 594 that had returned to the cell surface, followed by visualization of chased transferrin retained in endosomes by fluorescence imaging (Fig. 2, L–N). After 1 h at 37°C, most TfR was recycled back to the plasma membrane as indicated by the shedding of labeled transferrin from the cells. However, half of the chased transferrin was still retained in endosomes in cells transfected with siErbB3 (Fig. 2, M and N). In conclusion, ErbB3 promotes recycling of both Integrin β1 and TfR, consistent with a general role of ErbB3 in regulating endocytic recycling.

## ErbB3 colocalizes with Rab4-positive recycling endosomes and impairs EV release

Our data show a marked reduction in Integrin β1 recycling in ErbB3-depleted cells already 10 min after onset of tracing (Fig. 2), which corresponds to the reported peak time of colocalization of Rab4 with traced TfR, preceding Rab11 and TfR colocalization that peaks at 30 min (Lindsay et al., 2002; Sönnichsen et al., 2000). To determine which recycling compartment ErbB3 preferentially resides in, ErbB3-mCherry was co-expressed with either GFP-Rab4 or GFP-Rab11 in MCF10A cells, followed by treatment with either primaquine (PQ), an inhibitor of endocytic recycling (Arjonen et al., 2012; Woods et al., 2004), or vehicle alone for 10 min and subsequent visualization of the proteins by confocal imaging (Fig. 3 A). Colocalization was analyzed as enrichment of ErbB3 fluorescence in Rab4- or Rab11-positive compartments (Fig. 3 B) or visualized as average intensity projections of all structures (Fig. 3 C). A large

majority of ErbB3-mCherry did not colocalize with GFP-Rab4– or GFP-Rab11–positive structures in the absence of PQ treatment. However, upon PQ treatment for 10 min, ErbB3 enrichment in Rab4-positive structures increased 6.2-fold (P = 0.0003), while no significant enrichment was observed in Rab11-positive structures (Fig. 3 B). The preferred ErbB3 colocalization with Rab4-positive rather than Rab11-positive recycling endosomes and requirement for ErbB3 for early endocytic recycling (within 10 min) (Bridgewater et al., 2012; Tomas et al., 2014) suggest that ErbB3 is required for early Rab4-dependent endocytic recycling, although its involvement in other Rab-dependent pathways is not excluded.

We next asked whether ErbB3 might play a broader role in regulating exocytic trafficking, beyond its known involvement in endocytic recycling. Thus, we used a well-established system to analyze exocytic trafficking from the endoplasmic reticulum (ER) using ectopically expressed vesicular stomatitis virus, VSV-G-ts-GFP (Hirschberg et al., 1998). The temperature-sensitive form of VSV-G accumulates in the ER at 40°C, and upon shift to 32°C, this protein traffics via the trans-Golgi network (TGN) to the cell surface (Fig. 3 D). Of note, ErbB3 silencing in MCF10A cells did not affect the amount of VSV-G that was transported from the ER to the cell surface after 30 min, and 1 or 2 h at permissive temperature as monitored by immunoblotting of isolated surface-biotinylated proteins (Fig. 3, E and F; and Fig. S4 A). Immunoblotting against the transmembrane protein Muc1 on the isolated biotinylated proteins controlled for equal sampling. To complement the surface-biotinylation assay, traced VSV-G-ts-GFP was visualized by confocal imaging. In both siCtrl- or siErbB3-transfected MCF10A cells, VSV-G-ts-GFP exhibited predominant localization in the ER at 40°C. Following the shift to 32°C, VSV-G-ts-GFP was first detected in the GM130[+] TGN at 30 min and in the E-cadherin[+] membrane compartment at 1 h, in both control and ErbB3 siRNA-transfected cells (Fig. S4 B), suggesting that ErbB3 does not impact trafficking from the TGN to the plasma membrane.

Interestingly, Rab4 has been proposed to restrict exosome biogenesis by promoting early endosomal recycling (Zhang et al., 2022). To elucidate whether impaired endocytic recycling caused by loss of ErbB3 could also influence EV release, we isolated small EVs secreted from MCF10A cells transfected or

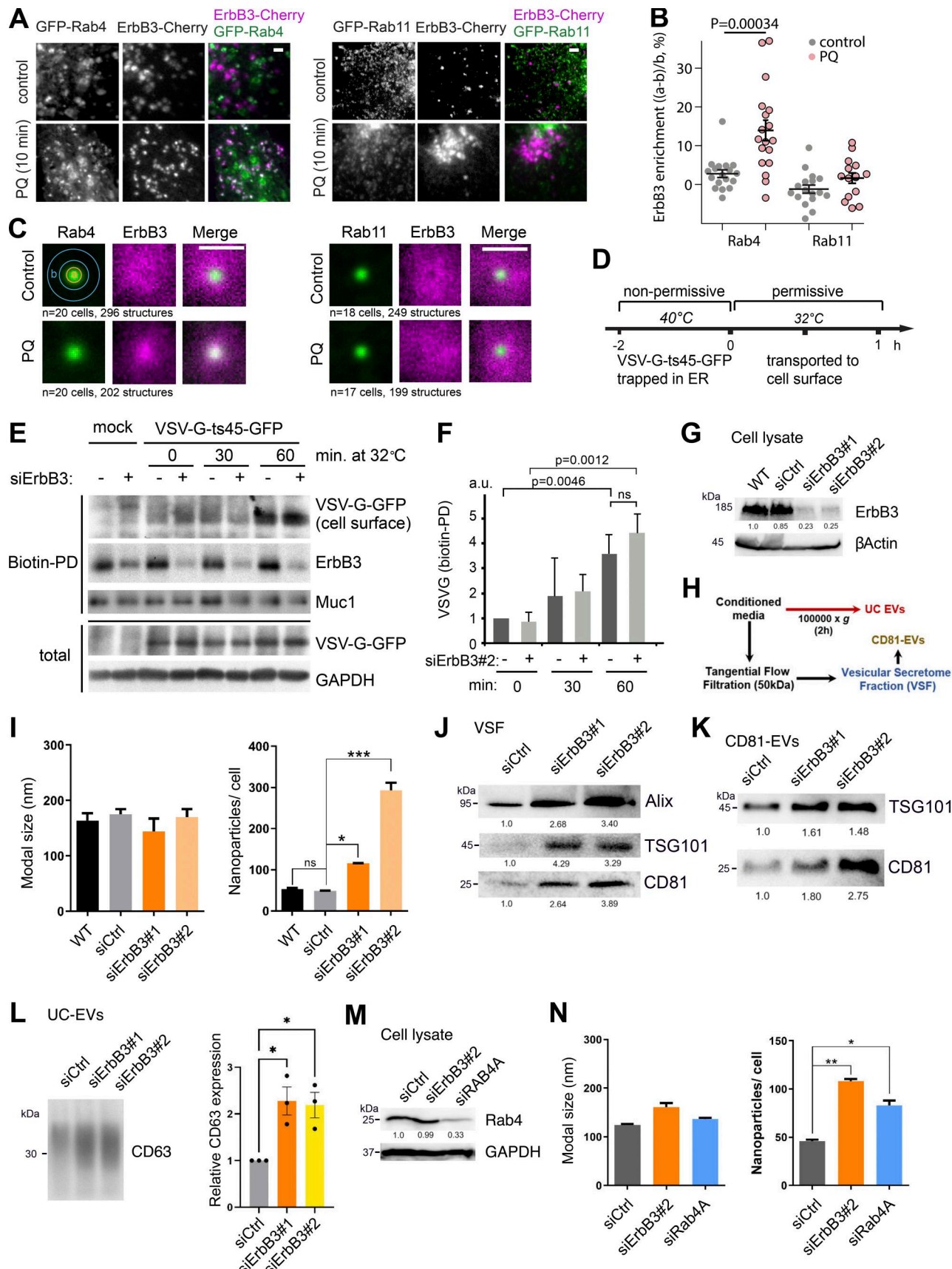

Figure 3. **Loss of ErbB3 directs Integrin β1 for secretion as EV cargo. (A)** Confocal imaging of ErbB3-mCherry and indicated Rab marker expressed in MCF10A cells, with or without prior treatment with the recycling inhibitor PQ. **(B)** Analysis of ErbB3-mCherry colocalization with Rab4 or Rab11. The relative

ErbB3 enrichment at the Rab-positive structures was determined by the formula (a–b)/b, where a is the ErbB3-mCherry intensity of the center in Rab4 structures, and b is the adjacent volume (background) for each structure. Each data point represents the minimum average of 20 structures in one cell. P values were determined using unpaired two-tailed Student's *t* test. **(C)** Average projections of all analyzed (indicated number) GFP-Rab4– or GFP-Rab11–positive structures from the indicated number of cells (three independent experiments). **(D)** Experimental outline of the VSV-G trafficking experiments in MCF10A cells transiently transfected with control siRNA (siCtrl) or siErbB3#2. **(E)** Immunoblot analysis of the surface pool of VSV-G-ts45-GFP (pull-down of surface-biotinylated VSV-G-ts45-GFP), after its release from the ER at permissive temperature for the indicated times. **(F)** Quantification of VSV-G-GFP in biotin pull-downs; normalized levels were determined by immunoblot band intensities (*n* = 3 independent experiments). **(G)** ErbB3 and β-Actin (as a loading control) protein expression in MCF10A protein extracts of WT and transiently transfected cells with the indicated siRNA. **(H)** Scheme of the EV enrichment protocol by (1) UC or (2) tangential flow filtration, followed by CD81 immunoaffinity capture of EVs from conditioned media of breast epithelial cells. **(I)** EVs enriched in the VSF released by the indicated MCF10A cells were quantified by NTA in terms of nanoparticle modal size (nm) and number after normalization to the total cell number. **(J and K)** Protein expression levels of the indicated EV marker proteins in EV extracts enriched as VSF (J) or CD81-EVs (K) derived from MCF10A cells transiently transfected with the indicated siRNAs; average densitometric values of at least three independent biological replicates were normalized to siCtrl. **(L)** CD63 protein expression level in EV extracts enriched by UC derived from MCF10A cells transiently transfected with the indicated siRNAs. Densitometric quantification of CD63 expression is presented as the mean ± SEM from three biological replicates, normalized to the corresponding siCtrl condition. **(M)** Rab4A and GAPDH (as a loading control) protein levels in MCF10A cells transiently transfected with the indicated siRNAs. **(N)** EVs enriched in the VSF released by MCF10A transiently transfected with the indicated siRNAs were quantified by NTA in terms of nanoparticle modal size (nm) and number upon normalization to the total cell number. Data were presented as mean values ± SEM. P values were determined by two-tailed paired Student's *t* test or one-way ANOVA, followed by multiple paired comparisons conducted by means of Bonferroni's posttest method. *P ≤ 0.05; **P ≤ 0.01; ***P ≤ 0.001; ns, nonsignificant. Scale bars, 1 µm.

not with ErbB3 or nontargeting siRNAs (Fig. 3 G) by an established protocol of tangential flow filtration, followed by immunoaffinity selection of the tetraspanin CD81 or by ultracentrifugation (UC) (Fig. 3 H) (Rodrigues-Junior et al., 2019; Théry et al., 2006). EVs enriched in the vesicular secretome fraction (VSF) were visualized by transmission electron microscopy (TEM), validating the presence of membrane-surrounded nanoparticles of 75–200 nm in size and consistent with small EVs (Fig. S4 C). Nanoparticle tracking analysis (NTA) revealed that the number of secreted EVs increased significantly in cells with ErbB3 silenced by two independent siRNAs, in comparison with siCtrl, while the EV modal size was not affected (Fig. 3 I and Fig. S4 D). Additionally, at equal EV fraction sample volume enriched as VSF, CD81-EVs, or UC EVs from ErbB3-depleted cells, the expression of established EV markers, including the endosomal sorting complexes required for transport–associated molecules ALIX and TSG101 or the tetraspanins CD81 and CD63, was increased (Fig. 3, J–L), validating the NTA results. Similarly, ErbB3 loss in prHMEC cells (Fig. S4 E) did not impact the modal size of EVs, but significantly enhanced the EV release (Fig. S4 F). Furthermore, Rab4A silencing in MCF10A cells (Fig. 3 M) significantly increased EV secretion (Fig. 3 N), which is in line with the previous findings of Zhang et al. (2022). These results were also reproduced in MCF7 cells, since ErbB3 or Rab4A silencing (Fig. S4 H) significantly boosted EV secretion (Fig. S4, H–J) (Fig. S4, H and I). Collectively, our data reveal a novel role of ErbB3 in promoting early endocytic recycling of different cell surface–derived receptors, and restricting EV secretion in breast epithelial cells.

## ErbB3 loss directs Integrin β1 for lysosomal degradation and secretion in EVs

We sought to trace the fate of internalized Integrin β1 that fails to recycle back to the cell surface in the absence of ErbB3. Toward this end, MCF10A cells were labeled with an Alexa 488–conjugated surface-binding Integrin β1 antibody on ice, followed by incubation at 37°C for 15 min to allow Integrin β1 internalization and subsequent quenching on ice of the Integrin β1 that remained on the cell surface (0 min). Finally, the cells were

returned to 37°C and the fate of the internalized pool of Integrin β1 was visualized by confocal fluorescence microscopy for up to half an hour (Fig. 4, A and B). Already after 15 min of tracing, Integrin β1 levels decreased by 42.4% in ErbB3-depleted cells at which time no significant change was observed in cells transfected with siCtrl (Fig. 4, B and C). Notably, traced Integrin β1 levels were somewhat reduced already at the onset of the tracing (0 min) in ErbB3-depleted cells compared with control cells (13.3%) (Fig. 4 C), which is probably due to accelerated Integrin β1 turnover in ErbB3-depleted cells evident within the 15-min internalization step that precedes quenching and the onset of tracing.

To further assess the impact of ErbB3 on Integrin β1 turnover, we subjected control or ErbB3 siRNA-transfected MCF10A cells to treatment with the protein synthesis inhibitor cycloheximide (CHX; up to 8 h) and monitored Integrin β1 levels by immunoblotting (Fig. S5, A and B). While Integrin β1 levels were slightly reduced for up to 8 h in the presence of CHX in the control cells, their levels were significantly decreased by 72% after 8 h of CHX treatment in ErbB3-depleted cells (Fig. S5, A and B). To evaluate directly Integrin β1 stability, we performed a pulse-chase analysis of Integrin β1 turnover. Control or ErbB3 siRNA-transfected MCF10A cells were labeled with $^{35}$S-methionine/cysteine for 1 h, followed by chasing with unlabeled amino acids at 37°C. Integrin β1 was subsequently immunoprecipitated, and the protein-incorporated $^{35}$S-label was determined by radiography. The data show that cells transfected with two independent siRNAs against ErbB3 exhibited a significantly accelerated Integrin β1 turnover (49% or 84% decrease), compared with control cells (Fig. 4, D and E).

The Integrin β1 turnover is slow; yet, limited degradation of this protein has been found to occur in the lysosomes (De Franceschi et al., 2015). To better characterize Integrin β1 turnover, we traced surface-labeled Integrin β1 in control or ErbB3 siRNA-transfected MCF10A cells by confocal fluorescence microscopy, in the presence or absence of chloroquine, which impairs phagolysosomal fusion (Fig. 4 F). Chloroquine treatment caused traced Integrin β1 to accumulate in cells bordering the migratory front of wounded monolayers of both wild-type (WT)

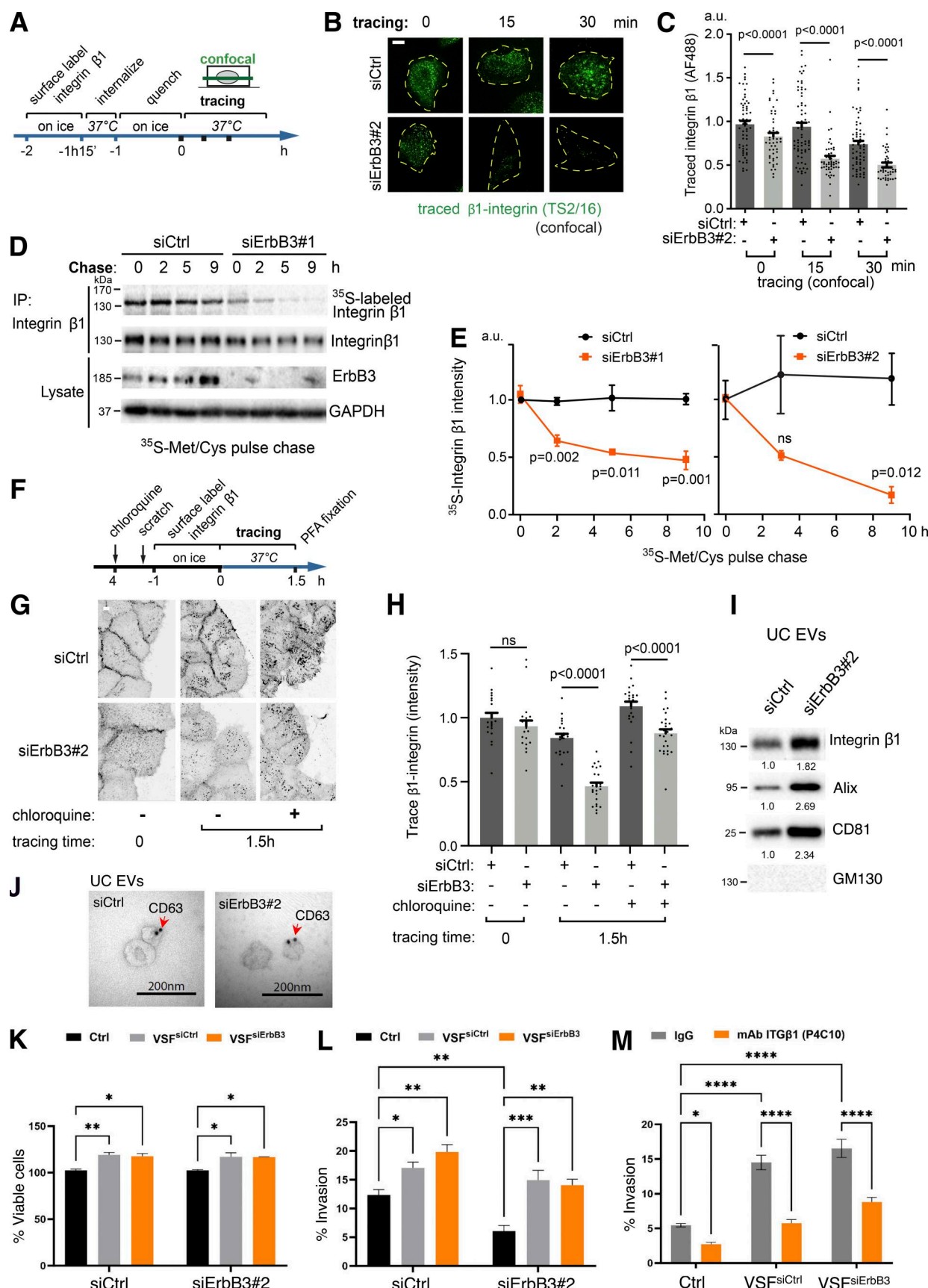

Figure 4. **ErbB3 loss directs Integrin β1 toward lysosomal degradation or EV secretion. (A)** Schematic outline of traced internalized integrin. **(B)** Confocal immunofluorescence imaging of traced internalized Integrin β1 in MCF10A cells transiently transfected with control siRNA (siCtrl) or siErbB3#2.

**(C)** Quantification of immunofluorescence intensity of internalized Integrin β1 that was traced for the indicated times after a 15-min internalization step (n > 32 cells per data point from 5 independent experiments). **(D)** Determination of Integrin β1 turnover by pulse-chase metabolic labeling: control or ErbB3 siRNA#1-transfected MCF10A cells were pulse chase–labeled with radioactive ($^{35}$S) methionine and cysteine. Radiolabeled Integrin β1 was visualized by radiography of immunoprecipitates (upper panel). Cell lysates and immunoprecipitates were analyzed by immunoblotting. **(E)** Quantification of pulse-chased $^{35}$S-labeled Integrin β1 (left panel, siErbB3#1; and right panel, siErbB3#2) (n = 4 independent experiments). **(F and G)** Confocal immunofluorescence imaging of surface-labeled Integrin β1 (using an Alexa 488–conjugated anti-Integrin β1 antibody), prior to (0 h) or after tracing at 37°C for 1.5 h. A scratch was inflicted prior to antibody incubation. Note that application of the lysosome inhibitor chloroquine caused Integrin β1 accumulation in intracellular vesicular compartments in MCF10A transiently transfected with siCtrl or siErbB3#2. **(H)** Quantification of Integrin β1 fluorescence intensity in cells bordering the migratory front in samples treated as in G, showing that chloroquine restored Integrin β1 levels in ErbB3-depleted cells. Data were presented as mean values ± SEM; n = 19–27 cells per data point from 3 independent experiments. P values were determined by two-tailed paired Student's t test. ns, nonsignificant. **(I)** Protein expression of Integrin β1 and the indicated EV markers in the UC EV protein extract; average densitometric values of at least three independent biological replicates were normalized to siCtrl. **(J)** Representative TEM micrographs of EVs enriched by ultracentrifugation (UC EVs) from control or ErbB3-depleted MCF10A cells and stained with gold-conjugated anti-CD63 antibody (CD63-EVs are indicated by red arrows). **(K)** Cell viability assay measured with PrestoBlue Cell Viability Reagent in MCF10A cells transiently transfected with siCtrl or siErbB3#2 incubated with EV-free media as vehicle (Ctrl) or 1 × 10$^9$ nanoparticles of EVs secreted from MCF10A cells transiently transfected with control (VSF$^{siCtrl}$) or ErbB3-siRNA (VSF$^{siErbB3}$) for 48 h. **(L)** Matrigel invasion assay in transwells with MCF10A cells stimulated with EV-free media (Ctrl) or incubated with equal number of VSF$^{siCtrl}$ or VSF$^{siErbB3}$ as described in K for 16 h. **(M)** Matrigel invasion assay in transwells with MCF10A WT cells prestimulated for 48 h with EV-free media (Ctrl) or incubated with equal number of VSF$^{siCtrl}$ or VSF$^{siErbB3}$ in the presence of 5 ug/ml IgG or Integrin β1 monoclonal antibody (mAb; P4C10) for an additional 16 h. The data in K–M are presented as mean values of three biological replicates ±SEM and P values shown based on two-way ANOVA, followed by multiple paired comparisons conducted by means of Bonferroni's posttest method (*P ≤ 0.05; **P ≤ 0.01; ***P ≤ 0.001; ****P ≤ 0.0001). Scale bars, 15 μm (A), 10 μm (G), 200 nm (J).

and ErbB3-depleted cells (Fig. 4 G). Quantification of fluorescence intensity showed that while the level of traced Integrin β1 was reduced in the ErbB3-depleted cells relative to siCtrl cells (by 44.8%, P < 0.0001), consistent with its increased turnover, chloroquine treatment significantly restored traced Integrin β1 levels (Fig. 4 H). Our results indicate that elevated Integrin β1 turnover caused by ErbB3 loss is in part due to increased lysosomal degradation.

It has been reported that EV-associated integrins assist organotropic metastasis, and in particular, Integrin β1 was found to be associated with increased tumor invasiveness and metastatic potential of BRCA cells (Hoshino et al., 2015; Weaver et al., 1997). In this context, we sought to assess whether the sorting of Integrin β1 into EVs could be affected by ErbB3 silencing. We found that siRNA-mediated depletion of ErbB3 in MCF10A cells (Fig. S5 C) increased the amount of UC EV-associated Integrin β1, as detected by immunoblotting (Fig. 4 I). The prevalence of EVs in the UC samples was further validated by the increased presence of ALIX and CD81, in addition to the absence of the Golgi marker GM130 (Fig. 4 I), and membranous nanovesicles labeled with CD63 immunogold were visualized by TEM (Fig. 4 J), attesting to the EV integrity. These results imply that the increased turnover of Integrin β1 in ErbB3-silenced cells occurs due to both increased shedding in EVs and lysosomal degradation.

We sought to validate that EVs could be involved in promoting viability or invasive properties of the breast epithelial cells. We therefore stimulated transfected MCF10A cells with normalized concentrations of VSF (1 × 10$^9$ nanoparticles/ml) derived from either siCtrl or siErbB3 cells for 48 h and assessed cell viability by PrestoBlue fluorescence or invasion into ECM (Matrigel) for an additional 16 h. We found that equalized amounts of EVs from either control or ErbB3-depleted cells, to the same extent, modestly promoted cell viability in the recipient cells (Fig. 4 K). Accordingly, ErbB3 absence reduced MCF10A motility in comparison with siCtrl cells (Fig. 1, A–D and Fig. 4 L), whereas the EV pools substantially enhanced the motility of not only the recipient ErbB3-depleted cells, but also the recipient siCtrl cells into a Matrigel matrix (Fig. 4 L). Together, our results

indicate that while ErbB3-silenced MCF10A cells exhibited lower motility, these cells are sensitive to external stimuli received from the EV cargo that elevate invasive behavior, while simultaneously releasing a larger amount of EVs with enhanced motility properties to neighboring WT cells. Furthermore, to test whether Integrin β1 was essential for promoting EV-induced invasive behavior, WT MCF10A cells were pretreated with EVs in the presence of 5 μg/ml IgG or Integrin β1 blocking monoclonal antibody (mAb; clone P4C10) for 48 h prior to the transfer into ECM (Fig. 4 M). We found that blocking Integrin β1 significantly reduced invasiveness of the cells both in the presence and in the absence of EVs derived from MCF10A (Fig. 4 M). Moreover, MCF10A cells transfected with siCtrl or siErbB3 were both sensitive to the inhibitory action of P4C10. The data are consistent with EV-derived Integrin β1 playing a role in promoting invasive behavior of the MCF10A cells, in line with its known functions (Sun et al., 2023; Weaver et al., 1997).

**ErbB3 promotes assembly of the Arf6–Rabaptin5–GGA3 sorting complex and stability of Rabaptin5 and GGA3**

Interestingly, selected components of the endosomal machinery that could be implicated in the Rab4-dependent early recycling of β integrins such as ADP-ribosylation factor-binding protein 3 (also known as Golgi-associated gamma adaptin ear containing ADP-ribosylation factor–binding protein 3, GGA3) (Arjonen et al., 2012) and Rab GTPase-binding effector protein 1 (Rabaptin5 or RABPT5) (Christoforides et al., 2012; Mattera et al., 2003) colocalized with ErbB3 in MCF7 cells (Fig. S5, D and E). Thus, to explore whether ErbB3 could interact with these proteins, we immunoprecipitated endogenous ErbB3 from MCF10A cell extracts and examined coprecipitates by immunoblotting. We found that ErbB3 coprecipitated efficiently with both GGA3 and Rabaptin5 at endogenous levels (Fig. 5 A), and the abundance of this complex was increased after a 30-min treatment with PQ, consistent with these proteins interacting in the recycling compartment. Thus, GGA3 and Rabaptin5 are plausible effectors of ErbB3 in early recycling endosomes.

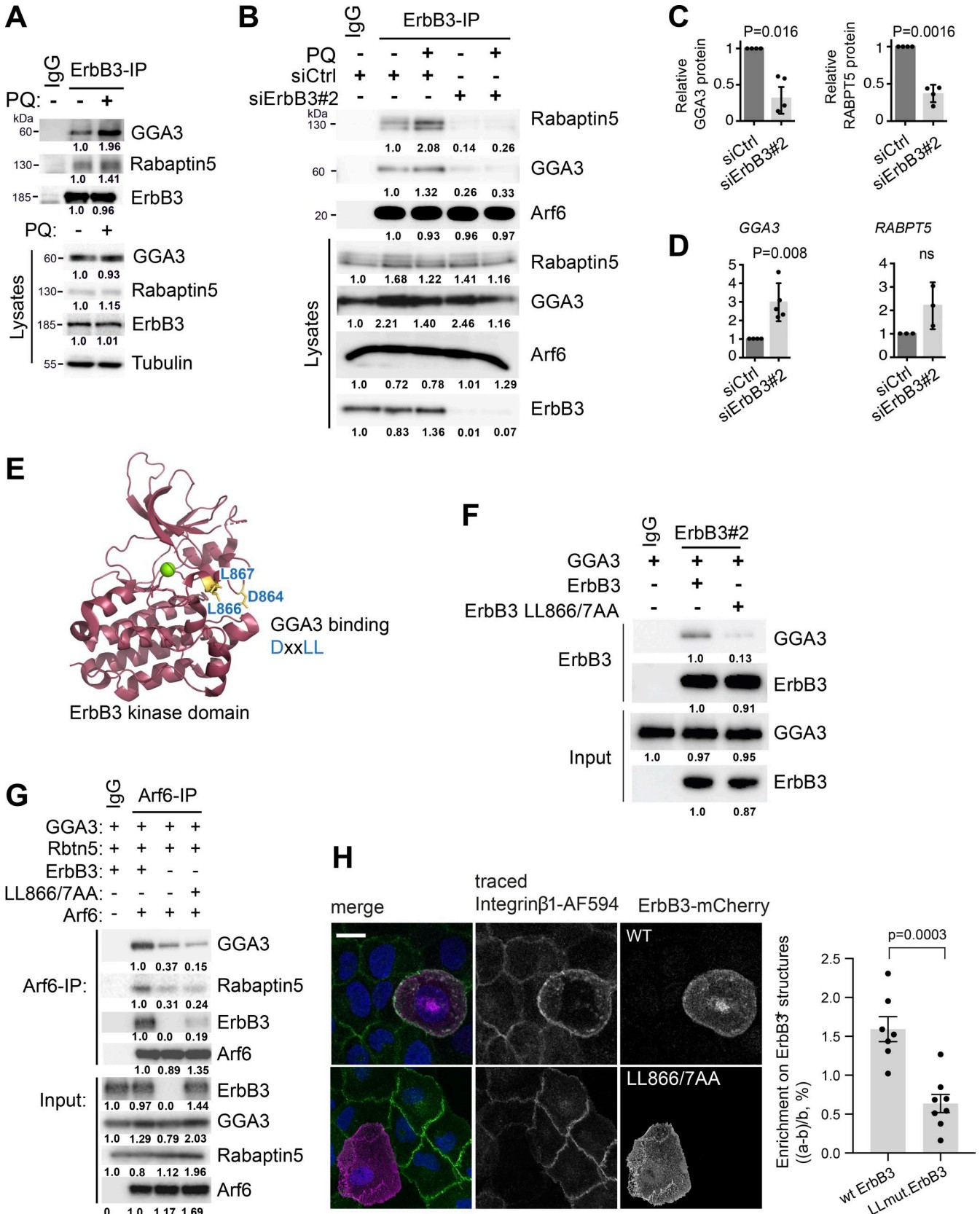

Figure 5. **ErbB3 scaffolds assembly of the Arf6–GGA3–Rabaptin5 endosomal sorting complex. (A)** Immunoblotting of ErbB3 immunoprecipitates or input cell lysates, after 30-min treatment with PQ, showing endogenous binding of ErbB3 with GGA3 and Rabaptin5 that increases upon PQ treatment, and the presumed accumulation of recycling endosomes (representative of three independent experiments). IgG was used as a negative control for immunoprecipitations. **(B)** Immunoblotting of Arf6 immunoprecipitates or input cell lysates, transfected with the indicated siRNAs, following PQ or vehicle

treatment for 10 min. **(C and D)** Quantification of GGA3 and Rabaptin5 protein levels (C: immunoblotting) and mRNA levels (D: RT-qPCR) in ErbB3 siRNA-transfected MCF10A cells relative to control cells (*n* = 4 experiments for protein and *n* = 3 for mRNA). Data are presented as mean values ± SEM. P values were determined by unpaired Student's *t* test. **(E)** Structural model highlighting the putative GGA3-binding motif 864-DxxLL-867 in the ErbB3 kinase domain. **(F)** Immunoblotting of ErbB3 immunoprecipitates or input cell lysates, after ectopic expression of ErbB3 or the ErbB3 LL866/867AA mutant with GGA3 in HEK293T cells. **(G)** LL866/867AA mutation compromises the ability of ErbB3 to promote assembly of the Arf6–GGA3–Rabaptin5 sorting complex: immunoblotting of Arf6 immunoprecipitates or input cell lysates, following ectopic expression of Arf6, GGA3, and Rabaptin5, with or without ErbB3 or ErbB3-LL866/867AA. Average densitometric values of at least two independent biological replicates were normalized to the respective controls and to the loading control protein when appropriate. **(H)** Confocal imaging of WT or LL866/7AA mutant ErbB3-mCherry and Alexa 594–conjugated Integrin β1 in MCF10A cells. The analysis of ErbB3-mCherry colocalization with Integrin β1 shows the relative ErbB3 enrichment at the Integrin β1–positive structures, as determined by the formula (a–b)/b, where a is the ErbB3-mCherry intensity of the center in Integrin β1 structures, and b is the adjacent volume (background) for each structure. Each data point represents the minimum average of 20 structures in one cell. P values were determined using unpaired two-tailed Student's *t* test. Scale bars, 15 μm (H).

GGA3 has been shown to interact with ADP-ribosylation factor 6 (Arf6) to drive recycling of different transmembrane proteins (Parachoniak et al., 2011; Ratcliffe et al., 2016) and with Rabaptin5 to regulate TGN cargo export and endosomal tethering/fusion events (Mattera et al., 2003; Miller et al., 2003). We therefore set out to determine the influence of ErbB3 on the Arf6–GGA3–Rabaptin5 adaptor network (Fig. S5 F). MCF10A cells were transfected with control or ErbB3 siRNA, prior to treatment with PQ or vehicle alone for 10 min. Harvested cell lysates were subsequently subjected to Arf6 immunoprecipitation and immunoblotting for bound GGA3 and Rabaptin5 (Fig. 5 B). The data revealed that loss of ErbB3 indeed reduced the amount of GGA3 and Rabaptin5 associated with Arf6 both in the absence and in the presence of PQ. Notably, PQ treatment of siCtrl cells led to a stronger interaction of Arf6 with GGA3 and Rabaptin5 (Fig. 5 B). Of note, Rabaptin5 and GGA3 protein levels were reduced in ErbB3-depleted cells (Fig. 5 C), which could explain the reduced interaction. Nevertheless, the lower protein levels of GGA3 and Rabaptin5 were not due to their reduced transcription, as determined by RT-qPCR (Fig. 5 D), suggesting reduced protein stability. This is supported by the observation that treatment of ErbB3-depleted MCF10A cells with the proteasome inhibitor MG132 partially restored GGA3 and Rabaptin5 protein levels (Fig. S5 G). It is possible that reduced engagement of GGA3 and Rabaptin5 in a stable complex with ErbB3 underlies its proteasomal degradation.

Deepening the mechanistic analysis, we noticed that ErbB3 has a unique putative GGA3-binding motif (DxxLL) in the activation loop of the pseudokinase domain, which is not found on EGFR or ErbB2 (Fig. 5 E). The DxxLL motif constitutes a consensus binding site for VHS domains found in GGA proteins. To test whether ErbB3 binds GGA3 via this motif, we mutated the two conserved leucines in ErbB3 to alanine (LL866/7AA). GGA3 was overexpressed with or without ErbB3 WT or the LL866/7AA mutant in HEK293T cells followed by immunoprecipitation of ErbB3 and immunoblotting against GGA3 (Fig. 5 F). We found that GGA3 co-immunoprecipitated with WT but not the LL866/7AA mutant ErbB3 (Fig. 5 F), suggesting that ErbB3 interacts with the VHS domain of GGA3 (Fig. 5 E). Thus, we tested the hypothesis that ErbB3 scaffolds the assembly of the Arf6–GGA3–Rabaptin5 complex by ectopically expressing these proteins in HEK293T cells with or without WT or LL866/7AA mutant ErbB3, followed by immunoprecipitation of Arf6 and immunoblotting analysis of associated ErbB3, GGA3, and Rabaptin5. We found

that GGA3 and Rabaptin5 coprecipitated more efficiently with Arf6 in the presence of WT but not mutant (LL866/7AA) ErbB3 (Fig. 5 G). In agreement, we found that the ectopically expressed LL866/7AA mutant form of mCherry-tagged ErbB3 colocalized with fewer internalized Integrin β1–positive structures compared with the WT receptor (Fig. 5 H). In addition to its interaction with GGA3 and consistent with its proposed scaffolding role of ErbB3, we found that ErbB3 can directly bind Rabaptin5 in vitro, as demonstrated by pull-down experiments using recombinant GST-ErbB3 incubated with recombinant MBP–Rabaptin5 (Fig. S5 H). Taken together, our data suggest that ErbB3 acts as a scaffold that associates with both GGA3 and Rabaptin5 to promote the assembly of the Arf6–GGA3–Rabaptin5 endosomal sorting complex. In the absence of ErbB3, GGA3 and Rabaptin5 protein levels are reduced possibly as a consequence of their release from this sorting complex.

### Rabaptin5 and GGA3 are effectors of ErbB3 in endosomal trafficking

Previous studies revealed the critical importance of Rabaptin5 for directing Integrin β3 toward Rab4-dependent recycling (Do et al., 2017; Christoforides et al., 2012); however, the same has not yet been demonstrated for Integrin β1. Thus, we traced surface-labeled Integrin β1 in MCF10A cells transfected with either control or Rabaptin5 siRNA and visualized the re-emergence of internalized Integrin β1 at the basal membrane by TIRF live-cell imaging (Fig. 6 A). Depletion of Rabaptin5 caused a significant reduction in the rate and amount of Integrin β1 that recycled back to the basal membrane (Fig. 6, B–D), quantified as a significant decrease of 28.7% in AUC.

Furthermore, we examined the impact of Rabaptin5 depletion on Integrin β1 stability by pulse-chase analysis of protein turnover. Control or Rabaptin5 siRNA-transfected MCF10A cells were labeled with $^{35}$S-methionine/cysteine for 1 h, followed by chasing with unlabeled amino acids at 37°C. Integrin β1 was subsequently immunoprecipitated, and the incorporated $^{35}$S-label was determined by autoradiography. The data show that Rabaptin5-depleted cells exhibited accelerated Integrin β1 turnover, with a significant 49% reduction in 9 h when compared to control-treated cells (Fig. 6, E and F). Likewise, previous work has shown that siRNA-mediated silencing of GGA3 reroutes Integrin β1 from Rab4-dependent recycling toward lysosomal degradation (Ratcliffe et al., 2016).

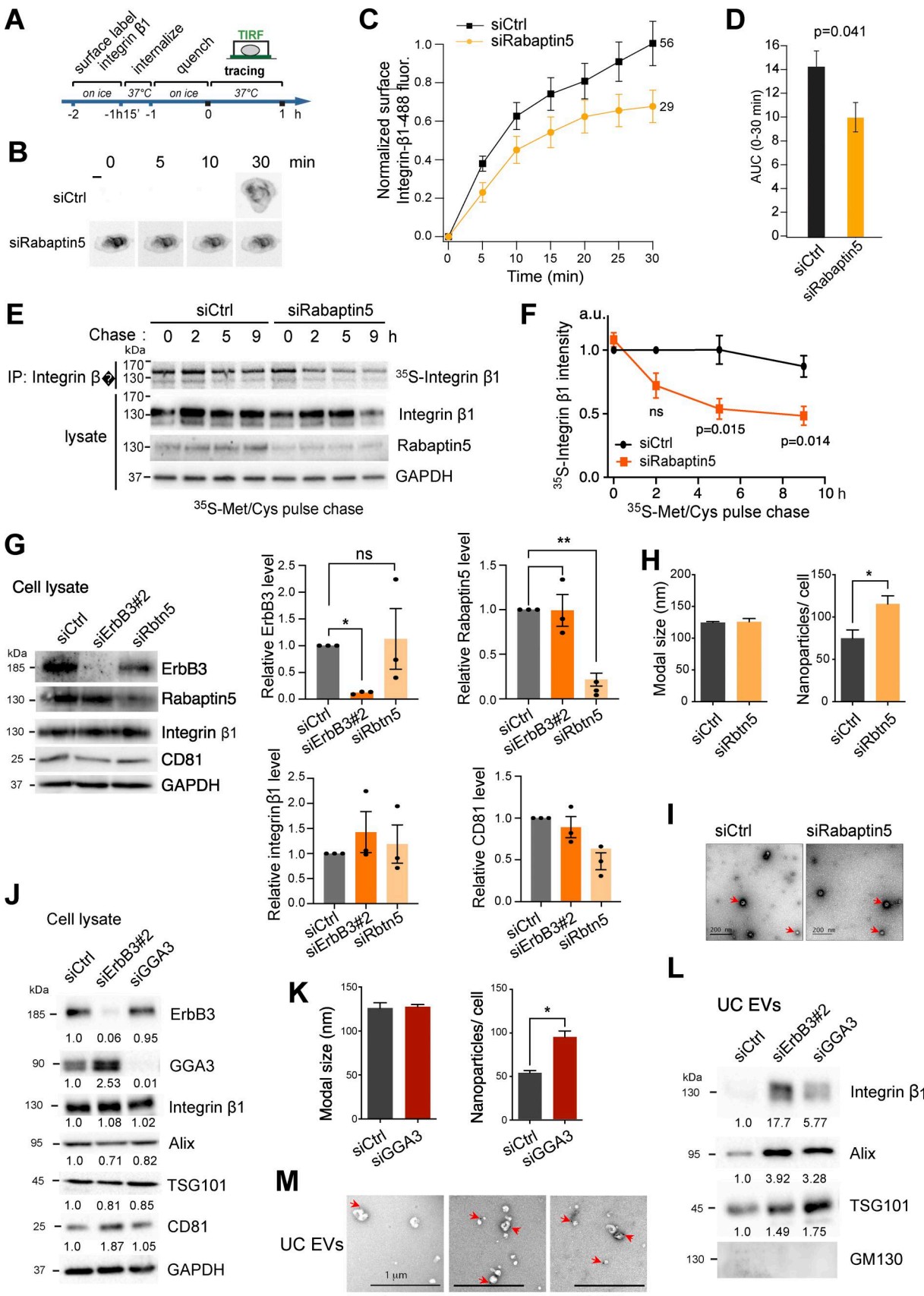

Figure 6. **Rabaptin5 and GGA3 are effectors of ErbB3 in endosomal trafficking. (A and B)** Representative TIRF microscopy images of Integrin β1 from peripheral areas of MCF10A cells transiently transfected with control siRNA (siCtrl) or siRabaptin5. **(C)** Quantification of recycled Integrin β1 was conducted on

the indicated number of cells (right-hand side of graphs) from three independent experiments and is shown as Alexa 488 intensity normalized against levels at the onset of tracing (0) and maximum intensity (1). **(D)** Columns show AUCs. **(E)** Determination of Integrin β1 turnover by pulse-chase metabolic labeling. Control or Rabaptin5 siRNA-transfected MCF10A cells were pulse chase–labeled with radioactive ($^{35}$S) methionine and cysteine. Radiolabeled Integrin β1 was visualized by radiography of Integrin β1 immunoprecipitates (upper panel). Cell lysates and immunoprecipitates were analyzed by immunoblotting as indicated. **(F)** Quantification of radiolabeled Integrin β1 at the indicated times of three independent pulse and chase experiments. Data are presented as mean values ± SEM. P values were determined by two-tailed paired Student's *t* test. ns, nonsignificant. **(G)** Expression of the indicated proteins and GAPDH (as a loading control) in the cell extract of MCF10A cells transiently transfected with siCtrl, siErbB3#2, or siRabaptin5. Densitometric quantification of the indicated protein expression in MCF10A cells is presented as the mean ± SEM from three biological replicates, normalized to the corresponding siCtrl condition. Data are presented as mean values ± SEM. P values were determined by one-way ANOVA, followed by multiple paired comparisons conducted by means of Bonferroni's posttest method. *P ≤ 0.05; **P ≤ 0.01. **(H)** EVs enriched in the VSF released by the indicated MCF10A transiently transfected with control or Rabaptin5 siRNAs were quantified by NTA in terms of nanoparticle modal size (nm) and number after normalization to the total cell number. **(I)** Representative TEM micrographs of EVs enriched in the VSF of control or Rabaptin5-depleted MCF10A cells (EVs are indicated by red arrows). **(J)** Protein expression in the cell extract of MCF10A control samples, ErbB3#2- or GGA3-depleted (left panel) samples, of ErbB3, Integrin β1, and GGA3, besides the indicated EV-specific positive markers (ALIX, TSG101, and CD81) and GAPDH (as a loading control). **(K)** EVs enriched by UC and released by the indicated MCF10A transiently transfected with control or GGA3 siRNAs were quantified by NTA in terms of nanoparticle modal size (nm) and number after normalization to the total cell number. Data in H and K are presented as mean values ± SEM. P values were determined by unpaired Student's *t* test. *P ≤ 0.05. **(L)** Protein expression levels of Integrin β1 and the indicated positive and negative EV markers in the UC EVs isolated from MCF10A cells described in I. Average densitometric values of at least two independent biological replicates were normalized to siRNA control. **(M)** Representative TEM micrographs of EVs enriched by UC secreted from control, and ErbB3- or GGA3-depleted MCF10A cells (EVs are indicated by red arrows). Scale bars, 10 μm (B), 200 nm (I), 1 μm (M).

We next assessed the impact of Rabaptin5 or GGA3 on EV release. Thus, Rabaptin5 knockdown in MCF10A and MCF7 cells (Fig. 6 G and Fig. S5 I), followed by NTA, revealed a significant increase in release of small EVs without impairing the EVs' modal size in comparison with siCtrl cells (Fig. 6 H and Fig. S5 J), while TEM confirmed the presence of membrane-surrounded nanovesicles in the enriched VSF (Fig. 6 I and Fig. S5 K). Moreover, MCF10A cells transfected with siCtrl, siErbB3, or siGGA3 (Fig. 6 J and Fig. S5 L) had their EVs enriched by UC and characterized by NTA, indicating that GGA3 depletion enhanced significantly EV secretion in comparison with the siCtrl cells (Fig. 6 K), similar to ErbB3 depletion (Fig. 3). In line with this finding, further analysis of different positive and negative EV markers by immunoblotting showed enrichment of ALIX, TSG101, and Integrin β1 in the UC EV fraction upon ErbB3 or GGA3 depletion, while the Golgi marker GM130 was absent (Fig. 6 L). Of note, the morphology of these EVs was monitored by TEM (Fig. 6 M). Hence, loss of either GGA3 or Rabaptin5 mimics the effect of loss of ErbB3 on endocytic trafficking of Integrin β1, consistent with the hypothesis that GGA3 and Rabaptin5 are effectors of ErbB3 in promoting endosomal recycling and impeding EV release.

## Discussion

We report that the RTK family protein ErbB3 promotes endocytic recycling of Integrin β1 and TfR in a ligand-independent manner, and that ErbB3 restricts the release of EV-associated Integrin β1 (Fig. 7). Endocytic trafficking of RTKs provides critical spatial and temporal control of the intracellular signaling events they trigger, which ultimately governs cellular response. Hence, the mechanisms by which RTKs internalize and traffic within the cell have been subject to intense scrutiny. While prevailing knowledge centers on RTKs as vesicular passengers, our study provides a compelling example of an RTK family member playing an integral role in the endocytic trafficking machinery per se.

RTK signaling is integrated at many levels with the regulation of endocytic trafficking of integrins (Caswell et al., 2009; Ivaska

and Heino, 2011; Woods et al., 2004). Growth factor–mediated activation of AKT or protein kinase D can regulate integrin recycling by phosphorylating proteins, such as ACAP1, an ARF6-GAP, or Rabaptin5 (Christoforides et al., 2012; Li et al., 2005). Conversely, several studies have shown that integrins alter recycling of other receptors, such as EGFR and vascular endothelial growth factor receptor (VEGFR2), further highlighting the close coordination of integrins with RTK trafficking (Caswell et al., 2008; Reynolds et al., 2009). We found that ErbB3 can promote endocytic recycling of Integrin β1 and TfR, independently of canonical ErbB3-driven tyrosine kinase signaling. Intriguingly, EGFR has been reported to function in a ligand-independent manner to scaffold an exocyst subcomplex needed to assemble autophagosomes (Tan et al., 2015). This invites wider speculation that "inactive" RTKs, as exemplified by EGFR and ErbB3, might play an instrumental role in intracellular vesicular trafficking as scaffolding centers.

We show that ErbB3 interacts with both GGA3 (via its VHS domain) and Rabaptin5 to promote assembly and stability of the Arf6–GGA3–Rabaptin5 endosomal sorting complex. GGA3 has previously been reported to be required to sort multiple cargos, such as Integrin β1, c-Met, and c-Ret for Rab4-dependent recycling (Crupi et al., 2020; Parachoniak et al., 2011; Ratcliffe et al., 2016). It is therefore possible that ErbB3 drives recycling of not only Integrin β1 and TfR as reported here, but also of diverse cargo that depends on Arf6-GGA3, although this remains to be addressed. Furthermore, loss of ErbB3 redirects Integrin β1 toward lysosomes for degradation, mimicking loss of GGA3 that similarly redirects both Integrin β1 (Ratcliffe et al., 2016) and c-Met toward lysosomal degradation (Parachoniak et al., 2011), or Rabaptin5 depletion that we find similarly redirects trafficking of internalized Integrin β1 toward lysosomal degradation. Taken together, these results lead us to propose that ErbB3 engages into endocytic recycling by promoting formation and stability of the Arf6–GGA3–Rabaptin5 vesicular adaptor complex, in line with the emerging concept that pseudokinases have evolved to acquire scaffolding functions.

In addition to this, it has been shown by others that Arf6 interacts with phospholipase D2, to promote synthesis of

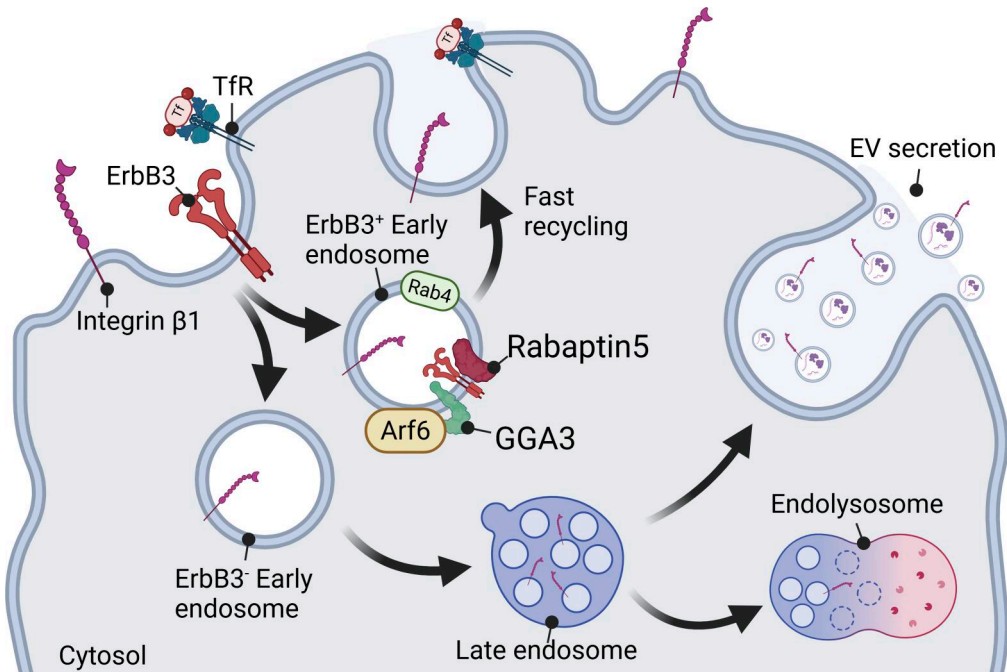

Figure 7. **Schematic model summarizing the effect of ErbB3 in vesicular trafficking.** ErbB3, in a ligand-independent manner, engages with Arf6–GGA3–Rabaptin5 endosomal sorting complex, to promote early recycling of TfR and Integrin β1. Depletion of ErbB3 or its effectors Rabaptin5 or GGA3 reroutes Integrin β1 from endocytic recycling toward lysosomal degradation and secretion as EV cargo. Created in BioRender.

phosphatidic acid and thereby formation of syntenin–ALIX intraluminal vesicles, which are later secreted as exosomes (Ghossoub et al., 2014). Likewise, Arf6 activation leads to the phosphorylation of myosin light chain and actomyosin contraction, which allows the vesicles to bud off from the membranes of cancer cells (Muralidharan-Chari et al., 2009). Thus, since ErbB3 promotes the assembly of the Arf6–GGA3–Rabaptin5 complex, we speculate that ErbB3 association with GGA3 could negatively control EV release by retaining Arf6 at early endosomes, which could suppress the formation of Arf6-PLD2-PA complexes, explaining the reduced EV secretion in breast epithelial cells, although further studies are required to address this hypothesis more rigorously.

Activation of EGFR and ErbB2 involves formation of asymmetric kinase domain dimers, whereby one kinase domain acts as an "activator" and the other as a "receiver" that becomes activated (Jura et al., 2009; Littlefield et al., 2014). However, the ErbB3 pseudokinase domain acts only as an activator, while the receiver interface diverges in sequence from other ErbB receptor family members and is not engaged in kinase domain dimerization (Jura et al., 2009). Interestingly, GGA3 interacts with a DxxLL motif situated on the activation loop within the dysfunctional catalytic site in the ErbB3 pseudokinase domain. This motif is not present in other ErbB receptor family members, where the corresponding position is instead involved in regulating kinase activity. It is tempting to speculate that the dysfunctional catalytic domain in ErbB3 may have evolved a novel kinase-independent function in regulating endocytic trafficking, perhaps shedding some light on the enigmatic nature of pseudokinases.

Previous studies have found that Integrin β1 can physically interact with EGFR and endocytose together (Caswell et al.,

2008; Yu et al., 2000). Although ErbB3 and Integrin β1 were partially colocalized at focal adhesions, cell–cell contacts, and recycling endosomes, we did not find that ErbB3 depletion influenced Integrin β1 internalization. Thus, it remains to be elucidated whether ErbB3 and Integrin β1 associate during endocytosis. Notwithstanding, our results suggest that ErbB3 encounters Integrin β1 in recycling endosomes, presumably at the step when the cargo is sorted into Rab4-positive vesicles through engagement of the Arf6–GGA3–Rabaptin5 complex. This is suggested by the observation that Integrin β1 and ErbB3 colocalization was primarily observed upon treatment with the recycling inhibitor PQ, when ErbB3 is retained in a Rab4-positive compartment, which coincides with enhanced binding of ErbB3 to GGA3 and Rabaptin5. This is in accordance with previous studies showing that the sortilin-related receptor (SorLA) attenuates ErbB3 lysosomal degradation by retaining ErbB3 within Rab4-positive recycling endosomes (Al-Akhrass et al., 2021). Additionally, GGA3 has been found to be enriched in Rab4-positive endosomes alongside endocytosed Met, with significantly lower enrichment observed in Rab5-positive early endosomes (Parachoniak et al., 2011). Interestingly, we found that ErbB3 is required for rapid recycling of Integrin β1 in nonmalignant breast epithelial cells, but failed to demonstrate the same effect in malignant MCF7 cells. It is well established that integrin trafficking is regulated by multiple context-dependent mechanisms (Moreno-Layseca et al., 2019). Thus, it will be interesting to investigate whether integrin endosomal sorting mechanisms change upon malignant transformation.

The proposed role of ErbB3 in endosomal sorting of Integrin β1 and TfR and possibly other cargo into recycling endosomes postulates that ErbB3 frequently resides in the sorting

compartment independent of ligand stimulation. In line with this prediction, ErbB3 has been found to continuously endocytose and recycle back to the plasma membrane (Cao et al., 2007; Fosdahl et al., 2017; Sak et al., 2012) independent of either ligand stimulation or other members of the EGFR family (Sak et al., 2012). Intriguingly, dynamic availability of ErbB3 in vesicular compartments may also be regulated independently of EGFR and ErbB2, since steady-state protein levels of ErbB3, but not EGFR or ErbB2, are tightly controlled by ligand-independent ubiquitination by the E3-ligase Nrdp1 that localizes in perinuclear membranes and targets ErbB3 for proteasomal degradation (Diamonti et al., 2002; Qiu and Goldberg, 2002). Furthermore, we found that ErbB3-dependent cell sheet migration also does not require EGF or other growth factors to signal. It is, however, noteworthy to stress that growth factor–induced EGFR or ErbB2 signaling is well known to promote cell migration, and our data do not rule out or address the possible ErbB3 cooperation with EGFR or ErbB2 to drive growth factor–induced cell motility. In this manuscript, we focus entirely on the unexpected and growth factor–independent roles of ErbB3, distinguishing our investigation from the frequently studied and complex growth factor–induced cell signaling events.

Hence, our data point to ErbB3 regulating Integrin β1 delivery to the leading edge of migrating sheets of epithelial cells, promoting their migration. Since several studies established that integrin recycling promotes cell migration and cancer cell invasiveness (Caswell et al., 2008; Pellinen and Ivaska, 2006), we reveal a new perspective of how ErbB3 could mechanistically contribute to cell dissemination in part by promoting integrin recycling. Conversely, we speculate that loss of ErbB3 could lead to increased dissemination of heterogeneous tumors by increasing EVs release, with the potential to increase invasiveness of surrounding WT or mutant cells. This is consistent with the findings of Luhtala et al. (2018), who identified low ErbB3 expression as a prognostic biomarker for shorter recurrence-free survival in patients with ErbB2-amplified breast cancer. Furthermore, reduced ErbB3 levels were associated with clinically aggressive features, including positive lymph node status, larger tumor size, triple-negative subtype, and basal-like phenotype (Luhtala et al., 2018). Yet, in what context either gain or loss of ErbB3 might be contributing to carcinogenesis remains to be explored. Of note, EVs carry multiple functional molecules and whether ErbB3 affects the sorting of selected protein EV cargo remains to be addressed by proteomics studies. Yet, our study revealed that the pseudokinase ErbB3 plays a role in the vesicular sorting machinery by promoting rapid endosomal recycling of other cargo and impeding EV release. This invites the wider speculation that other pseudo-RTKs may also have evolved scaffolding functions that might play instrumental roles in vesicular trafficking.

## Materials and methods

### Correlation analysis of gene expression
*ErbB3* mRNA expression comparison in BRCA tumor versus normal samples was retrieved from the GEPIA2 server (https://gepia2.cancer-pku.cn/) (Tang et al., 2019). BRCA patients' overall survival was acquired using the KM plotter database (KMplot) (Lánczky and Győrffy, 2021). Correlation of *ErbB3*

mRNA expression and chemotherapy response in BRCA patients was conducted using the ROC Plotter database (https://www.rocplot.org) (Fekete and Győrffy, 2019). BRCA patients were categorized as non- or responders to treatment, according to the relapse-free survival status at 5 years after surgery.

### Cell lines and treatments
MCF10A (RRID:CVCL_0598) cells were cultured in DMEM/F12 (Gibco) supplemented with glutamine (Sigma-Aldrich), 5% horse serum (Gibco), 20 ng/ml EGF (Miltenyi Biotec), 0.5 mg/ml hydrocortisone, 100 ng/ml cholera toxin, and 10 µg/ml insulin (all from Sigma-Aldrich). Where indicated, a starvation medium lacking serum, insulin, and EGF was used. Primary human mammary epithelial cells (prHMEC; RRID:CVCL_0307) from Gibco were cultured in HMEC Ready medium (Invitrogen). MCF7 (RRID:CVCL_0031) cells were cultured in EMEM supplemented with 0.01 mg/ml human recombinant insulin and 10% fetal bovine serum (FBS). HEK293T (RRID:CVCL_0063) cells were cultured in DMEM (Sigma-Aldrich) with 10% FBS (Gibco). Unless otherwise stated, the cell lines were purchased from ATCC. Culture conditions were 5% $CO_2$ at 37°C. Cell line authentication barcodes were purchased from Eurofins to confirm the cells' identity and absence of contamination.

The EGFR/ErbB2 inhibitor lapatinib was used at 1 µM (LC Laboratories) and was batch-tested for efficiency; CHX at 50 µM (from Sigma-Aldrich); chloroquine at 50 µM (#14774; Cell Signaling); PQ at 0.5 mM (#160393; Sigma-Aldrich) for 10 or 30 min; MG132 (Calbiochem) at 50 µM. In all cases, control cells were treated with vehicle alone (DMSO, from Sigma-Aldrich) at the corresponding concentration.

### Transfections
Cells were transfected with RNAiMAX (Invitrogen) according to the supplier's instructions. A final concentration of 20 nM of the following siRNAs was used: siErbB3#1 (5′-CAAUACCAGACACUGUACAAGCUCU5-3′); siErbB3#2 (5′-UCGUCAUGUUGAACUAUAA-3′); siRab4A (5′-GAACGAUUCAGGUCCGUGATT-3′); siRabaptin5 (5′-CCGGGCAAUUCUGAAUGAUACUAAA-3′); siGGA3 (5′-CCCGGGUCAUCAACUCUUATT-3′); or scramble siRNA (siCtrl: nontargeting predesigned ID 12935112). All siRNAs were purchased from Invitrogen, and the silencing experiments were performed 48 h after transfection.

### cDNAs and DNA transfection
MCF10A cells were transfected with Lipofectamine 3000 (#L3000075; Thermo Fisher Scientific) according to the supplier's instructions. Constructs for Rab4, Rab11, and Rabaptin5 were kindly provided from Marino Zerial (MPG Dresden), and mCherry- and Citrine-tagged ErbB3 were kindly provided by Martin Offterdinger (Innsbruck, Austria). Rabaptin5 was subcloned into pMALC2 (#75286; Addgene) for bacterial expression (see below). mCherry-tagged Rabaptin5 and YFP-tagged Arf6 were purchased from Addgene. Flag-ErbB3 cDNA was subjected to site-directed mutagenesis (Agilent) to generate LL866/7AA mutant Flag-ErbB3. For rescue experiments, mCherry-ErbB3 carrying two nonsense point mutations in the siRNA-recognition sequence was subcloned into pAdEasy XL

and adenovirus produced using the AdEasy XL adenovirus system (#240010; Agilent) according to the manufacturer's instructions, and MCF10A cells infected at a titter that yielded ErbB3 overexpression in ~90% of the cells.

## RT-qPCR
RNA was extracted with the RNeasy Mini Kit (#74104; Qiagen). cDNA was synthesized with the iScript cDNA Synthesis Kit (#170-8891; Bio-Rad). qPCR was performed using the KAPA SYBR Fast and primers for *ERBB3* (5′-CAACTCCAGATGAAGACT ATG-3′ and 5′-TGCTATGCCAGTAATCAGG-3′), *ITGB1* (5′-AGA TCCGAAGTTTCAAGGGC-3′ and 5′-GAACCATGACCTCGTTGT TC-3′), *GGA3* (5′GGGACAGGGTGTCTGAGAAAG-3′ and 5′-GTG CCTCGTCTTCCTTCACC-3′), *RABEP1* (5′-TTCCCAGCCTGACGT TTCTC-3′ and 5′-GCTGCTGTTGTGCACGTAAA-3′), and *GAPDH* (5′-CCCTTCATTGACCTCAACTA-3′ and 5′-CCAAAGTTGTCATG GATGAC-3′). The comparative $C_t$ method was used to calculate fold change in gene expression relative to *GAPDH*.

## Antibodies
The following primary antibodies were used: anti-ErbB3 (#05-390, clone 2F12; Millipore—for IP; and #12708S, clone D22C5; Cell Signaling Technology—for immunoblotting); anti-Integrin β1 monoclonal (#ab52971; Abcam—for immunoblotting; and #MAB1987Z; clone P4C10; Sigma-Aldrich—for blocking activity). For immunoblotting, we used the antibodies: anti-phospho-ErbB3 Tyr1289 (#4791; Cell Signaling Technology); anti-EGFR (#2232; Cell Signaling Technology); anti-ErbB2 (#06-562; Millipore); anti-phospho-AKT Thr308 (#2965; Cell Signaling Technology); anti-AKT (#9272; Cell Signaling Technology); anti-phospho ERK1/2 (#9101; Cell Signaling Technology); anti-ERK1/2 (#9102; Cell Signaling Technology); anti-phosphotyrosine (#05-1050-M; 4G10 Platinum; Millipore); anti-E-cadherin (#610182; BD Transduction Laboratories); anti-Muc1 (#4538; Cell Signaling Technology); anti-ALIX (#sc-53540, clone 1A12; Santa Cruz Biotechnology); anti-CD81 (#sc-166029, clone B11; Santa Cruz Biotechnology); anti-CD63 (#sc-5275, MX-49.129.5; Santa Cruz Biotechnology); anti-GM130 (#610822, clone 35; BD Biosciences); anti-Rab4A (#610888; BD Biosciences); anti-EDH1 (#ab109747; Abcam); anti-Rabaptin5 (#sc-271069; Santa Cruz Biotechnology); anti-GGA3 (#612311, clone 8; BD Transduction Laboratories); anti-Arf6 (#sc-7971, clone 3A-1; Santa Cruz Biotechnology); anti-GAPDH (##2118, clone 14C10; Cell Signaling Technology); anti-α-Tubulin (#T5168; Sigma-Aldrich); anti-β-actin (#sc-69879; Santa Cruz Biotechnology). HRP-conjugated goat anti-mouse (#62-6520), goat anti-rabbit (#65-6120), and rabbit anti-sheep (#31480) secondary antibodies were purchased from Thermo Fisher Scientific. Alexa Fluor 594 (#A-11037)– or Alexa Fluor 488 (#A-11008)–conjugated anti-rabbit antibodies (Thermo Fisher Scientific) were used for immunofluorescence. For the Integrin β1 recycling assay, we used the Alexa Fluor 488 anti-human Integrin β1 antibody (#303016, clone TS2/16; BioLegend).

## Immunoblotting and immunoprecipitation
Cells were lysed in sodium dodecyl sulfate (SDS) sample buffer (120 mM Tris-HCl, pH 6.8, 3 % SDS, 15 % glycerol, 0.03 %

bromophenol blue, 75 mM dithiothreitol (DTT)) and run on 8–10% polyacrylamide gels. Proteins were transferred onto Hybond-P polyvinylidene fluoride membranes at 100 V for 120 min at 4°C. Membranes were blocked in 4% milk or 5% bovine serum albumin (BSA), dissolved in TTBS (Tris-buffered saline, 0.1% Tween-20). The protein bands were visualized using ECL or ECL plus reagents and Hyperfilm ECL (all from GE Healthcare) and detected with a CCD camera (Bio-Rad). Quantification of western blots was performed using ImageJ software.

For ErbB3 immunoprecipitation, cells were lysed in 1% NP-40, 50 mM Tris–HCl, pH 8.0, 150 mM NaCl, 0.5% sodium deoxycholate, 0.1% SDS, and 1 mM EDTA supplemented with Halt phosphatase/protease inhibitor (Thermo Fisher Scientific). Preclearing of lysates was performed by incubating samples for 1 h at 4°C under rotation with Pierce control agarose resin. Antibody incubation was performed overnight at 4°C under rotation, and precipitates were washed four times in lysis buffers. Precipitated proteins were resolved by immunoblotting. Arf6 immunoprecipitations were performed as above but with lysis buffer: 1% Triton X-100, 100 mM sodium chloride (NaCl), 50 mM Tris-HCl, pH 7.4. IgG was used as a negative control for immunoprecipitations. Immunoisolation/detection kits coupled to CD63 or CD81 antibodies were purchased from Thermo Fisher Scientific (#10606D and #10616D) and used for EV enrichment. The 100K pellets were resuspended in 100 µl PBS with 0.1% BSA (filtered through a 0.2-µm filter) and mixed with 100 µl of magnetic bead slurry conjugated to the corresponding antibody according to the manufacturer's instructions.

## Immunofluorescence
MCF10A or MCF7 cells were cultured on type I bovine collagen (Advanced BioMatrix)-coated coverslips and fixed with 4% PFA in PBS for 20 min, permeabilized with 0.5% Triton in PBS for 10 min, and then blocked in SuperBlock T20 blocking buffer (Thermo Fisher Scientific). Primary and secondary antibodies and, where indicated, Alexa Fluor 488–conjugated Phalloidin (Thermo Fisher Scientific) were diluted in blocking buffer. Coverslips were mounted using ProLong Antifade Reagent (Thermo Fisher Scientific) with or without 4′,6-diamidino-2-phenylindole (DAPI). Images were taken with Leica SP8 (confocal) or Zeiss Axio Imager M2 (epifluorescence) microscopes and processed using Leica Suite or Zeiss ZEN 2011 software, respectively. Images were processed in 8-bit format. Deconvolution of confocal image stacks was carried out in Huygens Essential 19.04 (Scientific Volume Imaging B.V.) using a theoretical PSF, automatic background estimation, and CMLE deconvolution algorithm. The final signal-to-noise ratio was set to 10.

## Transferrin recycling assay
MCF7 cells were incubated in starvation media supplemented with 25 mM HEPES for 3 h prior to the assay. Cells were treated with 5 µg/ml of Alexa Fluor–conjugated transferrin (T13343; Thermo Fisher Scientific) for 30 min on ice. Thereafter, excess of labeled transferrin was removed by three washes with PBS and subsequently incubated in media containing 50 µg/ml unlabeled holo-transferrin (T4132; Sigma-Aldrich) for the indicated times (0, 10, 30, 60 min). Finally, the cells were treated with an acidic

solution (0.2 M acetic acid and 0.5 M NaCl) for 30 s to remove surface labeling, fixed in 4% PFA in PBS for 20 min at room temperature, and mounted. Quantification was performed on three independent experiments. Mean cell intensity was measured with ImageJ ($n > 17$ images of 1–4 cells for each data point). MCF7 cells were chosen for this assay due to efficient surface labeling with Alexa Fluor–conjugated transferrin. Attempts to perform the transferrin recycling assay in MCF10A cells were unsuccessful, due to insufficient surface labeling with Alexa Fluor–conjugated transferrin in these cells.

## Integrin recycling assay

MCF10A cells were plated on collagen-coated 24-well plates either sparsely or at high confluency for subsequent scratching. Cells were growth factor–deprived for 3 h and then put on ice for 5 min to cool down. Surface Integrin β1 was labeled at 4°C during 45 min, then washed twice with PBS, and returned to 37°C in order to allow internalization, as previously described (Arjonen et al., 2012). After 15 min, cells were put back on ice and treated for 1 h with anti-Alexa Fluor 488 (#A11094; Thermo Fisher Scientific) in order to quench the remaining surface signal. After quenching, cells were again washed twice with PBS and incubated in deprivation media at 37°C. At indicated times, cells were fixed with PFA 4% in PBS for 20 min at room temperature. When only integrin internalization was monitored, the quenching step was not performed. Quantification of total pixel intensity per cell, mean pixel intensity, and average projections was performed with ImageJ.

## TIRF microscopy

A prism-type TIRF microscope built around a Nikon E600FN upright microscope equipped with a 16× 0.8-NA water-immersion objective was used. The output from diode-pumped solid-state lasers (Cobolt) was merged by dichroic mirrors and homogenized and expanded by a rotating, light-shaping diffuser (Physical Optics Corp.). Excitation light was selected by interference filters (491 nm for Alexa 488 and 561 nm for mRFP; Chroma) mounted in a filter wheel (Lambda 10-3, Sutter Instruments) and refocused through a modified dove prism (Axicon) at a 70° angle to achieve total internal reflection. Cells grown on 25-mm glass coverslips (Menzel-Gläser, #1) were kept on ice and mounted in a modified Sykes–Moore perfusion chamber just before imaging, placed on top of the prism, and perfused with DMEM/F12 at a rate of 0.1 ml/min. Emission wavelengths were selected with interference filters (525/25 nm for Alexa 488, 590/20 for mRFP; Chroma) mounted in a filter wheel (Sutter Instruments), and fluorescence was detected by a back-illuminated EM-CCD camera (DU-887; Andor Technology). Filter wheels and camera were controlled by MetaFluor (Molecular Devices Corp.), and images were acquired every 10 s. TIRF penetration depth was 83 nm, and an electronic shutter prevented light exposure between image captures. Imaging was done at 37°C.

TIRF microscopy 8-bit images were analyzed offline using ImageJ. Briefly, cell footprints were manually identified and regions of interest covering the edges of the adherent cells were drawn. Intensity changes within these regions through the time

course of the experiment (45 min) were measured and exported to Excel. All data points were background-corrected, followed by normalization to the prestimulatory level (F/F0). Cells within large cell clusters were excluded from the analysis.

## Wound healing assay

MCF10A cells were seeded in Falcon Multiwell 48-well plates and grown to full confluency. Before performing the wound, cells were pretreated with 1 µM lapatinib or DMSO alone in serum-containing media lacking added growth factors for 1 h. Thereafter, a scratch was placed in the middle of each well with a sterile pipette tip. After two washes with PBS, lapatinib/DMSO-containing growth factor–deprived but serum-containing media were added back to the cells and the wound healing process was monitored using an IncuCyte ZOOM 40008 microscope for 14 h. The obtained images were analyzed using FIJI software.

## $^{35}$S-Met/Cys pulse-chase protein turnover assay

MCF10A cells transfected with control or ErbB3 siRNA were incubated with growth factor–deprived DMEM lacking methionine and cysteine supplemented with 10 µCi/ml $^{35}$S-methionine/cysteine (PerkinElmer) for 30 min at 37°C. The labeling media were replaced with growth factor–deprived MCF10A culture media containing 5 mM L-cysteine and 5 mM L-methionine (Sigma-Aldrich) for the indicated times, prior to cell lysis and immunoprecipitation of Integrin β1, SDS-polyacrylamide gel electrophoresis, and detection of incorporated radioactivity using a phosphorimager. Images were scanned, and quantification of three independent experiments was performed in ImageJ.

## Recombinant protein purification and in vitro binding assay

pMalC2–Rabaptin5 was transformed into the BL21S(DE3)pLysS bacterial strain (Invitrogen). Protein expression was induced with 0.3 mM IPTG for 2 h at 37°C, and recombinant MBP–Rabaptin5 was purified with an amylose resin (New England Biolabs) according to the manufacturer's instructions and dialyzed against lysis buffer (20 mM Tris-HCl, pH 7.4, 50 mM NaCl, 0.05 mM PMSF, 6 KIU7/ml aprotinin, 0.5 mM DTT, 5% glycerol, and 0.5% Triton X-100). Recombinant 6His-ErbB3 was purchased from Sigma-Aldrich. The in vitro binding assay was conducted by incubating 1 µg/ml recombinant ErbB3 with 1 µg/ml recombinant Rabaptin5 in a stringent buffer containing 50 mM Tris, pH 7.5, 150 mM NaCl, 1% NP-40, 0.5% sodium deoxycholate, 0.1% SDS, 1 mM EDTA, and Halt protease and phosphatase inhibitor overnight at 4°C. ErbB3 was immunoprecipitated and bound Rabaptin5 detected by immunoblotting.

## Colocalization analysis

Analysis was conducted on confocal fluorescence images and analyzed using ImageJ. Rab4/11 and ErbB3 colocalization: single Rab4- or Rab11-positive structures were identified and 40 × 40-pixel squares were drawn with the structures in the center and saved as separate images. For each cell, a minimum of 15 structures were randomly selected and the intensity of the center (7-pixel diameter circle, a) and the adjacent volume (background, b) for each structure were determined using ImageJ. These

regions were subsequently transferred to the other channel (ErbB3), and similar measurements were performed. The relative enrichment of ErbB3 at the Rab-positive structures was determined by the formula (a–b)/b. An average for each cell was determined, and a minimum of 16 cells was analyzed for each condition. Images shown in the paper are average projections of all structures from all cells analyzed. ErbB3 and Integrin β1 colocalization: deconvoluted confocal image stacks were processed into maximum intensity projections using Fiji/ImageJ. Signal intensities along manually drawn lines were obtained using the histogram tool. Colocalization was analyzed in ImageJ as enrichment of Integrin β1 in ErbB3-positive structures (as described for ErbB3 and Rab colocalization above).

### VSV-G trafficking assay

VSV-G ts045-GFP was expressed in MCF10A cells for 24 h. Cells were incubated at 40°C for 6 h, which causes misfolding and trapping of VSV-G in the ER. The cells were then shifted to permissive temperature (32°C) and either PFA-fixed at indicated times for immunofluorescence imaging or subjected to incubation on ice with Sulfo-NHS-SS-Biotin (Thermo Fisher Scientific) to biotinylate all surface proteins. Biotinylated cells were lysed in 1% Triton X-100 containing lysis buffer followed by pull-down with Streptavidin-agarose beads (Sigma-Aldrich) overnight and western blot analysis with indicated antibodies.

### EdU incorporation assay

EdU incorporation into DNA of MCF10A monolayers subjected to a scratch assay in the presence of 1 µM lapatinib or vehicle alone was detected using the Click-iT EdU Alexa Fluor 488 Imaging kit (Invitrogen) following the manufacturer's instructions. Cells were later fixed, double-stained with DAPI, and mounted for microscopic imaging with Zeiss Axio Imager M2. Quantification of EdU-positive cells at the indicated times in control or ErbB3-silenced cells was performed with ImageJ.

### EV isolation and characterization

EV isolation and characterization was performed as previously described (Rodrigues-Junior et al., 2019; Théry et al., 2006). The cells were incubated for 48 h in DMEM/F12 supplemented without horse serum as described above. The serum-free conditioned medium was centrifuged at 1,200 × g for 5 min to clear cell debris while measuring the corresponding number of cells. The medium supernatant was filtered through a 0.2-µm filter, was further concentrated 20× by tangential flow filtration on a 50-kDa Ultra-15 Centrifugal Filter (Amicon; Merck/Millipore) by centrifugation at 1,200 × g for 30 min, and defined as the VSF. EVs smaller than 200 nm were subfractionated based on the presence of the tetraspanin CD81 protein. VSF (150 µl) was incubated with 40 µl of anti-CD81 magnetic Dynabeads (#10616D; Thermo Fisher Scientific) in 100 µl of 0.2 µm filtered PBS, respectively, for 60 min at 37°C. The magnetic beads containing CD81-EVs were immobilized and washed with PBS, and the isolated EVs were stored at –20°C for up to 30 days. Additionally, conditioned media derived from MCF10A cells were first centrifuged at 1,200 × g to remove cell debris, and subsequently filtered through a 0.2-µm filter (Sarstedt) and transferred to a polypropylene ultracentrifuge tube (Beckman Coulter). UC was carried out for 2 h at 100,000 × g and 4°C in a swinging bucket rotor (SW28; Beckman Coulter).

Nanoparticle number and size distribution were measured through NTA (NanoSight System; Malvern Panalytical, equipped with a 532-nm laser and NTA 3.4 analytic software). Three videos of 60 s each were recorded of each sample, using the following settings: camera level 8 and detection threshold 10. The system was calibrated with polystyrene beads of 100 nm (Malvern). Modal diameter and per cell ratio of nanoparticles were analyzed statistically.

TEM preparation was performed on freshly purified VSF or UC enriched EVs. The EVs were mixed with an equal volume of 4% wt/vol formaldehyde and further incubated with uranyl oxalate, pH 7.0, for 5 min, then stained in a drop containing 4% wt/vol uranyl acetate and 2% wt/vol methylcellulose on ice, and protected from light for 10 min, as described previously (Rodrigues-Junior et al., 2019). Excess of UA/MC was removed by blotting on filter paper.

For immunogold staining, after the first 3 washes on PBS the protocol follows as indicated: sample was transferred to drops of 50 mM glycine in PBS for 3 × 1 min. Then, the sample was transferred to a drop of blocking solution, 1% BSA in PBS, for 20 min. Primary and secondary incubations were carried out in 0.1% BSA in PBS for 40 min and washed away in 0.1% BSA in PBS for 5 × 1 min. The sample was then processed for TEM contrasting as described above. Dried grids were examined by TEM (FEI Tecnai G2) operated at 80 kV with an ORIUS SC200 CCD camera and Gatan DigitalMicrograph software (Gatan, Inc/BlueScientific).

### Cell viability assay

Cell viability was quantified using PrestoBlue Cell Viability Reagent (Thermo Fisher Scientific) following the manufacturer's instructions. Briefly, $3 \times 10^3$ MCF10A cells were seeded into 96-well plates and incubated at 37°C for 24 h. Then, $1 \times 10^9$ nanoparticles/ml of VSF enriched from cells transfected with siCtrl or siErbB3 were added to each well. Upon 48 h of treatment, 10 µl of PrestoBlue reagent was added to each well and fluorescence (540-nm excitation/590-nm emissions) was measured. The fluorescence units from VSF-treated cells were normalized relative to those of vehicle-treated (EV-free media) cells, and the graphs show average values of % viability with standard error of the mean (SEM) of at least three biological experiments.

### Matrigel invasion assay

Transwell inserts (#351152) for 24-well plates (6.5 mm diameter, 8 µm pore) were coated with 300 µg/ml Matrigel matrix (#734-0269; Corning) diluted in coating buffer (10 mM Tris, pH 8.0, 0.7% wt/vol NaCl) and incubated at 37°C for 1 h. MCF10A cells ($5 \times 10^4$) treated for 48 h with VSF ($1 \times 10^9$ nanoparticles/ml) enriched from cells transfected with siCtrl or siErbB3, or with EV-free media (as control), in the absence or presence of IgG or Integrin β1 blocking antibody P4C10 (5 µg/ml), were seeded in DMEM/F12 in the upper chamber, while DMEM/F12 complete media were placed in the lower chamber and incubated at 37°C to induce the invasion through the Matrigel barrier for 16 h. After

incubation, the inserts were fixed in methanol and stained with DAPI (Sigma-Aldrich). Nuclei were counted in at least five pictures per insert, taken with the 20× objective, using ImageJ. Data were expressed as the percentage of invasion based on the ratio of the mean number of cells invading through Matrigel matrix per mean number of cells in the uncoated permeable support.

### Protein–protein interaction network analysis

A protein–protein interaction network for proteins interacting with GGA3 was constructed using the Search Tool for the Retrieval of Interacting Genes database (STRING-version 12.0; https://string-db.org/), with the required high-confidence score (>0.7).

### Statistical analyses

All data, unless otherwise pointed out, are presented as the mean ± SEM, from the indicated independent biological experiments performed in triplicate or quintuplicate technical repeats. Comparisons were performed using two-tailed paired Student's $t$ test or one-way ANOVA, followed by multiple paired comparisons conducted by means of Bonferroni's posttest method when applicable. Additional statistical methods are explained in the respective figure legends. The data were analyzed with GraphPad Prism 10.1 (GraphPad Software). A P value <0.05 was necessary to determine statistically significant differences.

### Online supplemental material

Fig. S1 shows ErbB3 promotes Integrin β1 endocytic recycling. Fig. S2 shows ErbB3 promotes endocytic recycling of Integrin β1 in the absence of EGFR or ErbB2. Fig. S3 shows ErbB3 gene expression correlates with BRCA patient survival and chemotherapy response. Fig. S4 shows ErbB3 does not affect exocytic trafficking of VSV-G but reduces EV release. Fig. S5 shows ErbB3 enhances Integrin β1 stability and binds to GGA3 and Rabaptin5 in vitro.

### Data availability

Further information and requests for resources and reagents should be directed to and will be fulfilled by the Lead Contacts, Ingvar Ferby (ingvar.ferby@igp.uu.se) and Dorival Mendes Rodrigues-Junior (dorival.mrj@imbim.uu.se). The plasmids generated in this study are available upon request.

## Acknowledgments

We thank Marino Zerial, Lukas Huber, and Martin Offterdinger for kindly providing cDNA constructs. We also thank Carl-Henrik Heldin for valuable discussion and for critically reading the manuscript.

This research was supported by the Swedish Cancer Society, grants CAN2018/469, CAN2021/1506Pj01H, and CAN2024/24/3580Pj01H (Aristidis Moustakas); the Swedish Research Council, grants 2018-02757 and 2023-02865 (Aristidis Moustakas); Lars Hierta Memorial Foundation, grant FO2024-0173 (Dorival Mendes Rodrigues-Junior); O.E. och Edla Johanssons Vetenskapliga Stiftelsen (Dorival Mendes Rodrigues-Junior); Stiftelsen Längmanska kulturfonden, grant BA25-1369 (Dorival Mendes Rodrigues-Junior); and The Sigurd and Elsa Golje Memorial Foundation, grant LA2025-0172 (Dorival Mendes Rodrigues-Junior).

Author contributions: Dorival Mendes Rodrigues-Junior: conceptualization, formal analysis, funding acquisition, investigation, methodology, resources, supervision, validation, visualization, and writing—original draft, review, and editing. Ana Rosa Sáez-Ibáñez: conceptualization, data curation, formal analysis, investigation, and visualization. Takeshi Terabayashi: investigation. Nina Daubel: formal analysis and investigation. Taija Mäkinen: resources and writing—review and editing. Olof Idevall-Hagren: formal analysis, funding acquisition, investigation, resources, visualization, and writing—review and editing. Aristidis Moustakas: funding acquisition, investigation, project administration, resources, supervision, and writing—review and editing. Ingvar Ferby: conceptualization, data curation, formal analysis, funding acquisition, investigation, methodology, project administration, resources, supervision, validation, visualization, and writing—original draft, review, and editing.

Disclosures: All authors have completed and submitted the ICMJE Form for Disclosure of Potential Conflicts of Interest. A.R. Sáez-Ibáñez reported personal fees from Genmab A/S outside the submitted work. No other disclosures were reported.

Submitted: 30 January 2025

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

# Supplemental material

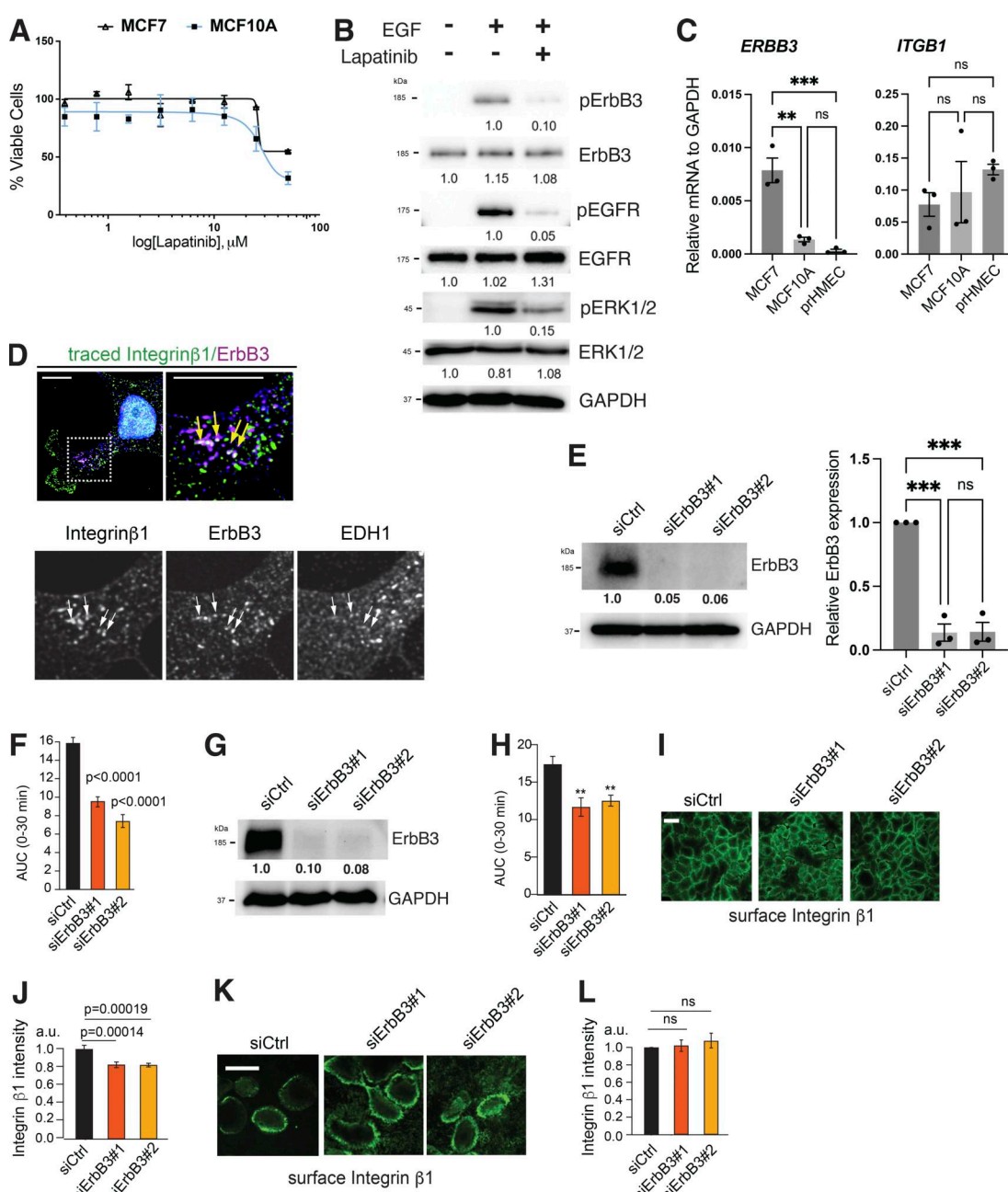

Figure S1. **ErbB3 promotes Integrin β1 endocytic recycling. (A)** Viability curves of MCF7 and MCF10A cells in the presence of increasing concentrations of lapatinib (logarithmic scale) assessed by PrestoBlue fluorescence. Note the lack of significant differences. **(B)** Protein expression levels of the indicated signaling proteins and GAPDH (as a loading control) in MCF10A cells stimulated with 20 ng/ml EGF in the absence or presence of 1 μM lapatinib for 30 min. Average densitometric values of at least two independent biological replicates were normalized to the respective control condition (EGF-negative; lapatinib-positive). **(C)** RT-qPCR analysis of the indicated mRNA levels in MCF7, MCF10A, and prHMEC cells. Values represent fold change of *ErbB3* (left) and *Integrin β1* (*ITGB1*; right) mRNA expression relative to *GAPDH*. Data are presented as mean values of three biological replicates ± SEM, each in technical duplicates. **(D)** Analysis of colocalization of endogenous traced Integrin β1, ErbB3, and the recycling endosomal marker EDH1: MCF7 cells were labeled on ice with an Alexa 488–conjugated anti-Integrin β1 antibody (green) prior to incubation for 30 min at 37°C to allow Integrin β1 internalization, and subsequent immunolabeling of ErbB3 (magenta), counterstained for EDH1 (white, right panel). Note that the squared region in the left panel represents the magnified images of D. **(E and G)** Protein expression of ErbB3 and GAPDH (as a loading control) in the cell extract of MCF10A (E) and prHMEC (G) cells transiently transfected with control siRNA (siCtrl) or two independent siRNAs targeting ErbB3 (#1 and #2). Densitometric values of ErbB3 protein expression in three biological replicates ± SEM of MCF10A cells are shown normalized to the respective siCtrl. **(F and H)** Columns show the AUC of the Integrin β1 recycling data in MCF10A (F: related to Fig. 2 G) and prHMEC cells (H: related to Fig. 2 I). Data are presented as mean values ± SEM; *n* values are indicated in the main figures. **(I–L)** Immunofluorescence imaging of the surface pool of Integrin β1 labeled with Alexa 488–conjugated antibody on ice for 1 h, on MCF10A (I and J) or prHMEC cells (K and L) transfected with siCtrl or ErbB3-targeting siRNAs. The columns in J and L show quantification of Integrin β1 fluorescence intensity, normalized against control siRNA-treated samples. Data are presented as the mean ± SEM, from six independent experiments. P values in C, E, F, H, J, and L were determined by one-way ANOVA followed by multiple paired comparisons conducted by Bonferroni's posttest method. *P ≤ 0.05; **P ≤ 0.01; ***P ≤ 0.001; ****P ≤ 0.0001; ns, nonsignificant. Scale bars, 10 μm (D), 20 μm (I and K).

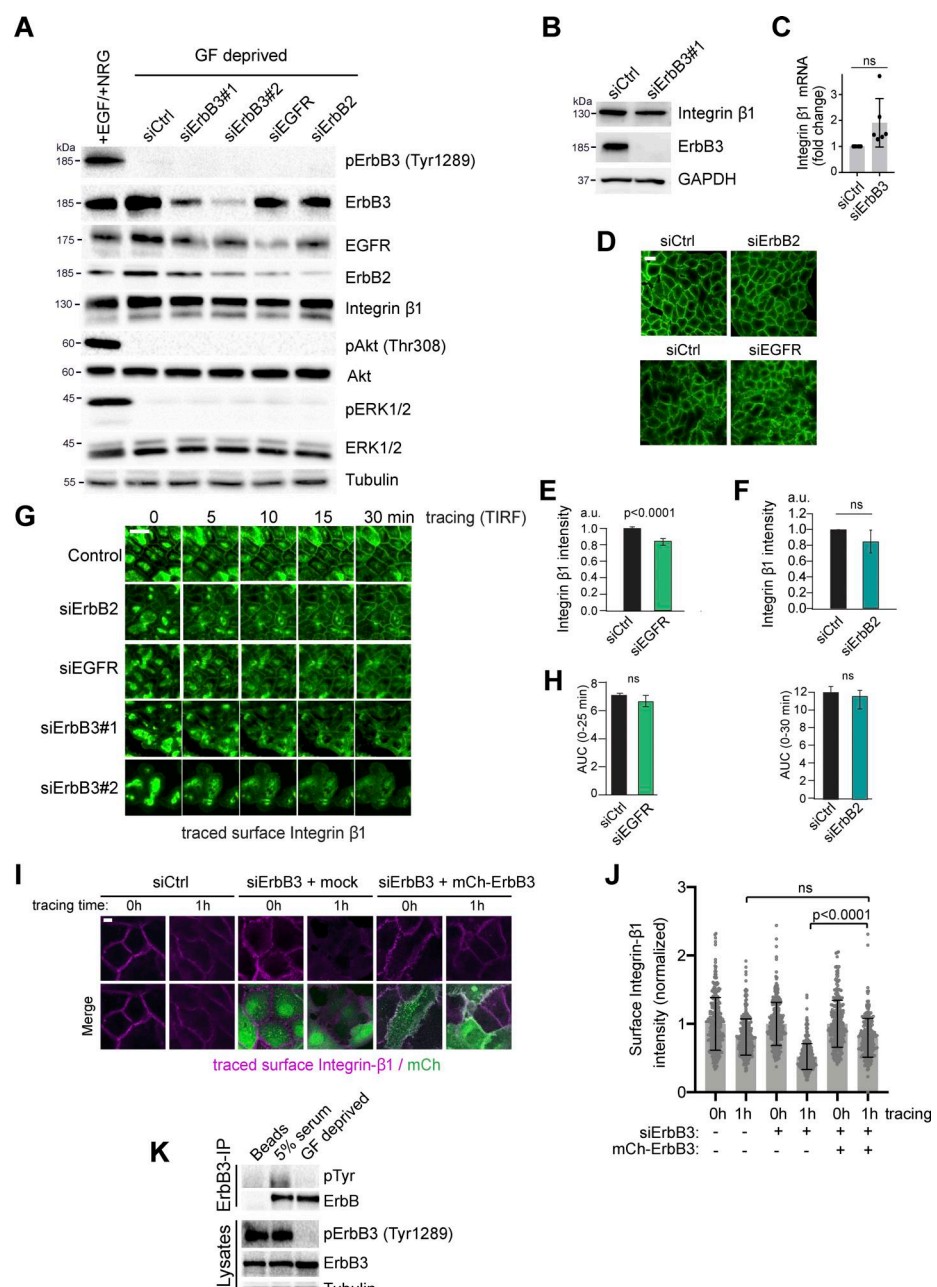

**Figure S2.** **ErbB3 promotes endocytic recycling of Integrin β1 in the absence of EGFR or ErbB2. (A)** Immunoblotting analysis of the indicated proteins and α-Tubulin (as a loading control) in the total MCF10A cell lysates under GF-deprived culture conditions or upon supplementation of the EGF and βNRG, after depletion of ErbB3, EGFR, or ErbB2. **(B)** Immunoblotting analysis of ErbB3, Integrin β1, and GAPDH (as a loading control) upon transfection of prHMEC cells with control or ErbB3 siRNA#1. **(C)** Relative levels of *Integrin β1* (*ITGB1*) mRNA in MCF10A control or ErbB3 siRNA#1-transfected cells as determined by RT-qPCR (n = 6 independent experiments). Data presented in C are shown as the mean ± SEM. The P value was determined by the Mann–Whitney U test. **(D)** Immuno-fluorescence imaging of the surface pool of Integrin β1 labeled with Alexa 488–conjugated antibody on ice for 1 h, on MCF10A cells transfected with control siRNA (siCtrl) or siRNAs targeting EGFR or ErbB2. **(E and F)** Quantification of Integrin β1 fluorescence intensity upon silencing of EGFR (E) or ErbB2 (F) and normalized against the respective control siRNA-treated samples. Data are presented as mean values ± SEM. **(G)** Live-cell TIRF imaging of MCF10A cells to monitor recycling of labeled Integrin β1, after prior siRNA-mediated depletion of ErbB2, EGFR, or ErbB3 (#1 and #2). **(H)** Related to Fig. 2, J and K: shows AUC of the Integrin β1 recycling data. Data are presented as mean values ± SEM. n values are indicated in main figures. **(I and J)** Expression of siRNA-resistant ErbB3 restores the surface pool of traced Integrin β1 in ErbB3-depleted MCF10A cells. **(I)** Confocal imaging of Integrin β1 labeled on the cell surface with an Alexa 488–conjugated antibody, prior to (0 h) or after tracing (1 h), on cells expressing siRNA-resistant mCh-ErbB3 or fluorophore alone with or without transfection of control (siCtrl) or ErbB3 siRNA as indicated. **(J)** Fluorescence intensity (Integrin β1) along cell–cell borders was quantified from three independent experiments. Values were normalized against intensities prior to tracing (0 h). The results suggest that off-target effects of the ErbB3 siRNAs do not underlie the observed recycling defect. Data are presented as mean values ± SD; n > 230 cell–cell borders from 3 independent experiments, and P values were determined by two-tailed paired Student's *t* test. **(K)** Immunoblotting analysis of ErbB3 immunoprecipitates or total cell lysates of MCF10A cells cultured in the presence of 5% horse serum or under GF-deprived culture conditions. Antibodies detecting total phosphotyrosine (4G10; pTyr) or the specific phosphotyrosine 1289 on ErbB3 were used, and α-Tubulin was used as a loading control for the total cell lysates. Scale bars, 20 μm (D and G), 10 μm (I). GF, growth factor.

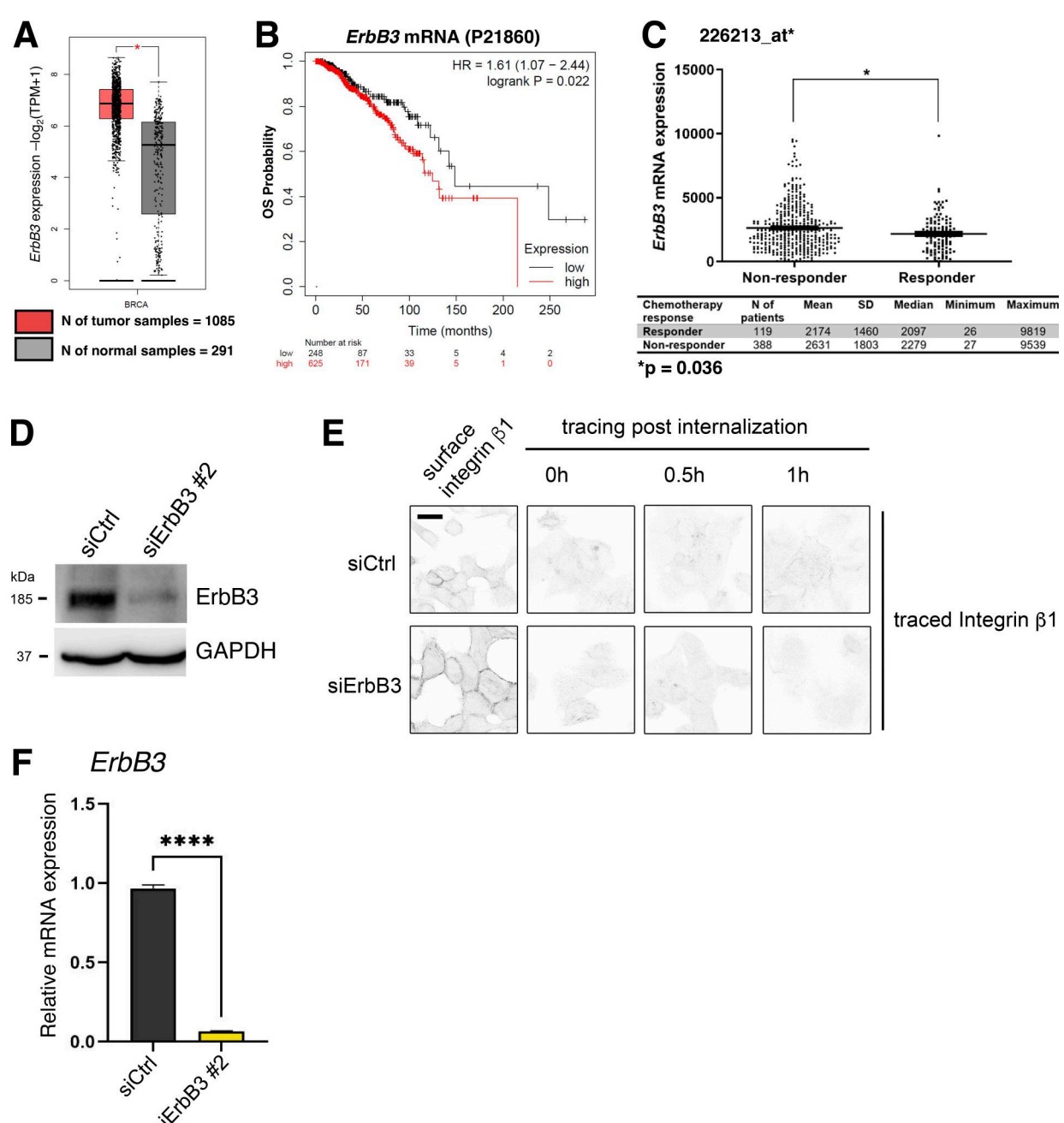

Figure S3. **ErbB3 gene expression correlates with BRCA patient survival and chemotherapy response. (A)** *ErbB3* mRNA expression levels as log$_2$-transformed transcripts per million (TPM) in tumor or corresponding normal tissue from TCGA BRCA tumor vs TCGA normal + GTEx normal datasets were retrieved from the GEPIA2 portal. The data in the bottom indicate the number (N) of tissue samples analyzed in each group, and the differential expression analysis was performed with one-way ANOVA, using disease state as variable for calculating differential expression. Red, *P ≤ 0.01. **(B)** BRCA patient overall survival was calculated after sample stratification and compared with the Kaplan–Meier plotter database by auto select cutoff, based on the low and high expression level of *ErbB3* mRNA. **(C)** BRCA patients were classified as chemotherapy nonresponders or responders based on the relapse-free survival at 5 years after treatment. The diagram presents median *ErbB3* mRNA expression values, along with SEM and minimal and maximal values, as explained in the table below the graph, which also presents the number (N) of analyzed patients. Comparisons were performed using the Mann–Whitney U test (*P ≤ 0.05). **(D)** Representative protein expression of ErbB3 and GAPDH (as a loading control) in the cell extract of MCF7 transiently transfected with control siRNA (siCtrl) or siRNA targeting ErbB3 (#2). **(E)** Confocal immunofluorescence imaging of traced internalized Integrin β1 in MCF7 cells transiently transfected with control siRNA (siCtrl) or siErbB3#2. Integrin β1 that was traced for the indicated times after a 15-min internalization step. **(F)** RT-qPCR analysis of *ErbB3* mRNA levels in MCF7 cells transiently transfected with siRNAs targeting ErbB3 (#2). Values represent fold change of *ErbB3* mRNA expression relative to *GAPDH*. Data are presented as mean values of three biological replicates ± SEM, each in technical duplicates, and P values are shown based on unpaired Student's t test. P values: ****P ≤ 0.0001. Scale bar, 50 µM (E).

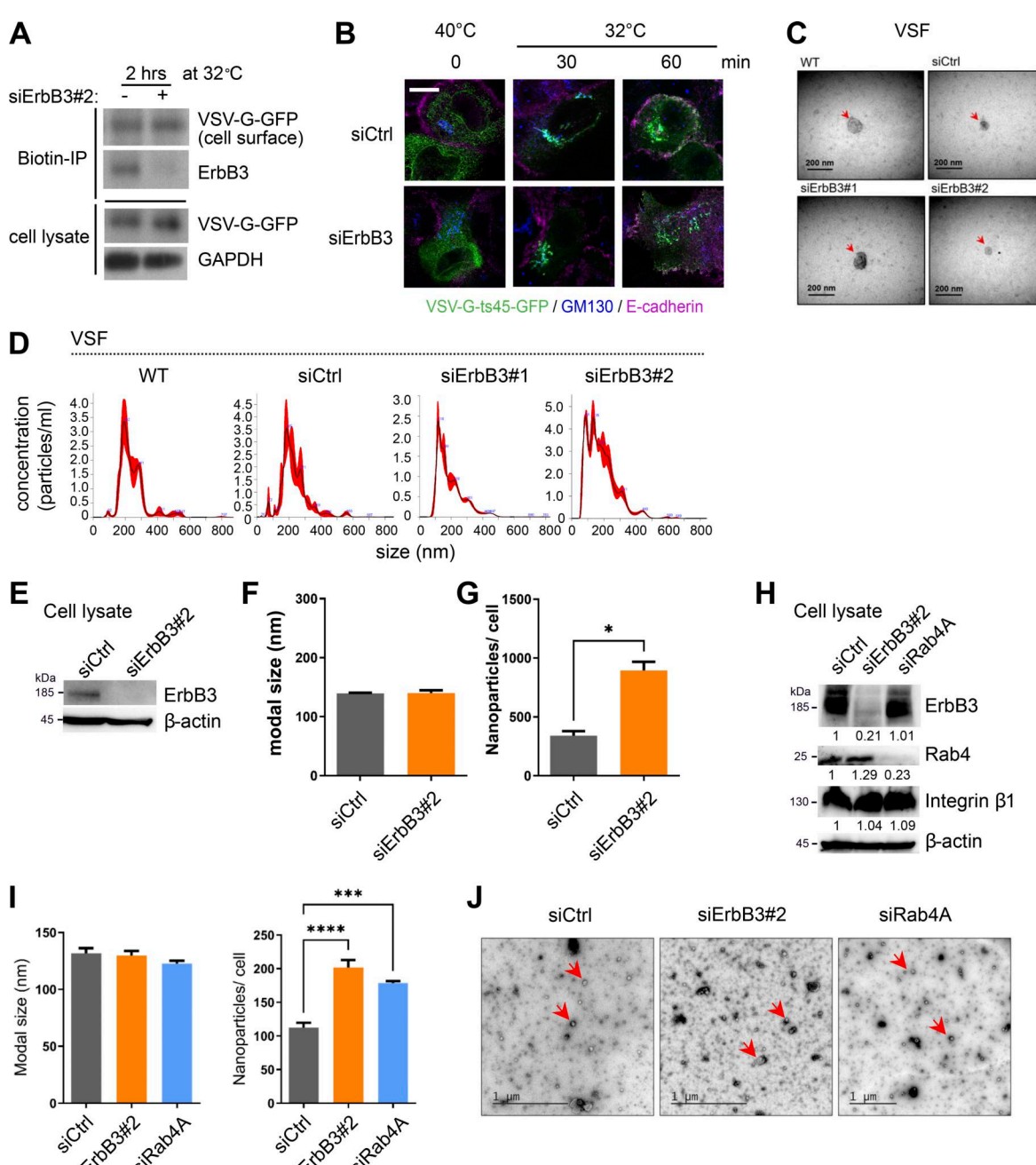

Figure S4. **ErbB3 does not affect exocytic trafficking of VSV-G but reduces EV release. (A)** Immunoblotting analysis of isolated surface-biotinylated VSV-G-ts45-GFP, following its temperature-dependent release from the ER for 2 h in MCF10A cells transiently transfected with control siRNA (siCtrl) or siErbB3#2. Note that ErbB3 depletion does not influence the emergence of VSV-G at the plasma membrane. **(B)** Confocal imaging of VSV-G-ts45-GFP (green) and markers of the TGN (GM130; blue) and the cell surface (E-cadherin; magenta) in control or ErbB3 siRNA-transfected MCF10A cells at nonpermissive temperature (40°C) and following its subsequent shift to permissive (32°C) temperature. Note that VSV-G-ts45-GFP localizes to the ER at 40°C, while after 30 min at permissive temperature, most of the VSV-G-ts45-GFP has moved to the TGN and after 1 h to the cell surface, both in the presence and in the absence of ErbB3. **(C)** Representative TEM micrographs of EVs enriched in the VSF of control or ErbB3-depleted MCF10A cells (EVs are indicated by red arrows). **(D)** Representative nanoparticle traces measured by NTA. The VSF from MCF10A transiently transfected or not (WT) with control siRNA (siCtrl) or independent ErbB3 siRNAs were analyzed by NTA 3.4 software, according to the manufacturer's protocol. 100-nm beads were used as a positive control. **(E)** ErbB3 and β-Actin (as a loading control) protein levels in prHMEC cells transiently transfected with the indicated siRNAs. **(F and G)** EVs enriched in the VSF released by prHMEC transiently transfected with the indicated siRNAs were quantified by NTA in terms of nanoparticle modal size (nm) and number normalized to the total cell number. **(H)** ErbB3, Rab4, Integrin β1, and β-Actin (as a loading control) protein levels in MCF7 cells transiently transfected with the indicated siRNAs. **(I)** EVs enriched in the VSF released by MCF7 transiently transfected with the indicated siRNAs were quantified by NTA in terms of modal size (nm) and number of nanoparticles, upon normalization to the total cell number. **(J)** Representative TEM micrographs of EVs enriched in the VSF of control and ErbB3-depleted or Rab4-depleted MCF7 cells (EVs are indicated by red arrows). Data in F and H are presented as mean values ± SEM. The P value in F was determined by unpaired Student's *t* test, while in H, the P value was determined by one-way ANOVA followed by multiple paired comparisons conducted by Bonferroni's posttest method. *P ≤ 0.05; ***P ≤ 0.001; ****P ≤ 0.0001. **(J)** Scale bars, 10 µm (B), 200 nm (C), 1 µm (J).

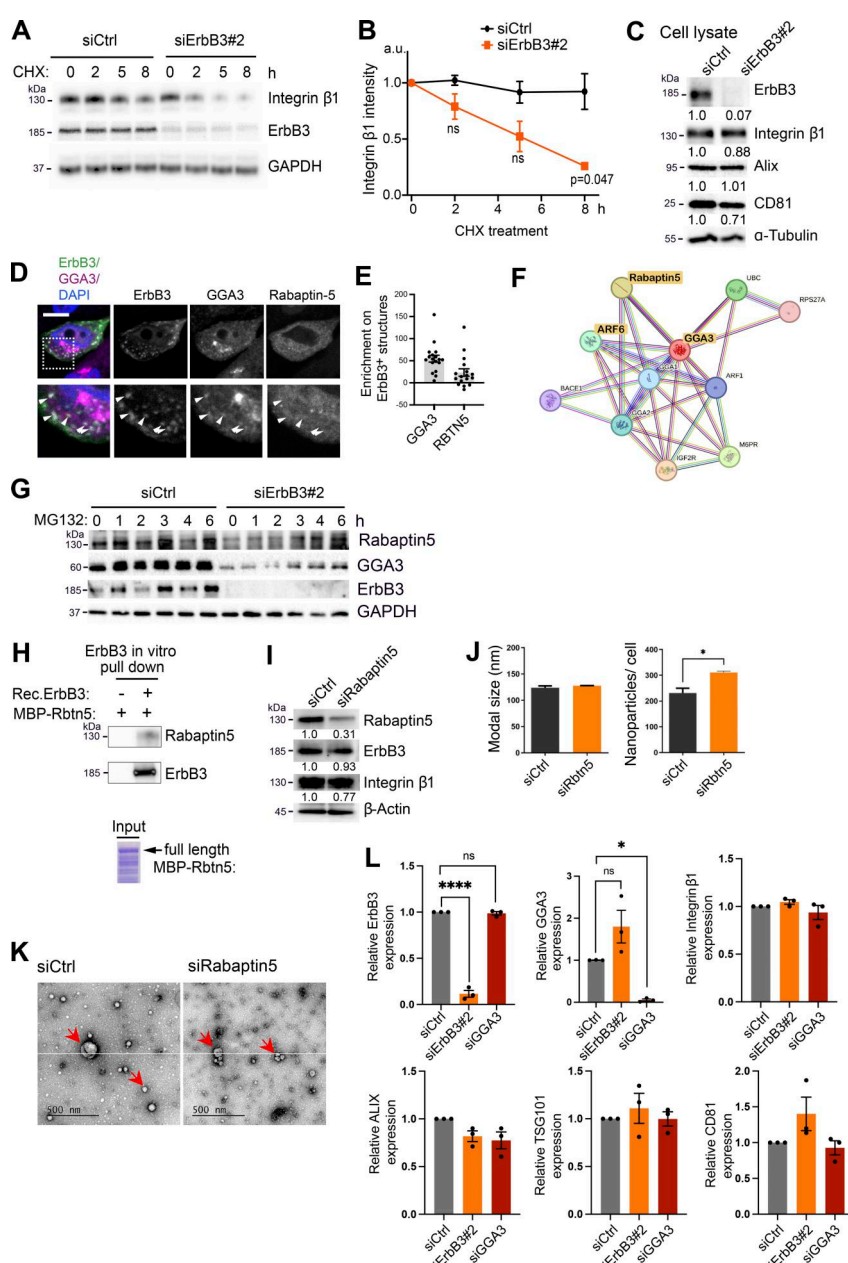

Figure S5. **ErbB3 enhances Integrin β1 stability and binds to GGA3 and Rabaptin5 in vitro. (A)** Immunoblotting analysis of the indicated proteins and GAPDH (as a loading control) in MCF10A cells, transfected with control (siCtrl) or ErbB3 siRNA #2 prior to treatment with protein synthesis inhibitor CHX or vehicle alone for the indicated times in hours. **(B)** Quantification of Integrin β1 band intensities presented as mean values ± SEM; n = 3 independent experiments. **(C)** Protein expression of ErbB3, Integrin β1, and the indicated EV-specific positive (ALIX and TSG101) and α-Tubulin (as a loading control) in the cell extract of MCF10A control or ErbB3-depleted samples. **(D)** Confocal imaging of colocalization of endogenous ErbB3 with ectopically expressed mCherry-GGA3 and endogenous Rabaptin5 in MCF7 cells. **(E)** Analysis of ErbB3 colocalization with mCherry-GGA3 or Rabaptin5. The relative ErbB3 enrichment at the positive colocalization was determined by the formula (a−b)/b, where a is the ErbB3 intensity of mCherry-GGA3 or Rabaptin5 structure center, and b is the adjacent volume (background) for each structure. Each data point represents the minimum average of 20 structures in one cell. **(F)** GGA3 protein–protein interaction network analysis using STRING and a confidence score >0.7. **(G)** Immunoblotting analysis of MCF10A cells, transfected with control (siCtrl) or ErbB3 siRNA #2 prior to treatment with the proteasome inhibitor MG132 (50 μM) for the indicated times in hours. MG132 partially restores GGA3 and Rabaptin5 levels in ErbB3-depleted cells. **(H)** In vitro binding of ErbB3 to Rabaptin5. Immunoblotting of ErbB3 immunoprecipitates following incubation of recombinant ErbB3 and Rabaptin5. **(I)** Rabaptin5, ErbB3, Integrin β1, and β-Actin (as a loading control) protein levels in MCF7 cells transiently transfected with control siRNA (siCtrl) or siRabaptin5. **(J)** EVs enriched in the VSF released by MCF7 transiently transfected with siCtrl or siRabaptin5 were quantified by NTA in terms of nanoparticle modal size (nm) and number upon normalization to the total cell number. **(K)** Representative TEM micrographs of EVs enriched in the VSF of control or Rabaptin5-depleted MCF7 cells (EVs are indicated by red arrows). **(L)** Related to Fig. 6 J. Densitometric quantification of the indicated protein expression in MCF10A cells is presented as the mean ± SEM from three biological replicates, normalized to the corresponding siCtrl condition. Data are presented as mean values ± SEM. The P value in B was determined by two-tailed paired Student's t test, while in J, the P value was determined by unpaired Student's t test. P values in L were determined by one-way ANOVA followed by multiple paired comparisons conducted by Bonferroni's posttest method. *P ≤ 0.05; ****P ≤ 0.0001. Scale bars, 10 μm (B), 500 nm (C).

