## [Peer Review File · The Journal of Cell Biology]

Loss of ErbB3 redirects Integrin β 1 from endosomal recycling to secretion in extracellular vesicles

Dorival Rodrigues-Junior, Ana Saez-Ibanez, Takeshi Terabayashi, Nina Daubel, Taija Mäkinen, Olof Idevall-Hagren, Aristidis Moustakas, and Ingvar Ferby

Corresponding Author(s): Dorival Rodrigues-Junior, Uppsala University

Review Timeline:

Submission Date:	2025-01-30
Editorial Decision:	2025-02-18
Revision Received:	2025-08-19
Editorial Decision:	2025-09-09
Revision Received:	2025-10-02

Monitoring Editor: Johanna Ivaska

Scientific Editor: Andrea Marat

Transaction Report:

DOI: <https://doi.org/10.1083/jcb.202501255>

Revision 0

Review #1

1. Evidence, reproducibility and clarity:

Evidence, reproducibility and clarity (Required)

ErbB3 is well-known for its significance in cancer, which is dependent on ligand-binding and heterodimerization with other ErbB family members. In the current work, Rodrigues-Junior et al. identified novel, unexpected functions of ErbB3 in promoting early endocytic recycling and restricting exocytic trafficking (extracellular vesicles secretion) of membrane receptors, such as integrin b1 and transferrin receptor, via stabilizing the Arf6-GGA3-Rabaptin5 endosomal sorting complex.

Via ErbB3 siRNA knockdown, they observed an impaired recycling of transferrin receptor and integrin b1 back to the cell membrane. The recycling assay condition (growth factor-deprived) provided a very clean result to support that this ErbB3-dependent endocytic trafficking is ligand-binding independent. The trafficking-dependence on ErbB3 (both the endocytic and the exocytic) was further supported by integrin b1 functional assays (scratch closure assay and Matrigel invasion assay). There are still some details that need to be clarified to fully understand the conclusion.

****Major points:****

1. The manuscript started with a pathological correlation between high ErbB3 level and poor patient survival rate. In Fig.1, the impaired TfR recycling, and the co-localization between ErbB3 and integrin b1 were also performed in the pathological breast cancer cell line, MCF7. While investigating integrin b1 recycling, the authors suddenly switched to another two non-malignant human breast epithelial cell lines, which led to a difficult correlation of ErbB3-mediated recycling back to the disease situation. The authors should state more clearly this point, rather than data not shown. This inconsistency occurred also in other assays, for example, when addressing the trafficking from TGN to cell surface, MCF7 was utilized; while when addressing extracellular vesicle secretion, MCF10A was utilized.
2. It was shown before that ErbB3 undergoes constitutive internalization and degradation within several hours that is independent of ligand-binding (ref#13). Can the authors provide experimental evidences to show the correlation of TfR or integrin b1 recycling with this dynamic ErbB3 levels rather than ErbB3 knockdown?
3. The efficiency of siRNA knockdown of ErbB3 (both #1 and #2) should support the

observed phenotype (Fig. 1I-J, K-L). Is there a correlation between the ErbB3 level with integrin recycling? For example, siRNA#2 led to more efficient knockdown of ErbB3 in MCF10A?

4. ErbB3 loss led to more extracellular vesicles secretion, but also lysosomal degradation of integrin b1. This conclusion is supported by results shown in Fig.4D-E and Fig. S8A-B, while the analysis from the same cell line (MCF10A, Fig. S3A) results in no change of integrin b1 levels upon ErbB3 depletion. Fig. S3B showed also no change in a second non-malignant cell line (prHMEC). How do the authors explain this conflict?

****Minor points:****

1. Is TfR also colocalizing with endogenous ErbB3?

2. Fig. 3J, TSG101, T is masked by 3I

3. Page 10, the description of the EV secretion in prHMEC cells is annotated to the wrong figure.

Fig S5D◇ S7D; S5E◇ S7E

4. Fig. 4M: How was the motility/invasion into Matrigel determined? Images? Only quantifications are shown.

5. Fig. 4M: Exosomes collected from ErbB3-depleted cells promotes the migration in MCF10A-wild type cells, how about the effects on ErbB3-depleted cells? This group should be included for analysis.

6. Quantification of the blots should be provided for Fig. 5A (GGA3), 5B (GGA3, Rabaptin5 and Arf6), 5F (GGA3) and 5G (GGA3, Rabaptin5 and Arf6). What is mock IP in each graph? The mock IP is neither mentioned in methods nor in legends.

2. Significance:

Significance (Required)

***Strength:** The recycling assay condition (growth factor-deprived) provided a very clean result to support that this ErbB3-dependent endocytic trafficking is ligand-binding independent.

***Limitations:** Constantly change cell lines when addressing different questions

3. How much time do you estimate the authors will need to complete the suggested revisions:

Estimated time to Complete Revisions (Required)

(Decision Recommendation)

Between 1 and 3 months

4. Review Commons values the work of reviewers and encourages them to get credit for their work. Select 'Yes' below to register your reviewing activity at Web of Science Reviewer Recognition Service (formerly Publons); note that the content of your review will not be visible on Web of Science.

Yes

Review #2

1. Evidence, reproducibility and clarity:

Evidence, reproducibility and clarity (Required)

****Summary:****

In their manuscript, Rodrigues-Junior and colleagues identify a novel ligand-independent function of the tyrosine kinase receptor (RTK) ErbB3 as a regulator of integrin $\beta 1$ recycling. In particular, the authors demonstrate that ErbB3 depletion reduce $\beta 1$ integrin surface expression, triggering its lysosomal degradation and increasing its secretion in extracellular vesicles (EVs). Moreover, the authors show that these EVs enhanced the invasive capacity of ErbB3 wild type breast epithelial cells. In addition, the authors evidence the interaction between ErbB3, GGA3 and Rabaptin5. Loss of any of these proteins destabilizes this interaction, which abrogates integrin $\beta 1$ recycling and leads to its degradation and secretion.

The work is potentially interesting; however, there are some aspects that need to be analyzed in a more robust manner.

****Major comments:****

1. The manuscript is mainly focused on $\beta 1$ integrin endocytic and post-endocytic fate following ErbB3 silencing, describing also a molecular mechanism underlying these observations. Despite the cited manuscript by Deneka, A. and colleagues indicates a

similar mechanism for transferrin receptor (TfR) recycling, the Authors only studied the receptor internalization upon ErbB3 silencing. Therefore, this observation does not add any significance to the main topic of the manuscript and its removal should be considered.

2. Data from Figure S1A seems to be not normally distributed. Have the Authors tested the data for normal distribution? If not, please consider it. If the data is not normally distributed, a non parametric Mann-Whitney U-Test would be more suitable.

3. The Authors studied the colocalization of ErbB3, Rab4 and Rab11, observing an increased colocalization between ErbB3 and Rab4 10 minutes following primaquine. However, the Authors previously referred to Sönnichsen, B et al. manuscript, in which TfR colocalized with Rab11 at 30min. It would be interesting to see whether ErbB3 and Rab11 colocalize at later time points in the presence or absence of primaquine. This will reinforce the conclusion that ErbB3 is involved in early Rab4-dependent recycling.

4. In Figure 4C the Authors observed a reduction in $\beta 1$ integrin levels in ErbB3 silenced cells compared to the control already at the beginning of tracing (0 min), which might be due to accelerated turnover at the internalization step of their experimental design. To confirm this, immunofluorescence of $\beta 1$ integrin in control and ErbB3 silenced cells could be performed just right after the 15min integrin internalization.

5. In the discussion, the Authors indicate that "loss of ErbB3 redirects Integrin $\beta 1$ towards lysosomes for degradation, mimicking loss of GGA3 that similarly redirects both Integrin $\beta 123$ and c-Met towards lysosomal degradation²¹, or Rabaptin5 depletion that we find similarly redirects trafficking of internalised Integrin $\beta 1$ towards lysosomal degradation". However, the involvement of lysosomal degradation was only studied for ErbB3 silencing by employing chloroquine. To further support this statement, the use of chloroquine in Rabaptin5- and GGA3-depleted cells is recommended.

****Minor comments:****

6. The Authors should consider shortening the following sentences from the Introduction: "GGA proteins contain several functional domains that...thereby regulating sorting of cargo including Integrin $\beta 3$ and TfR into recycling endosomes".

7. The Authors do not show ErbB3 silencing efficiency at the protein level until Figure 3G, which should have been shown in Figure 1 or Supplementary Figure 1, as all the research is based on it. Moreover, GGA3 silencing efficiency was never tested.

8. Figure 1I and Figure 1K may include the representative images for the missing siErbB3 to

properly illustrate the associated quantifications

9. Consider including a Western blot showing the effect of lapatinib in EGFR, ErbB2 and ErbB3 protein expression, including their phosphorylated forms.

10. Some supplementary figures are mislabeled, such as Supplementary Figure S5D and S5E on page 10, which should be S7D and S7E, respectively. Supplementary Figure S7C on page 15 should be S9C.

11. The following sentence on page 8 should be revised as a verb is missing: "which corresponds to the reported peak time when colocalization of Rab4 with traced TfR, preceding Rab11 and TfR colocalization that peaks later at 30 minutes".

12. The main text indicates that the amount of VSV-G transported to the cell surface after 30min it is not affected by ErbB3 silencing. However, in Figure 3E seems to slightly decrease following the silencing. The Authors may consider employing another Western blot image to match the main text and the quantification in Figure 3F.

13. In the main text, a significant difference in the nanoparticles/cell between ErbB3-depleted cells and wild type or control cells were reported. However, Figure 3I only showed the statistics of each siRNA vs the control and not the wild type condition.

14. The Authors concluded that "chloroquine treatment significantly restored traced Integrin β 1 levels". However, this conclusion is not reflected in the statistical analysis reported in Figure 4H, which only showed the differences between control and ErbB3 silenced cells. Thus, the statistics reported for the chloroquine results should be added.

15. The Authors concluded that "loss of either GGA3 or Rabaptin5 mimics the effect of loss of ErbB3 on endocytic trafficking of Integrin β 1, consistent with the hypothesis that GGA3 and Rabaptin5 are effectors of ErbB3 in promoting endosomal recycling and impeding EV release". To confirm this conclusion, the inclusion of siRabaptin5 results in Figures 6H and 6J is suggested.

16. To be consistent with the results presentation:

- The inclusion of Modal size is recommended in Figure 6I.
- Some graphs show the number of cells or biological replicates while other ones no.
- Figure 4E showed different time points for both siRNAs.

17. Figure 1E represents the squared regions of Figure 1D, but it is not indicated in the figure legend.

18. In the legend of Figure 1D-G, 30min of integrin internalization is reported, where it should be 15min according to main text and methods.

19. The addition of representative images in Figure 6A is recommended, as already present in Figure 1I.

20. As two different siRNAs for ErbB3 were used and not in all experiments, the employed siRNA should be indicated in each experiment. In the cases where both ErbB3 siRNAs were employed, figures should report them either as main results or supplementary.

21. Why do the Authors use EVs enriched in the VSF or by UC to show the same result? What is the criteria to choose one or the other one? For example in Figures 6G and 6K.

2. Significance:

Significance (Required)

Various studies highlight the involvement of the RTK ErbB3 in cancer development, as well as its potential use as a biomarker for prognosis and therapy resistance. It is also known that ErbB3 is constitutively internalized and degraded, in a process controlled by PKC (Dietrich, M. et al. 2019, Exp Cell Res). However, the novelty of this manuscript resides in the idea that ErbB3, as other transmembrane receptors, may regulate the endocytosis and post-endocytic fate of different cargoes, such as integrin β 1. The discovery and understanding of new molecular mechanisms might help in the identification of new potential targets for cancer treatment, as well as other diseases in which the endocytic pathways are altered.

Field of expertise: integrin-mediated cell adhesion and migration, integrin endocytosis and recycling

3. How much time do you estimate the authors will need to complete the suggested revisions:

Estimated time to Complete Revisions (Required)

(Decision Recommendation)

Between 1 and 3 months

4. Review Commons values the work of reviewers and encourages them to get credit for their work. Select 'Yes' below to register your reviewing activity at Web of Science Reviewer Recognition Service (formerly Publons); note that the content of your review will not be visible on Web of Science.

Yes

Review #3

1. Evidence, reproducibility and clarity:

Evidence, reproducibility and clarity (Required)

The paper by Dorival Mendes Rodrigues-Junior et al., focuses on a novel ligand-independent role of ErbB3 receptor, modulating Transferrin receptor and integrin beta1 early recycling. Authors perform several in vitro studies where they show how ErbB3 depletion diverts integrin beta1 from recycling towards lysosomal degradation and extracellular vesicle secretion, impairing cell migration. They also provide mechanistic experiments showing the role of ErbB3 on Arf6-GGA3-Rabaptin5 endosomal complex assembly.

****Major comments:****

- Fig. 1. Authors should co-stain with early endosomal markers (such as EEA1) to clearly show endogenous ErbB3 and Beta1 integrin endosomal co-localization. Including some insets with higher magnifications would also improve visual inspection of such interactions.
- Fig. 1H and 1I. Authors need to provide TIRF penetration depth to better evaluate the potential cytosolic contribution. Additionally, plasma membrane purification studies would help to validate their live imaging results.
- Fig. 1J. Authors should explain better how they calculated normalized fluorescence
- Fig. 1J, 1M-1L: beta1 integrin endocytic recycling should be compared across the same time-points to better evaluate kinetic differences.
- Fig. 2B. Authors should include some plasma membrane markers (such as WGA) to better localize cell surface after beta1 integrin tracing.
- Fig. 3. Author should consider adding additional experiments with Rab4 and Rab11 dominant negative forms to validate their results.
- Fig. 4M. To validate authors' claim on the role of integrin Beta1-containing EVs on invasive behaviour, they should repeat the experiment using blocking beta1 antibodies prior to EV addition.
- While authors claim that their results could potentially clarify different aspects of tumour dissemination, most of their experiments are done in MCF10A, a non-tumorigenic epithelial cell line. To better support their conclusion, they should reproduce key experiments in MCF7 or other tumorigenic cell line.

****Minor comments:****

- Fig. 1D-1F: please explain better if beta1 integrin surface signal was quenched in these specific set of studies
- Suppl. Fig. 3A: last WB lane should read "siErB2" instead of "siErbB3".

2. Significance:

Significance (Required)

The paper gathers important observations showing a new role of ErbB3 in vesicular trafficking. While these results provide new mechanistic insights that potentially deepen our understanding of tumor dissemination, most of the experiments are done with a non-tumorigenic cell line, and therefore key results should be validated in a tumor cell line context before considering for publication.

The evidence gathered could be of interest for experts across different biomedical fields, specially within cellular and molecular oncology

My expertise: cell competition, cancer, mechanobiology, integrin recycling

3. How much time do you estimate the authors will need to complete the suggested revisions:

Estimated time to Complete Revisions (Required)

(Decision Recommendation)

Between 1 and 3 months

Yes

Manuscript number: RC-2024-02662

Corresponding author(s): Dorival Mendes Rodrigues-Junior; Ingvar Ferby

1. General Statements

We are grateful for the overall positive feedback and constructive suggestions. We have been able to experimentally address several of the suggested points and provide here a revision plan addressing all of the reviewers' additional concerns.

In summary, this study is of fundamental novelty and high impact as it:

- 1. Reveals an unexpected role of ErbB3 in controlling Integrin β 1 trafficking and thus epithelial cell motility and extracellular vesicles secretion. This may shed important insights into the role of ErbB3 in cancer.***
- 2. Uncovers the first ligand-independent, non-canonical cellular function for ErbB3 as a scaffold for the Arf6-Rabaptin5-GGA3 endosomal sorting complex.***
- 3. Provoking the notion that pseudo-RTKs may have evolved cellular functions beyond receptor signaling, such as by scaffolding endosomal sorting compartments.***

We hope that you share our view that these conceptually ground breaking findings will be of interest to a broad cross-disciplinary audience interested in cell signaling, cancer biology, endocytic trafficking and integrin biology.

Reviewer #1 (Evidence, reproducibility and clarity (Required)):

ErbB3 is well-known for its significance in cancer, which is dependent on ligand-binding and heterodimerization with other ErbB family members. In the current work, Rodrigues-Junior et al. identified novel, unexpected functions of ErbB3 in promoting early endocytic recycling and restricting exocytic trafficking (extracellular vesicles secretion) of membrane receptors, such as integrin b1 and transferrin receptor, via stabilizing the Arf6-GGA3-Rabaptin5 endosomal sorting complex. Via ErbB3 siRNA knockdown, they observed an impaired recycling of transferrin receptor and integrin b1 back to the cell membrane. The recycling assay condition (growth factor-deprived) provided a very clean result to support that this ErbB3-dependent endocytic trafficking is ligand-binding independent. The trafficking-dependence on ErbB3 (both the endocytic and the exocytic) was further supported by integrin b1 functional assays (scratch closure assay and Matrigel invasion assay). There are still some details that need to be clarified to fully understand the conclusion.

Major points:

1. The manuscript started with a pathological correlation between high ErbB3 level and poor patient survival rate. In Fig.1, the impaired TfR recycling, and the co-localization between ErbB3 and integrin b1 were also performed in the pathological breast cancer cell line, MCF7. While investigating integrin b1 recycling, the authors suddenly switched to another two non-malignant human breast epithelial cell lines, which led to a difficult correlation of ErbB3-mediated recycling back to the disease situation. The authors should state more clearly this point, rather than data not shown. This inconsistency occurred also in other assays, for example, when addressing the trafficking from TGN to cell surface, MCF7 was utilized; while when addressing extracellular vesicle secretion, MCF10A was utilized.

Response: we thank the reviewer for the comment. The rationale for using different cell-lines or primary cells is now better explained in the manuscript. We found that depletion of ErbB3 impaired recycling of Integrin β 1 in the non-malignant cells, including MCF10A and primary breast epithelial cells, but not in malignant MCF7 cells that overexpress ErbB3 (data not shown). We now speculate in the manuscript that perhaps the dependence on ErbB3 for Integrin b1 recycling is lost at some point during carcinogenesis, although further studies will be needed to address this possibility. MCF7 cells were used to detect endogenous ErbB3 as normal expression levels of ErbB3 (primary MECs and MCF10A) were not detectable by immunofluorescence microscopy in our hands with a range of antibodies we tested. With regard to the transferrin recycling assay, we first attempted to use MCF10A cells for consistency, however we found that transferrin internalized poorly in these cells and the limited pool of transferrin that internalised was retained in these cells for an extended time (3 h), thus rendering them unsuitable for our transferrin experiments.

Concerning the data on trafficking from the TGN to cell surface we mistakenly wrote that they were performed in MCF7 cells although they were in fact done in MCF10A cells. This is now corrected in the new version of this manuscript.

Additionally, based on the constructive comment by this reviewer, we have now extended the analysis of EV secretion in ErbB3, Rab4 and Rabaptin5 silenced cells to MCF7 cells. The new data is in line with our findings in MCF10A and prHMEC cells, that absence of ErbB3 significantly increased EV secretion. Moreover, Rab4 and Rabaptin5 knockdown also enhanced the amount of EVs secreted by MCF7 cells. These results were incorporated in the manuscript as new Supplementary Figure S7F-G and new Supplementary Figure S9F-G, as recommended. Furthermore, we also included in this new version that GGA3 and to a lesser extent Rab GTPase-binding effector protein 1 (Rabaptin5 or RABPT5) shared colocalisation with endogenous ErbB3 in MCF7 cells as the new Supplementary Figure 9A, B. Finally, we also attempted to conduct the Arf6 IP

in MCF7 cells, but as opposed to MCF10A cells, the yield of Arf6 in pull down experiments was much lower than in MCF10A cells (see figure below – reviewers only), and interacting proteins were not detectable.

Suppl. Figure S7

Suppl. Figure S9

MCF7: Arf6 IP (Reviewers only)

2. It was shown before that ErbB3 undergoes constitutive internalization and degradation within several hours that is independent of ligand-binding (ref#13). Can the authors provide experimental evidences to show the correlation of TfR or integrin b1 recycling with this dynamic ErbB3 levels rather than ErbB3 knockdown?

Response: we have performed colocalization of ErbB3, traced Integrin β 1 and the recycling endosome marker EHD1, showing triple colocalization in a subset of endosomes, as shown in the new Supplementary Figure S2H. Experimental limitations prevented us from including EEA1 in triple staining for mCherry-ErbB3 or endogenous ErbB3 protein, please note the figures below (for reviewers). Furthermore, ectopically expressed ErbB3 in MCF10A cells did not show convincing co-localisation (see example below). We hope that the new EHD1 triple colocalization with ErbB3 and Integrin β 1 in endosomal compartments satisfies this specific comment.

Supplementary Figure S2

MCF10A (Confocal microscopy) – Revision only

As mentioned above, regarding the transferrin recycling assay, we first attempted to use MCF10A cells for consistency, however we found that transferrin internalized poorly in these cells and the limited pool of transferrin that internalised was retained in these cells for an extended time (3 h), thus preventing their use.

3. The efficiency of siRNA knockdown of ErbB3 (both #1 and #2) should support the observed phenotype (Fig. 1I-J, K-L). Is there a correlation between the ErbB3 level with integrin recycling? For example, siRNA#2 led to more efficient knockdown of ErbB3 in MCF10A?

Response: notably, the immunoblots presented here to assess the efficiency of the two different siRNAs are one example and we noted some variability between different experiments but find that both siRNAs work well and yield comparable effects on recycling of Integrin β 1. Importantly, the recycling data represents biological repeats of independently performed experiments, and have yielded reproducible and consistent ErbB3 silencing using both siRNAs. This is noted by the lack of significance between ErbB3 knocked down cells in Fig. 1I-J and K-L. Hence, we consider that both siRNAs against ErbB3 worked efficiently with comparable outcome. Please also note our reply to Rev2 #07.

4. ErbB3 loss led to more extracellular vesicles secretion, but also lysosomal degradation of integrin b1. This conclusion is supported by results shown in Fig.4D-E and Fig. S8A-B, while the analysis from the same cell line (MCF10A, Fig. S3A) results in no change of integrin b1 levels upon ErbB3 depletion. Fig. S3B showed also no change in a second non-malignant cell line (prHMEC). How do the authors explain this conflict?

Response: we thank the reviewer for this comment. We believe that the increase in EV secretion and lysosomal degradation is compensated by increase in de novo synthesis of Integrin β 1 (see data below, from Fig. S3C). In the original manuscript we did not perform the appropriate statistical analysis of the RT-qPCR data. The unpaired two-tail Student's T-test is only suitable for normally distributed samples, which is not the case here. Instead, we performed the appropriate Mann-Whitney U-test assuming non-normal distribution, yielding an exact p-value of 0.017. The figure S3A and associated text has been modified accordingly.

Suppl. Figure S3

Minor points:

1. Is TfR also colocalizing with endogenous ErbB3?

Response: as mentioned in the major comment #02, we attempted to perform the transferrin recycling assay using MCF10A cells to enable direct comparisons with the integrin b1 recycling, but found that transferrin internalized poorly in these cells.

2. Fig. 3J, TSG101, T is masked by 3I

Response: we apologize for this oversight. We have gone through the manuscript in detail and corrected all pointed errors accordingly.

3. Page 10, the description of the EV secretion in prHMEC cells is annotated to the wrong figure. Fig S5D◇ S7D; S5E◇ S7E

Response: we apologize for this oversight and have now corrected the mistake.

4. Fig. 4M: How was the motility/invasion into Matrigel determined? Images? Only quantifications are shown.

Response: the matrigel invasion assay was described in the Material and Methods section. Accordingly, the data were expressed as the percentage of invasion based on the ratio of the mean number of cells invading through Matrigel matrix per mean number of cells in the uncoated support. For this rebuttal letter, the reviewer can find representative micrographs of invaded MCF10A siCtrl non-treated (Ctrl) or treated with VSF secreted from MCF10A siCtrl or siErbB3. Since this is an established method to measure cell invasion, we hope the reviewer agrees that these images do not add value to the manuscript. Please see figure below (for reviewers).

(Revision only)

5. Fig. 4M: Exosomes collected from ErbB3-depleted cells promotes the migration in MCF10A-wild type cells, how about the effects on ErbB3-depleted cells? This group should be included for analysis.

Response: as proposed, we have treated both control and ErbB3-silenced MCF10A cells with normalized concentrations of EVs secreted from siCtrl and siErbB3 (1×10^9 nanoparticles/ mL) for 48 hours, followed by cell viability and cell invasion assays. The new data show that both EV pools modestly increased cell viability and substantially increased invasiveness of both wild-type and ErbB3-depleted cells through Matrigel (new Figures 4K and L). Together, our results indicate that while ErbB3-silenced MCF10A cells exhibited lower basal motility, ErbB3 is not required for the observed EV induced motility. The new Figures 4K and L were included and further discussed in this manuscript.

Figure 4

6. Quantification of the blots should be provided for Fig. 5A (GGA3), 5B (GGA3, Rabaptin5 and Arf6), 5F (GGA3) and 5G (GGA3, Rabaptin5 and Arf6). What is mock IP in each graph? The mock IP is neither mentioned in methods nor in legends.

Response: we have now carried out densitometry analysis in all the requested immunoblots shown in Figure 5. We also changed the mock IP term to IgG IP for clarity. The use of non-immunogenic IgG in control IPs is now specified in the methods and respective figure legend.

Reviewer #2 (Evidence, reproducibility and clarity (Required)):**Summary:**

In their manuscript, Rodrigues-Junior and colleagues identify a novel ligand-independent function of the tyrosine kinase receptor (RTK) ErbB3 as a regulator of integrin $\beta 1$ recycling. In particular, the authors demonstrate that ErbB3 depletion reduce $\beta 1$ integrin surface expression, triggering its lysosomal degradation and increasing its secretion in extracellular vesicles (EVs). Moreover, the authors show that these EVs enhanced the invasive capacity of ErbB3 wild type breast epithelial cells. In addition, the authors evidence the interaction between ErbB3, GGA3 and Rabaptin5. Loss of any of these proteins destabilizes this interaction, which abrogates integrin $\beta 1$ recycling and leads to its degradation and secretion. The work is potentially interesting; however, there are some aspects that need to be analyzed in a more robust manner.

Major comments:

1. The manuscript is mainly focused on $\beta 1$ integrin endocytic and post-endocytic fate following ErbB3 silencing, describing also a molecular mechanism underlying these observations. Despite the cited manuscript by Deneka, A. and colleagues indicates a similar mechanism for transferrin receptor (TfR) recycling, the Authors only studied the receptor internalization upon ErbB3 silencing. Therefore, this observation does not add any significance to the main topic of the manuscript and its removal should be considered.

Response: we agree with the reviewer the fate of Integrin $\beta 1$ is the main focus of this manuscript. We would however favour retaining the TfR data as it implies a wider role of ErbB3, beyond trafficking of Integrin $\beta 1$. We ask for the reviewer's understanding of our rationale.

2.Data from Figure S1A seems to be not normally distributed. Have the Authors tested the data for normal distribution? If not, please consider it. If the data is not normally distributed, a non parametric Mann-Whitney U-Test would be more suitable.

Response: we thank the reviewer for the comment. The differential ErbB3 mRNA expression analysis was retrieved from the widely used GEPIA2 portal (to date about 600 manuscripts cite this portal on PubMed), based on the selected datasets ("TCGA tumors vs TCGA normal + GTEx normal" or "TCGA tumors vs TCGA normal"). The method for differential analysis is one-way ANOVA, using disease state (Tumor or Normal) as variable for calculating differential expression, as it considers differential expression among several tumors.

70. Tang, Z., Kang, B., Li, C., Chen, T., and Zhang, Z. (2019). GEPIA2: an enhanced web server for large-scale expression profiling and interactive analysis. Nucleic Acids Res. 47, W556–W560. <https://doi.org/10.1093/nar/gkz430>.

3. The Authors studied the colocalization of ErbB3, Rab4 and Rab11, observing an increased colocalization between ErbB3 and Rab4 10 minutes following primaquine. However, the Authors previously referred to Sönnichsen, B et al. manuscript, in which TfR colocalized with Rab11 at 30min. It would be interesting to see whether ErbB3 and Rab11 colocalize at later time points in the presence or absence of primaquine. This will reinforce the conclusion that ErbB3 is involved in early Rab4-dependent recycling.

Response: we appreciate the reviewer's comment. However, we consider that these requested experiments will not add significant value to the novelty of this manuscript and hope that the reviewer accepts that we politely refrain from reproducing them.

4. In Figure 4C the Authors observed a reduction in $\beta 1$ integrin levels in ErbB3 silenced cells compared to the control already at the beginning of tracing (0 min), which might be due to accelerated turnover at the internalization step of their experimental design. To confirm this, immunofluorescence of $\beta 1$ integrin in control and ErbB3 silenced cells could be performed just right after the 15min integrin internalization.

Response: this is likely a misunderstanding as the timepoint (0 min) is defined as the point after the 15 min internalization step when the imaging-based tracing begins, which aligns perfectly with the reviewer's request.

5. In the discussion, the Authors indicate that "loss of ErbB3 redirects Integrin $\beta 1$ towards lysosomes for degradation, mimicking loss of GGA3 that similarly redirects both Integrin $\beta 1$ and c-Met towards lysosomal degradation, or Rabaptin5 depletion that we find similarly redirects trafficking of internalised Integrin $\beta 1$ towards lysosomal degradation". However, the involvement of lysosomal degradation was only studied for ErbB3 silencing by employing chloroquine. To further support this statement, the use of chloroquine in Rabaptin5- and GGA3-depleted cells is recommended.

Response: we appreciate the reviewer's comment, but since these findings have been published earlier, we think that they will not add significant value to the manuscript and hope that the reviewer accepts that we politely refrain from reproducing them.

Minor comments:

6. The Authors should consider shortening the following sentences from the Introduction: "GGA proteins contain several functional domains that...thereby regulating sorting of cargo including Integrin $\beta 3$ and TfR into recycling endosomes".

Response: we thank the reviewer for the comment. We have now divided this sentence into two for smoother reading.

7. The Authors do not show ErbB3 silencing efficiency at the protein level until Figure 3G,

Full Revision

which should have been shown in Figure 1 or Supplementary Figure 1, as all the research is based on it. Moreover, GGA3 silencing efficiency was never tested.

Response: we thank the reviewer for this comment. We have included a new immunoblot confirming the silencing of ErbB3 by two independent siRNAs in MCF7 cells, as the new Supplementary Figure S2A. Please, note that GGA3 silencing was shown in the main Figure 6J.

8. Figure 1I and Figure 1K may include the representative images for the missing siErbB3 to properly illustrate the associated quantifications.

Response: we thank the reviewer for the comment. We have now included the representative images, as suggested.

9. Consider including a Western blot showing the effect of lapatinib in EGFR, ErbB2 and ErbB3 protein expression, including their phosphorylated forms.

Response: we thank the reviewer for the comment. As requested, we now show that at used concentration, lapatinib efficiently blocked tyrosine phosphorylation of ErbB3 and ERK1/2, without perturbing EGFR or ErbB3 expression levels. We also considered it relevant to show that 1 μM lapatinib used was not cytotoxic to MCF10A and MCF7 cells. We hope that these new results satisfy this specific request.

10. Some supplementary figures are mislabelled, such as Supplementary Figure S5D and S5E on page 10, which should be S7D and S7E, respectively. Supplementary Figure S7C on page 15 should be S9C.

Response: we apologize for this oversight and have performed the corrections.

11. The following sentence on page 8 should be revised as a verb is missing: "which

corresponds to the reported peak time when colocalization of Rab4 with traced TfR, preceding Rab11 and TfR colocalization that peaks later at 30 minutes".

Response: we apologize for this oversight. It now reads: "which corresponds to the reported peak time of colocalization of Rab4 with traced TfR, which precedes Rab11 and TfR colocalization that peaks later at 30 minutes".

12. The main text indicates that the amount of VSV-G transported to the cell surface after 30min it is not affected by ErbB3 silencing. However, in Figure 3E seems to slightly decrease following the silencing. The Authors may consider employing another Western blot image to match the main text and the quantification in Figure 3F.

Response: as the reviewer noted the immunoblot showed a slight decrease. It is however a very modest decrease that is also observed in the positive control (MUC1) in the same Streptavidin IP sample. We ask for permission to keep these representative images.

13. In the main text, a significant difference in the nanoparticles/cell between ErbB3-depleted cells and wild type or control cells were reported. However, Figure 3I only showed the statistics of each siRNA vs the control and not the wild type condition.

Response: we apologize for this oversight. We removed from the text the comparison with the wild-type non-transfected cells to avoid misunderstanding.

14. The Authors concluded that "chloroquine treatment significantly restored traced Integrin β 1 levels". However, this conclusion is not reflected in the statistical analysis reported in Figure 4H, which only showed the differences between control and ErbB3 silenced cells. Thus, the statistics reported for the chloroquine results should be added.

Response: we appreciate the comment by the reviewer. The requested comparison is now included in the new Figure 4H.

15. The Authors concluded that "loss of either GGA3 or Rabaptin5 mimics the effect of loss of ErbB3 on endocytic trafficking of Integrin β 1, consistent with the hypothesis that GGA3 and Rabaptin5 are effectors of ErbB3 in promoting endosomal recycling and impeding EV release". To confirm this conclusion, the inclusion of siRabaptin5 results in Figures 6H and 6J is suggested.

Response: we thank the reviewer for the comment. We have now included immunoblots of MCF10A cell lysate after silencing ErbB3 or Rabaptin5, as the results shown in the previous Figure 6G. We believe that these new data satisfy the specific request.

16. To be consistent with the results presentation:

- The inclusion of Modal size is recommended in Figure 6I.
- Some graphs show the number of cells or biological replicates while other ones no.
- Figure 4E showed different time points for both siRNAs.

Response: we appreciate the comment and we have now included as the new main Figure 6H the modal size for the EVs secreted by MCF10A cells upon Rabaptin5 silencing. We will ensure that all respective Figure legends indicate the number of replicates. The intermediate time points showed in the main Figure 4E are different, however since the final read out at 9 h using two independent siRNAs against ErbB3 are directly comparable we ask permission to maintain the time points with respect to the analysis we performed.

17. Figure 1E represents the squared regions of Figure 1D, but it is not indicated in the figure legend.

Response: we apologize for this oversight. We have now indicated in Figure 1 legend that Figure 1E represents the squared regions of Figure 1D, as suggested.

18. In the legend of Figure 1D-G, 30min of integrin internalization is reported, where it should be 15min according to main text and methods.

Response: we apologize for this oversight and we thank the reviewer for this comment. We have now indicated the correct time point in Figure 1 legend.

19. The addition of representative images in Figure 6A is recommended, as already present in Figure 11.

Response: we thank the reviewer for the comment. Representative images of Fig. 6A-D were included as the new panel Fig. 6B.

Figure 6B

20. As two different siRNAs for ErbB3 were used and not in all experiments, the employed siRNA should be indicated in each experiment. In the cases where both ErbB3 siRNAs were employed, figures should report them either as main results or supplementary.

Response: we appreciate this meticulous comment. We have now indicated in the figure and in the respective figure legends which siRNA was used in the respective set of experiments (siErbB3 #01 or #02).

21. Why do the Authors use EVs enriched in the VSF or by UC to show the same result? What is the criteria to choose one or the other one? For example, in Figures 6G and 6K.

Response: based on the guidelines suggested by MISEV 2018 and 2023, there is no gold standard method for EV isolation. Thus, by using at least two independent methods (i.e., tangential flow filtration, followed by immuno-affinity and ultracentrifugation; UC) we validate the enrichment of EVs in our sample preparations, showing reproducible results among the different EV enrichment protocols (Figure 3).

Reviewer #3 (Evidence, reproducibility and clarity (Required)):

The paper by Dorival Mendes Rodrigues-Junior et al., focuses on a novel ligand-independent role of ErbB3 receptor, modulating Transferrin receptor and integrin beta1 early recycling. Authors perform several in vitro studies where they show how ErbB3 depletion diverts integrin beta1 from recycling towards lysosomal degradation and extracellular vesicle secretion, impairing cell migration. They also provide mechanistic experiments showing the role of ErbB3 on Arf6-GGA3-Rabaptin5 endosomal complex assembly.

Major comments:

1. Fig. 1. Authors should co-stain with early endosomal markers (such as EEA1) to clearly show endogenous ErbB3 and Beta1 integrin endosomal co-localization. Including some insets with higher magnifications would also improve visual inspection of such interactions.

Response: as requested, we have performed colocalization of ErbB3, traced Integrin β 1 and the recycling endosome marker EHD1, showing triple colocalization in a subset of endosomes, as shown in the new Supplementary Figure S2H. Experimental limitations prevented us from including EEA1 in triple staining for mCherry-ErbB3 or endogenous ErbB3 protein, please note the figures below (for reviewers). Furthermore, ectopically expressed ErbB3 in MCF10A cells did not show convincing co-localisation with EEA1 (see example below). We believe that the new triple colocalization showing ErbB3 and Integrin β 1 in EHD1-positive endosomal compartments satisfies this specific comment.

Supplementary Figure S2

MCF10A (Confocal microscopy) – Revision only

2. Fig. 1H and 1I. Authors need to provide TIRF penetration depth to better evaluate the potential cytosolic contribution. Additionally, plasma membrane purification studies would help to validate their live imaging results.

Response: the TIRF penetration depth was 83nm which has now been added to the methods section. Purifications of plasma membrane fractions, following recycling of traced surface-labelled Integrin $\beta 1$ in control or siErbB3 depleted cells, by cell surface biotinylation and immunoblotting of the recovered proteins is indeed a valuable approach to validate our findings. Nevertheless, we are confident about the results of our confocal imaging results. Thus, including these results might not contribute significantly to the novelty of this manuscript. Hence, we ask permission to publish the paper at this stage, without the plasma membrane purification, as this requires optimizations and will delay the publication of our paper, in addition to exhausting our limited financial resources.

3. Fig. 1J. Authors should explain better how they calculated normalized fluorescence.

Response: the normalized fluorescence is explained in the Fig. 1J legend and in the respective method section. Alexa488 intensity was normalized between 0-1, with the control as reference where $F_{norm} = ((F_{max} - F_{min}) / (F - F_{min}))$. All data points were background corrected, followed by normalization to the pre-stimulatory level (F/F0).

4. Fig. 2B. Authors should include some plasma membrane markers (such as WGA) to better localize cell surface after beta1 integrin tracing.

Response: we appreciate the reviewer's comment, and have attempted the suggested experiment, but in our hands, WGA did not give a clear membrane staining but a diffuse faint signal in MCF10A cells for reasons we do not fully understand.

5. Fig. 1J, 1M-1L: beta1 integrin endocytic recycling should be compared across the same time-points to better evaluate kinetic differences.

Response: the intermediate time points showed in the main Figure 1J, M-L are based on the final read out. We understand that it could be interesting evaluating the kinetic differences but this will generate a substantial number of comparisons that might be difficult for visualization. We ask permission to keep the comparisons among the latest respective time points with respect to the performed analysis.

6. Fig. 3. Author should consider adding additional experiments with Rab4 and Rab11 dominant negative forms to validate their results.

Response: the experiments proposed have been performed, but the ectopic expression of dominant negative Rab4 and Rab11 had detrimental effects to the cells, with the formation of large endosomal blobs and rounding up of the MCF10A cells. Subsequently we do not feel confident with the possible conclusions from these data. We ask the reviewer to understand this technical detail and accept the fact that we are not able to address this point.

7. Fig. 4M. To validate authors' claim on the role of integrin Beta1-containing EVs on invasive behaviour, they should repeat the experiment using blocking beta1 antibodies prior to EV addition.

Response: we thank the reviewer for this comment. As requested, we performed the experiment using the Integrin β 1 blocking monoclonal antibody (mAb; clone P4C10). The new data show that P4C10A treatment alone or in combination with EVs derived from MCF10A cells transfected with siCtrl or siErbB3 significantly reduced invasiveness in comparison to IgG treatments, confirming the mechanistic role of Integrin β 1 promoting MCF10A invasive behaviour. The new Figure 4M was included and further discussed in this manuscript.

Figure 4

8. While authors claim that their results could potentially clarify different aspects of tumour dissemination, most of their experiments are done in MCF10A, a non-tumorigenic epithelial cell line. To better support their conclusion, they should reproduce key experiments in MCF7 or other tumorigenic cell line.

Response: we thank the reviewer for the comment. As explained in response to reviewer 1, the rationale for using different cell-lines or primary cells is now better explained in the manuscript. We found that depletion of ErbB3 impaired recycling of Integrin β 1 in the normal non-malignant cells including MCF10A and primary breast epithelial cells, but not in malignant MCF7 cells that overexpress ErbB3 (data not shown), which is now discussed in the paper. Moreover, MCF7 cells were used to detect endogenous ErbB3 as normal expression levels of ErbB3 (primary MECs and MCF10A) were not detectable by immunofluorescence microscopy with a range of antibodies we tested. Furthermore, we also included in this new version that GGA3 and Rab GTPase-binding effector protein 1 (Rabaptin5 or RABPT5) shared colocalisation with endogenous ErbB3 in MCF7 cells as the new Supplementary Figure 9A, B. Finally, we also attempted to conduct the Arf6 IP in MCF7 cells, but as opposed to MCF10A cells, the yield of Arf6 in pull down experiments was much lower than in MCF10A cells (see figure below – reviewers only), and interacting proteins were not detectable.

Suppl. Figure S7

Suppl. Figure S9

Minor comments:

1. Fig. 1D-1F: please explain better if beta1 integrin surface signal was quenched in these specific set of studies.

Response: Beta1 Integrin was quenched on ice with an antibody against Alexa488 as described by Arjonen et al. (Traffic, 2012; DOI: 10.1111/j.1600-0854.2012.01327.x), and further outlined in the methods section and results section (page 6 and schematic Fig4A).

2. Suppl. Fig. 3A: last WB lane should read "siErB2" instead of "siErbB3".

Response: we thank the reviewer and we apologize for this oversight. We corrected the siErbB2 lane in Supplementary Figure 3A, as requested.

Full Revision

Figure S3

February 18, 2025

Re: JCB manuscript #202501255T

Dorival Rodrigues-Junior
Uppsala University

Dear Dr. Rodrigues-Junior,

Thank you for submitting your manuscript entitled "Loss of ErbB3 redirects Integrin β 1 from early endosomal recycling to secretion in extracellular vesicles" from Review Commons. We appreciate your revisions in response to the reviewers from Review Commons and agree that the advance provided by your study seems interesting for the JCB readership. However, some of the revisions do not currently meet the standards for JCB, therefore the concerns outlined below need to be completely addressed before we can proceed with re-review. Please let us know if you are able to address the major issues outlined below and wish to submit a revised manuscript to JCB. Please note that papers are generally considered through only one revision cycle, so any revised manuscript will likely be either accepted or rejected. If you do not wish to pursue this revision please let us know so that we can inform Review Commons to reopen your submission for consideration at another partner journal.

Reviewer#1 point 1

This indeed is an important point. The authors respond to the criticism by saying they observed "that depletion of ErbB3 impaired recycling of Integrin β 1 in the non-malignant cells, including MCF10A and primary breast epithelial cells, but not in malignant MCF7 cells that overexpress ErbB3 (data not shown)."

This is an important point and distinction between receptor trafficking behaviour of oncogene-overexpressing (cancerous) and normal. These data need to be included in the manuscript demonstrating 1) HER3 expression and activity in the two cell lines 2) comparing integrin recycling between the two in control and ErbB3 depleted conditions.

Reviewer 2 point 3.

The reviewer wonders whether siRNA efficiency scales with effect. The authors reply : "the immunoblots presented here to assess the efficiency of the two different siRNAs are one example and we noted some variability between different experiments but find that both siRNAs work well and yield comparable effects on recycling of Integrin β 1." This is problematic and reflects a general issue throughout the manuscript. Related to this in response to reviewer#1 point 6 the authors provide in many instances just numerical quantifications of band intensities from individual blots. Instead, quantification and statistics from multiple experiments are required throughout. Where are the ErbB3 silencing validations from Figure 1 I-N?

Reviewer2 point 4. Please edit the text and figure labelling to avoid further misunderstanding.

Reviewer 2 point 5. The authors make a lot of claims about lysosomal degradation and the reviewer is correct in requesting experimental evidence for these claims. It is true indeed, that the role of lysosomal degradation has been shown in the earlier studies but these are in different cell types such as HeLa and therefore are not valid comparisons. These experiments should be carried out as requested by the reviewer. In addition, the authors should consider whether another ErbB3 binding protein regulating lysosomal degradation of ErbB2/3 and integrin recycling in breast cancer cells, SORL1/SORLA, would be relevant?

Reviewer 2 point 7. This is a serious issue and silencing efficacy needs to be shown for all experiments (still lacking for MCF10A and non-malignant cells for Fig 1 as mentioned above) with proper quantification

Reviewer 2 point 9 the new S5 figure provided lacks quantification of the bands from 3 or more experiments

Reviewer 2 point 15 The new Rabaptin5 data in Fig 6G has only one independent siRNA for both ErbB3 and Rapaptin 5, is of poor quality and lacks quantification for 3 or more experiments. The overall level of the robustness of the data in this manuscript is problematic and this is yet another example of this that need to be addressed properly with new experimentation.

If you choose to revise and resubmit your manuscript, please also attend to the following editorial points. Please direct any editorial questions to the journal office.

GENERAL GUIDELINES:

Text limits: Character count is < 40,000, not including spaces. Count includes title page, abstract, introduction, results, discussion, and acknowledgments. Count does not include materials and methods, figure legends, references, tables, or supplemental legends.

Figures: Your manuscript may have up to 10 main text figures. To avoid delays in production, figures must be prepared according to the policies outlined in our Instructions to Authors, under Data Presentation,

<https://jcb.rupress.org/site/misc/ifora.xhtml>. All figures in accepted manuscripts will be screened prior to publication.

*****IMPORTANT:** It is JCB policy that if requested, original data images must be made available. Failure to provide original images upon request will result in unavoidable delays in publication. Please ensure that you have access to all original microscopy and blot data images before submitting your revision. *******

Supplemental information: There are strict limits on the allowable amount of supplemental data. Your manuscript may have up to 5 supplemental figures. Up to 10 supplemental videos or flash animations are allowed. A summary of all supplemental material should appear at the end of the Materials and methods section.

Please note that JCB now requires authors to submit Source Data used to generate figures containing gels and Western blots with all revised manuscripts. This Source Data consists of fully uncropped and unprocessed images for each gel/blot displayed in the main and supplemental figures. Since your paper includes cropped gel and/or blot images, please be sure to provide one Source Data file for each figure that contains gels and/or blots along with your revised manuscript files. File names for Source Data figures should be alphanumeric without any spaces or special characters (i.e., SourceDataF#, where F# refers to the associated main figure number or SourceDataFS# for those associated with Supplementary figures). The lanes of the gels/blots should be labeled as they are in the associated figure, the place where cropping was applied should be marked (with a box), and molecular weight/size standards should be labeled wherever possible.

If you choose to resubmit, please include a cover letter addressing the reviewers' comments point by point. Please also highlight all changes in the text of the manuscript.

Regardless of how you choose to proceed, we hope that the comments below will prove constructive as your work progresses. We would be happy to discuss them further once you've had a chance to consider the points raised. You can contact the journal office with any questions at cellbio@rockefeller.edu.

Thank you for thinking of JCB as an appropriate place to publish your work.

Sincerely,

Johanna Ivaska
Monitoring Editor

Andrea L. Marat
Deputy Editor

Journal of Cell Biology

Dear Dr. Ivaska and Dr. Marat,

We are pleased to submit our revised manuscript entitled: "Loss of ErbB3 redirects integrin β 1 from early endosomal recycling to secretion in extracellular vesicles", for publication as a Research Article by the Journal of Cell Biology.

We present a revised version of the manuscript, incorporating data addressing the editors and reviewers' concerns. We have endeavoured to respond to all the queries and hope that the current version is acceptable. In the full-revision file, you can find a point-by-point response to all the concerns.

In summary, this study is of fundamental novelty and high impact as it:

- 1. Reveals an unexpected role of ErbB3 in controlling Integrin β 1 trafficking and thus epithelial cell motility and extracellular vesicle secretion.*
- 2. Uncovers the first ligand-independent, non-canonical cellular function for ErbB3 as a scaffold for the Arf6-Rabaptin5-GGA3 endosomal sorting complex.*
- 3. Provokes the notion that pseudo-RTKs may have evolved cellular functions beyond receptor signalling, such as by scaffolding endosomal sorting compartments.*
- 4. Uncovers a novel aspect of protein trafficking, highlighting not only its routing to recycling or lysosomal degradation, but also its secretion via extracellular vesicles. These findings offer fresh perspectives by bridging the EV and protein recycling fields. In current literature, these pathways are typically illustrated in isolation, either one or the other, but rarely as interconnected processes.*

We hope that you share our view that these conceptually ground breaking findings will be of interest to a broad cross-disciplinary audience interested in cell signalling, cancer biology, endocytic trafficking and integrin biology.

We hope you will consider this submission favourably and look forward to hearing from you.

With best regards,

Ingvar Ferby, PhD

Dorival Mendes Rodrigues-Junior, PhD

Dear Dr. Rodrigues-Junior,

Thank you for submitting your manuscript entitled "Loss of ErbB3 redirects Integrin β 1 from early endosomal recycling to secretion in extracellular vesicles" from Review Commons. We appreciate your revisions in response to the reviewers from Review Commons and agree that the advance provided by your study seems interesting for the JCB readership. However, some of the revisions do not currently meet the standards for JCB, therefore the concerns outlined below need to be completely addressed before we can proceed with re-review. Please let us know if you are able to address the major issues outlined below and wish to submit a revised manuscript to JCB. Please note that papers are generally considered through only one revision cycle, so any revised manuscript will likely be either accepted or rejected. If you do not wish to pursue this revision please let us know so that we can inform Review Commons to reopen your submission for consideration at another partner journal.

1. Editors: Reviewer#1 point 1

This indeed is an important point. The authors respond to the criticism by saying they observed "that depletion of ErbB3 impaired recycling of Integrin β 1 in the non-malignant cells, including MCF10A and primary breast epithelial cells, but not in malignant MCF7 cells that overexpress ErbB3 (data not shown)". This is an important point and distinction between receptor trafficking behaviour of oncogene-overexpressing (cancerous) and normal. These data need to be included in the manuscript demonstrating 1) HER3 expression and activity in the two cell lines 2) comparing integrin recycling between the two in control and ErbB3 depleted conditions.

Response: we thank the editors for these valuable comments. To address these suggestions, we performed RT-qPCR to assess ErbB3 mRNA levels in the cell lines used in this study. As expected, based on the TCGA datasets - Supplementary Figure S3A (previous Supplementary Figure 1A), our new data show that the breast cancer cell line, MCF7, expresses significantly higher levels of ErbB3 mRNA in comparison to the non-tumorigenic cells (New Supplementary Figure S1C). Furthermore, we now also include data assessing Integrin β 1 recycling in MCF7 cells (New Supplementary Figures 3D, E). We found that Integrin β 1 internalisation was less effective in MCF7 cells as compared to MCF10A and prHMEC with limited detectable return of internalised Integrin β 1 to the cell surface of both control or ErbB3-depleted cells, precluding reliable assessment of Integrin β 1 recycling in these cells (Supplementary Figures S3D, E). As the editor previously indicated, Integrin trafficking is regulated in multiple context-dependent ways (a topic covered in the Moreno-Layseca et al. Nat Cell Biol, 2019). Our observation may reflect differing Integrin β 1 trafficking dynamics and underlying mechanisms between malignant and normal breast epithelial cells. It would be interesting to address if this hypothesis holds true and in such case by what

mechanism/s, but that we believe falls that are beyond the scope of this work. We have opted to include this data (Supplementary Fig S3D, E) and discuss the implications along the lines outlined here, also in the manuscript.

New Supplementary Figure S1C

New Supplementary Figures S3D, E

2. Editors: Reviewer 2 point 3.

The reviewer wonders whether siRNA efficiency scales with effect. The authors reply: "the immunoblots presented here to assess the efficiency of the two different siRNAs are one example and we noted some variability between different experiments but find that both siRNAs work well and yield comparable effects on recycling of Integrin β 1." This is problematic and reflects a general issue throughout the manuscript. Related to this and in response to reviewer#1 point 6, the authors provide in many instances just numerical quantifications of band intensities from individual blots. Instead, quantification and statistics from multiple experiments are required throughout. Where are the ErbB3 silencing validations from Figure 1 I-N?

Response: we thank the editors for this comment. We have now included the knockdown confirmation related to main Figure 1B-E as the new Figure 1A alongside the quantification and statistics from three biological replicates. Additionally, we present, as the new Supplementary Figures S1E, the quantification and statistics from three biological replicates of ErbB3 silencing in MCF10A cells, related to main Figure 2F, G (previous Figure 1I, J). This confirms the similar knockdown efficiency of the two independent siRNAs targeting ErbB3. We have also included the ErbB3 silencing data in prHMEC cells, as the Supplementary Figure S1G, related to main Figure 2H, I (previous Figure 1K, L). The silencing of EGFR and ErbB2 is shown in Supplementary Figure S2A. The new Supplementary Figure S3F demonstrates the efficiency of ErbB3 silencing in MCF7 cells by RT qPCR, related to Main Figure 2M, N (previous Main Figure 1B, C). We also present, as the new Figure 6G and Supplementary Figure S5L, related to Figure 6J, the quantification and statistics from three biological replicates of the indicated proteins upon ErbB3, Rabaptin5 or GGA3 silencing in MCF10A cells. In addition to these figures, throughout our manuscript several immunoblots confirm gene silencing (Figures 3G, L; Figure 5B; Suppl. Fig. S2B; Suppl. Fig. S4E, G; Suppl. Fig. S5A, C, G, I). To maintain clarity of complex figures, we have adopted the practice of presenting average band intensities from three independent biological replicates, as indicated in the respective figure legend, rather than displaying the more detailed quantification diagrams, which would result in an excessive number of graphs. We kindly ask for the editor's understanding of this rationale and we believe that the new panels address this important point sufficiently.

New Figure 1A

New Supplementary Figure S2E, G

New Supplementary Figure S3F

New Figure 6G

New Supplementary Figure S5L, related to Figure 6J

3. Editors: Reviewer2 point 4. Please edit the text and figure labelling to avoid further misunderstanding.

Response: we have now edited the main text related to the main Figure 4B, C to avoid further misunderstanding.

4. Editors: Reviewer 2 point 5. The authors make a lot of claims about lysosomal degradation and the reviewer is correct in requesting experimental evidence for these claims. It is true indeed, that the role of lysosomal degradation has been shown in the earlier studies but these are in different cell types such as Hela and therefore are not valid comparisons. These experiments should be carried out as requested by the reviewer. In addition, the authors should consider whether another ErbB3 binding protein regulating lysosomal degradation of ErbB2/3 and integrin recycling in breast cancer cells, SORL1/SORLA, would be relevant?

Response: we appreciate the editor's comment and since the more novel finding of our work is the re-routing of Integrin β 1 towards secretion in EVs, we have opted to put emphasis on strengthening the EV data, and milder reference to the lysosomal degradation. However, since the original comment by the reviewer originated from our Discussion, where we aimed at providing a more rounded presentation of the findings in relation to the broader, relevant literature, we find this comment rather excessive and ask the editor to accept that these experiments are not performed at this stage. Furthermore, the possibility that the trafficking dynamics changes in cancer is very interesting and worth investigating. We agree that additional players might affect ErbB3 stability, such as SorLA, which attenuates ErbB3 lysosomal degradation by retaining ErbB3 within Rab4-positive recycling endosomes. We have now discussed these findings accordingly, including the reference to Al-Akhrass et al.

Oncogene, 2021. We hope that the editors will accept that we politely refrain from reproducing these data.

5. Editors: Reviewer 2 point 7. This is a serious issue and silencing efficacy needs to be shown for all experiments (still lacking for McF10A and non-malignant cells for Fig 1 as mentioned above) with proper quantification.

Response: we agree with the comment and we indicate that this comment has been addressed above, in response to Editors' comment #2.

6. Editors: Reviewer 2 point 9 the new S5 figure provided lacks quantification of the bands from 3 or more experiments.

Response: As also addressed above in response to Editors' comment #2, we now provide the average densitometry of 3 independent biological replicates in the new Supplementary Figure S1B (previous Supplementary Figure S5E).

7. Editors: Reviewer 2 point 15, The new Rabaptin5 data in Fig 6G has only one independent siRNA for both ErbB3 and Rapaptin5, is of poor quality and lacks quantification for 3 or more experiments. The overall level of the robustness of the data in this manuscript is problematic and this is yet another example of this that need to be addressed properly with new experimentation.

Response: We now provide data quantification of knock-down efficiency related to Figure 6G, in addition to experiments throughout the manuscript as detailed in response to Editor's comment #2. Notably, ErbB3 silencing was consistently effective with both siRNA #1 and #2, both of which are used for key experiments throughout our work, for a few experiments only the independent siRNA (#2) was used and validated through a rescue of the integrin trafficking phenotype by a non-siRNA targeted ectopically expressed ErbB3. We also now provide quantification from multiple repeats of knock-down efficiency of GGA3 (Supplementary Figure S5L) and Rabaptin5 (Figure 6G). Although only one independent siRNA was used for Rabaptin5, it is important to note that the results in MCF10A cells were also reproducible in MCF7 cells (Supplementary Figure S5I-K).

New Figure 6G

Additional comments:

- To enhance the clarity and logical progression of our manuscript, we have revised the order of Figures 1 and 2, as well as rearranged several panels within these figures. These changes also required updates to the corresponding Supplementary Figures. In the revised workflow, we begin by presenting the impact of ErbB3 on cell migration and proliferation (Figures 1A–E), followed by its role in the recycling of Integrin β 1 and the Transferrin receptor (Figures 1F–G and Figure 2). We believe this new structure offers a more coherent and intuitive narrative for readers. We ask for the editors and reviewer’s understanding of our rationale.*
- We provided a validation of the Figure 5G data using an independent approach. We found that the ectopically expressed LL866/7AA mutant form of mCherry-tagged ErbB3 co-localised with fewer internalised Integrin β 1-positive structures compared to the wild-type (WT) receptor (new Figure 5H).*

Figure 5H

- We have included a new schematic figure summarizing the findings of this work as the new Figure 7.*

Figure 7

GENERAL GUIDELINES:

Text limits: *The character count in our manuscript is < 40,000 (not including spaces).*

Figures: *Our manuscript has 7 main figures.*

Supplemental information: *We have adjusted our manuscript accordingly. As requested, we have reduced from nine to five Supplementary Figures.*

Point-by-point description of the revisions

We are grateful for the overall positive feedback and constructive suggestions. We have been able to experimentally address several of the suggested points and provide here a revision plan addressing all the reviewers' additional concerns.

Reviewer #1 (Evidence, reproducibility and clarity (Required)):

ErbB3 is well-known for its significance in cancer, which is dependent on ligand-binding and heterodimerization with other ErbB family members. In the current work, Rodrigues-Junior et al. identified novel, unexpected functions of ErbB3 in promoting early endocytic recycling and restricting exocytic trafficking (extracellular vesicles secretion) of membrane receptors, such as integrin β 1 and transferrin receptor, via stabilizing the Arf6-GGA3-Rabaptin5 endosomal sorting complex. Via ErbB3 siRNA knockdown, they observed an impaired recycling of transferrin receptor and integrin β 1 back to the cell membrane. The recycling assay condition (growth factor-deprived) provided a very clean result to support that this ErbB3-dependent endocytic trafficking is ligand-binding independent. The trafficking-dependence on ErbB3 (both the endocytic and the exocytic) was further supported by integrin β 1 functional assays (scratch closure assay and Matrigel invasion assay). There are still some details that need to be clarified to fully understand the conclusion.

Major points:

1. The manuscript started with a pathological correlation between high ErbB3 level and poor patient survival rate. In Fig.1, the impaired TfR recycling, and the co-localization between ErbB3 and integrin β 1 were also performed in the pathological breast cancer cell line, MCF7. While investigating integrin β 1 recycling, the authors suddenly switched to another two non-malignant human breast epithelial cell lines, which led to a difficult correlation of ErbB3-mediated recycling back to the disease situation. The authors should state more clearly this point, rather than data not shown. This inconsistency occurred also in other assays, for example, when addressing the trafficking from TGN to cell surface, MCF7 was utilized; while when addressing extracellular vesicle secretion, MCF10A was utilized.

Response: we thank the reviewer for the comment. The rationale for using different cell lines or primary cells is now better explained in the manuscript along the line outlined below, and supported by new data addressing the effect of ErbB3-depletion on recycling of integrin β 1 (new Supplementary Figure S3F). While we found that depletion of ErbB3 impaired recycling of Integrin β 1 in the non-malignant cells, including MCF10A and primary breast epithelial HMEC cells, internalisation of Integrin β 1 was less effective in malignant MCF7 cells that overexpress ErbB3 (new Supplementary Figures S1C and S3D, E), with limited detectable return of internalised Integrin β 1 to the cell surface of both control or ErbB3-depleted MCF7 cells. Furthermore, MCF7 cells were used to detect endogenous ErbB3 as normal expression levels of ErbB3 in primary HMEC and MCF10A were not detectable by immunofluorescence microscopy with a range of antibodies that we tested. Furthermore, we now provide a direct comparison of ErbB3 mRNA levels between the three cell lines used in our study (new Supplementary Figure S1C), and discuss the discrepancy between normal/non-malignant cells and cancer cells, that raises the interesting question whether the mechanism and dynamics by which Integrin β 1 traffic is affected during cancer progression. With regard to the transferrin recycling assay, we first attempted to use MCF10A cells for consistency. However, we found that transferrin internalized poorly in these cells and the limited pool of transferrin that internalised was retained in these cells for an extended time (3 h), thus preventing their

use. We have also re-arranged the data, in a manner that brings the main findings of the study better into focus; ErbB3-dependent trafficking of Integrin $\beta 1$ now comes before the Transferrin data, the latter which we see primarily as an attempt to assess the putative general nature of our findings. Furthermore, the analysis of ErbB3 expression in breast cancer does not constitute a central element of our work (and should not have started the manuscript) and has thus been moved to supplementary data (new Supplementary Figures S3A-C) and presented in the context of the new analysis of the effect of ErbB3 on integrin recycling in malignant MCF7 (Supplementary Figures S3D, E)

New Supplementary Figure S1C

New Supplementary Figures S3D, E

Concerning the data on trafficking from the TGN to cell surface we mistakenly wrote that they were performed in MCF7 cells although they were in fact done in MCF10A cells. This is now corrected in the new version of this manuscript.

Additionally, based on the constructive comment by this reviewer, we have now extended the analysis of EV secretion in ErbB3, Rab4 and Rabaptin5 silenced cells to MCF7 cells. Interestingly, the new data is in line with our findings in MCF10A and prHMEC cells, that absence of ErbB3 significantly increased EV secretion. Moreover, Rab4 and Rabaptin5 knockdown also enhanced the amount of EVs secreted by MCF7 cells. These results were incorporated in the manuscript as new Supplementary Figures S4G-I and new Supplementary Figures S5I-K, as recommended. Furthermore, we also included in this new version the data showing that GGA3 and to a lesser extent Rab GTPase-binding effector protein 1 (Rabaptin5 or RABPT5), shared colocalisation with endogenous ErbB3 in MCF7 cells as the new Supplementary Figures S5D, E. Finally, we also attempted to conduct the Arf6 IP in MCF7

cells, but as opposed to MCF10A cells, the yield of Arf6 in pull down experiments was much lower than in MCF10A cells (see figure below – reviewers only), and interacting proteins were not detectable.

New Supplementary Figure S4G-I

New Supplementary Figure S5I-K

Supplementary Figure S5D, E

MCF7 protein extract: Arf6 IP (Reviewers only)

2. It was shown before that ErbB3 undergoes constitutive internalization and degradation within several hours that is independent of ligand-binding (ref#13). Can the authors provide experimental evidences to show the correlation of TfR or integrin b1 recycling with this dynamic ErbB3 levels rather than ErbB3 knockdown?

Response: Given that steady-state levels of ErbB3 seem stable, we are somewhat unsure what approach the reviewer has in mind. We reason that the best answer may be to validate that steady state ErbB3 co-exist with Integrin β 1 in the recycling compartment at the endogenous level. To this end, we have now performed immunofluorescence on ErbB3, traced Integrin β 1 and the recycling endosome marker EHD1, showing triple colocalization at the endogenous level in a subset of endosomes, as shown in the new Supplementary Figure S2D. We hope that the new EHD1 triple colocalization showing ErbB3 and Integrin β 1 in endosomal compartments at the endogenous level satisfies this specific comment.

New Supplementary Figure S1D

3. The efficiency of siRNA knockdown of ErbB3 (both #1 and #2) should support the observed phenotype (Fig. 1I-J, K-L). Is there a correlation between the ErbB3 level with integrin recycling? For example, siRNA#2 led to more efficient knockdown of ErbB3 in MCF10A?

Response: notably, the immunoblots presented here to assess the efficiency of the two different siRNAs are one example but find that both siRNAs work well and yield comparable effects on recycling of Integrin β 1. Importantly, the recycling data represents biological repeats of independently performed experiments and have yielded reproducible and consistent ErbB3 silencing using both siRNAs. This is noted by the lack of significance between ErbB3 knocked down cells in Figures 2F-I (previous Fig. 1I-J and K-L). Hence, we consider that both siRNAs against ErbB3 worked efficiently with comparable outcome and the variability detected by immunoblotting did not seem correlate with variability in the phenotype. Please also note our reply to Editors' comment #2, with multiple new quantifications of knockdown efficiencies performed.

4. ErbB3 loss led to more extracellular vesicles secretion, but also lysosomal degradation of integrin b1. This conclusion is supported by results shown in Fig.4D-E and Fig. S8A-B, while the analysis from the same cell line (MCF10A, Fig. S3A) results in no change of integrin b1 levels upon ErbB3 depletion. Fig. S3B showed also no change in a second non-malignant cell line (prHMEC). How do the authors explain this conflict?

Response: we thank the reviewer for this comment. We believe that the increase in EV secretion and lysosomal degradation is compensated by increase in de novo synthesis of Integrin $\beta 1$ (see data below, from Fig. S2C). In the original manuscript we did not perform the appropriate statistical analysis of the RT-qPCR data. The applied unpaired two-tail Student's T-test is only suitable for normally distributed samples, which is not the case here. Instead, we performed the appropriate Mann-Whitney U-test assuming non-normal distribution, yielding an exact p-value of 0.017 (Supplementary Figures S2B, C – previous Supplementary Figures S3B, C).

Suppl. Figure S2C

Minor points:

1. Is TfR also colocalizing with endogenous ErbB3?

Response: we have attempted the suggested experiment in MCF7 cells, but initial attempts failed. Further optimisation and possible addition of the endocytic recycling inhibitor primaquine might be required, but we have opted to instead focus on strengthening the key Integrin $\beta 1$ trafficking data. The TfR recycling assay was performed to address whether ErbB3 could play a broader trafficking role beyond Integrin $\beta 1$. To make this clearer, we have re-arranged the beginning of the manuscript so that the integrin $\beta 1$ data comes before the TfR recycling data.

2. Fig. 3J, TSG101, T is masked by 3I

Response: we apologize for this oversight. We have gone through the manuscript in detail and corrected all pointed errors accordingly.

3. Page 10, the description of the EV secretion in prHMEC cells is annotated to the wrong figure. Fig S5D \diamond S7D; S5E \diamond S7E

Response: we apologize for this oversight and have now corrected the mistake.

4. Fig. 4M: How was the motility/invasion into Matrigel determined? Images? Only quantifications are shown.

Response: the matrigel invasion assay was described in the Material and Methods section. Accordingly, the data were expressed as the percentage of invasion based on the ratio of the

mean number of cells invading through matrigel matrix per mean number of cells in the uncoated support. In this letter, the reviewer can find representative micrographs of invaded MCF10A siCtrl non-treated (Ctrl) or treated with VSF secreted from MCF10A siCtrl or siErbB3. Since this is an established method to measure cell invasion, we hope that the reviewer agrees that these images do not add value to the manuscript. Please see figure below (for reviewers).

(Revision only: the visible spots are considered invading cells)

5. Fig. 4M: Exosomes collected from ErbB3-depleted cells promotes the migration in MCF10A-wild type cells, how about the effects on ErbB3-depleted cells? This group should be included for analysis.

Response: as proposed, we have treated both control and ErbB3-silenced MCF10A cells with normalized concentrations of EVs secreted from siCtrl and siErbB3 (1×10^9 nanoparticles/mL) for 48 hours, followed by cell viability and cell invasion assays. The new data show that both EV pools modestly increased cell viability and substantially increased invasiveness of both wild-type and ErbB3-depleted cells through Matrigel (new Figures 4K and L). Together, our results indicate that while ErbB3-silenced MCF10A cells exhibited lower basal motility, ErbB3 is not required for the observed EV induced motility.

Figure 4

6. Quantification of the blots should be provided for Fig. 5A (GGA3), 5B (GGA3, Rabaptin5 and Arf6), 5F (GGA3) and 5G (GGA3, Rabaptin5 and Arf6). What is mock IP in each graph? The mock IP is neither mentioned in methods nor in legends.

Response: we have now carried out densitometry analysis in all the requested immunoblots in Figure 5. We also changed the mock IP term to IgG IP for clarity. The use of non-immunogenic IgG in control IPs is now specified in the methods and respective figure legend.

Reviewer #2 (Evidence, reproducibility and clarity (Required)):

Summary:

In their manuscript, Rodrigues-Junior and colleagues identify a novel ligand-independent function of the tyrosine kinase receptor (RTK) ErbB3 as a regulator of integrin $\beta 1$ recycling. In particular, the authors demonstrate that ErbB3 depletion reduce $\beta 1$ integrin surface expression, triggering its lysosomal degradation and increasing its secretion in extracellular vesicles (EVs). Moreover, the authors show that these EVs enhanced the invasive capacity of ErbB3 wild type breast epithelial cells. In addition, the authors evidence the interaction between ErbB3, GGA3 and Rabaptin5. Loss of any of these proteins destabilizes this interaction, which abrogates integrin $\beta 1$ recycling and leads to its degradation and secretion. The work is potentially interesting; however, there are some aspects that need to be analyzed in a more robust manner.

Major comments:

1. The manuscript is mainly focused on $\beta 1$ integrin endocytic and post-endocytic fate following ErbB3 silencing, describing also a molecular mechanism underlying these observations. Despite the cited manuscript by Deneka, A. and colleagues indicates a similar mechanism for transferrin receptor (TfR) recycling, the Authors only studied the receptor internalization upon ErbB3 silencing. Therefore, this observation does not add any significance to the main topic of the manuscript and its removal should be considered.

Response: we agree with the reviewer that the fate of Integrin $\beta 1$ is the main focus of this manuscript. Although TfR trafficking arguably is of limited biological interest we favour retaining the TfR data as it implies a wider role of ErbB3, beyond the endocytic trafficking of Integrin $\beta 1$. We have now altered the order and description so that the key Integrin recycling data comes before the TfR data. Please let us clarify that the TfR recycling assay does not monitor endocytosis since we trace the fate of the internalised pool of fluorophore-conjugated transferrin. Any conjugated transferrin that is present, or returns to, the plasma membrane is displaced by the large excess of unlabelled holo-transferrin added to the culture media.

2. Data from Figure S1A seems to be not normally distributed. Have the Authors tested the data for normal distribution? If not, please consider it. If the data is not normally distributed, a non parametric Mann-Whitney U-Test would be more suitable.

Response: we thank the reviewer for the comment. The differential ErbB3 mRNA expression analysis was retrieved from the widely used GEPIA2 portal (to date about 600 manuscripts cite this portal on PubMed; reference shown below), based on the selected datasets ("TCGA tumors vs TCGA normal + GTEx normal" or "TCGA tumors vs TCGA normal"). The method for differential analysis is one-way ANOVA, using disease state (Tumor or Normal) as variable for calculating differential expression, as it considers differential expression among several tumors.

Reference 72: Tang, Z., Kang, B., Li, C., Chen, T., and Zhang, Z. (2019). GEPIA2: an enhanced web server for large-scale expression profiling and interactive analysis. Nucleic Acids Res. 47, W556–W560. <https://doi.org/10.1093/nar/gkz430>.

3. The Authors studied the colocalization of ErbB3, Rab4 and Rab11, observing an increased colocalization between ErbB3 and Rab4 10 minutes following primaquine. However, the Authors previously referred to Sönnichsen, B et al. manuscript, in which TfR colocalized with Rab11 at 30min. It would be interesting to see whether ErbB3 and Rab11 colocalize at later

time points in the presence or absence of primaquine. This will reinforce the conclusion that ErbB3 is involved in early Rab4-dependent recycling.

Response: we appreciate the reviewer's comment and agree that it would be interesting and relevant. We have attempted to extend the time of PQ treatment to 30 min to an hour, but we found that at least MCF7 and MCF10A cells exhibit adverse effects such as rounding-up and cell detachment already after 30 min of exposure to the drug. Furthermore, following the editors comment #4, we also discussed that additional players might affect ErbB3 stability, such as SorLA, which attenuates ErbB3 lysosomal degradation by retaining ErbB3 within Rab4-positive recycling endosomes (Al-Akhrass et al. Oncogene, 2021).

4. In Figure 4C the Authors observed a reduction in $\beta 1$ integrin levels in ErbB3 silenced cells compared to the control already at the beginning of tracing (0 min), which might be due to accelerated turnover at the internalization step of their experimental design. To confirm this, immunofluorescence of $\beta 1$ integrin in control and ErbB3 silenced cells could be performed just right after the 15min integrin internalization.

Response: this is likely a misunderstanding as the timepoint (0 min) is defined as the point after the 15 min internalization step when the imaging-based tracing begins, which aligns perfectly with the reviewer's request. We have now edited the text related to the main Figure 4A-C to avoid further misunderstanding.

5. In the discussion, the Authors indicate that "loss of ErbB3 redirects Integrin $\beta 1$ towards lysosomes for degradation, mimicking loss of GGA3 that similarly redirects both Integrin $\beta 1$ and c-Met towards lysosomal degradation, or Rabaptin5 depletion that we find similarly redirects trafficking of internalised Integrin $\beta 1$ towards lysosomal degradation". However, the involvement of lysosomal degradation was only studied for ErbB3 silencing by employing chloroquine. To further support this statement, the use of chloroquine in Rabaptin5- and GGA3-depleted cells is recommended.

Response: we appreciate the reviewer's comment. Since the more novel finding of our work is the re-routing of Integrin $\beta 1$ towards secretion in EVs, rather than lysosomal degradation, we have opted to put emphasis on strengthening the EV data. We now added the note that the published work describing that loss of GGA3 re-routes Integrin $\beta 1$ towards the lysosomes was performed in a different cell type (e.g., squamous cell carcinoma: A431). The possibility that the trafficking dynamics changes in cancer is very interesting and worth investigating (as mentioned above). However, this would constitute a major endeavour that we believe fall outside the scope of this work.

Minor comments:

6. The Authors should consider shortening the following sentences from the Introduction: "GGA proteins contain several functional domains that...thereby regulating sorting of cargo including Integrin $\beta 3$ and TfR into recycling endosomes".

Response: we thank the reviewer for the comment. We have now divided this sentence into two for smoother reading.

7. The Authors do not show ErbB3 silencing efficiency at the protein level until Figure 3G, which should have been shown in Figure 1 or Supplementary Figure 1, as all the research is based on it. Moreover, GGA3 silencing efficiency was never tested.

Response: we thank the reviewer for this comment. We have included, throughout our manuscript, validations for gene silencing (see below). Please also check our answer to the

editor's comment #02. Please also refer to Figures 3G, L; Figure 5B; Suppl. Fig. S2B; Suppl. Fig. S4E, G; Suppl. Fig. S5A, C, G, I.

New Figure 1A

New Supplementary Figures 1E, G

New Supplementary Figure S3F

New Figure 6G

New Supplementary Figure S5L.

8. Figure 1I and Figure 1K may include the representative images for the missing siErbB3 to properly illustrate the associated quantifications.

Response: we thank the reviewer for the comment. We have now included the representative images, as suggested.

New Figure 2F

New Figure 2H

9. Consider including a Western blot showing the effect of lapatinib in EGFR, ErbB2 and ErbB3 protein expression, including their phosphorylated forms.

Response: we thank the reviewer for the comment. As requested, we now show that at the used concentration of 1 μM, lapatinib efficiently blocked tyrosine phosphorylation of ErbB3 and ERK1/2, without perturbing EGFR or ErbB3 expression levels (Supplementary Figures S1B). We have also considered it relevant to show that 1 μM lapatinib used was not cytotoxic to MCF10A and MCF7 cells (Supplementary Figures S1A). We believe that these new results satisfy this specific request.

Supplementary Figures S1A, B

10. Some supplementary figures are mislabelled, such as Supplementary Figure S5D and S5E on page 10, which should be S7D and S7E, respectively. Supplementary Figure S7C on page 15 should be S9C.

Response: we apologize for this oversight and have performed the corrections.

11. The following sentence on page 8 should be revised as a verb is missing: "which corresponds to the reported peak time when colocalization of Rab4 with traced TfR, preceding Rab11 and TfR colocalization that peaks later at 30 minutes".

Response: we apologize for this oversight. It now reads: "which corresponds to the reported peak time of colocalization of Rab4 with traced TfR, which precedes Rab11 and TfR colocalization that peaks later at 30 minutes".

12. The main text indicates that the amount of VSV-G transported to the cell surface after 30min it is not affected by ErbB3 silencing. However, in Figure 3E seems to slightly decrease following the silencing. The Authors may consider employing another Western blot image to match the main text and the quantification in Figure 3F.

Response: as the reviewer noted the immunoblot showed a slight decrease. It is however a very modest decrease that is also observed in the positive control (MUC1) in the same Streptavidin IP sample.

13. In the main text, a significant difference in the nanoparticles/cell between ErbB3-depleted cells and wild type or control cells were reported. However, Figure 3I only showed the statistics of each siRNA vs the control and not the wild type condition.

Response: we apologize for this oversight. We have corrected this and removed from the text the comparison with the wild-type non-transfected cells.

14. The Authors concluded that "chloroquine treatment significantly restored traced Integrin $\beta 1$ levels". However, this conclusion is not reflected in the statistical analysis reported in Figure 4H, which only showed the differences between control and ErbB3 silenced cells. Thus, the statistics reported for the chloroquine results should be added.

Response: we appreciate the comment by the reviewer. The requested comparison is now included in the new Figure 4H.

15. The Authors concluded that "loss of either GGA3 or Rabaptin5 mimics the effect of loss of ErbB3 on endocytic trafficking of Integrin $\beta 1$, consistent with the hypothesis that GGA3 and Rabaptin5 are effectors of ErbB3 in promoting endosomal recycling and impeding EV release". To confirm this conclusion, the inclusion of siRabaptin5 results in Figures 6H and 6J is suggested.

Response: we thank the reviewer for the comment. We have now included quantified immunoblots of MCF10A cell lysate after silencing ErbB3 or Rabaptin5 (Figure 6G). We believe that these new data satisfy the specific request.

New Figure 6G

16. To be consistent with the results presentation:

- The inclusion of Modal size is recommended in Figure 6I.
- Some graphs show the number of cells or biological replicates while other ones no.
- Figure 4E showed different time points for both siRNAs.

Response: we appreciate the comment, and we have now included as the new main Figure 6H the modal size for the EVs secreted by MCF10A cells upon Rabaptin5 silencing. We have also ensured that all respective Figure legends indicate the number of replicates. The intermediate time points showed in the main Figure 4E are different, however since the final read outs at 9 hours using two independent siRNAs against ErbB3 are directly comparable we ask permission to maintain the time points with respect to the analysis we performed.

Figure 6H

17. Figure 1E represents the squared regions of Figure 1D, but it is not indicated in the figure legend.

Response: we apologize for this oversight. We have now indicated in Figure 2 (previous Figure 1) legend that Figure 2B (previous Figure 1E) represents the squared regions of Figure 2A (previous Figure 1D), as suggested.

18. In the legend of Figure 1D-G, 30min of integrin internalization is reported, where it should be 15min according to main text and methods.

Response: we apologize for this oversight, and we thank the reviewer for this comment. We have now indicated the correct time point in the Figure 1 legend.

19. The addition of representative images in Figure 6A is recommended, as already present in Figure 11.

Response: we thank the reviewer for the comment. Representative images of Fig. 6A-D were included as the new panel Fig. 6B.

Figure 6B

20. As two different siRNAs for ErbB3 were used and not in all experiments, the employed siRNA should be indicated in each experiment. In the cases where both ErbB3 siRNAs were employed, figures should report them either as main results or supplementary.

Response: we appreciate this meticulous comment. We have now indicated in the figure and in the respective figure legends which siRNA was used in the respective set of experiments (siErbB3 #1 or #2).

21. Why do the Authors use EVs enriched in the VSF or by UC to show the same result? What is the criteria to choose one or the other one? For example, in Figures 6G and 6K.

Response: based on the guidelines suggested by MISEV 2018 and 2023, there is no gold standard method for EV isolation and multiple methods should be employed. Thus, by using at least two independent methods (i.e., tangential flow filtration, followed by immuno-affinity and ultracentrifugation; UC) we validate the enrichment of EVs in our sample preparations, showing reproducible results among the different EV enrichment protocols (Figure 3).

Reviewer #3 (Evidence, reproducibility and clarity (Required)):

The paper by Dorival Mendes Rodrigues-Junior et al., focuses on a novel ligand-independent role of ErbB3 receptor, modulating Transferrin receptor and integrin beta1 early recycling. Authors perform several in vitro studies where they show how ErbB3 depletion diverts integrin beta1 from recycling towards lysosomal degradation and extracellular vesicle secretion, impairing cell migration. They also provide mechanistic experiments showing the role of ErbB3 on Arf6-GGA3-Rabaptin5 endosomal complex assembly.

Major comments:

1. Fig. 1. Authors should co-stain with early endosomal markers (such as EEA1) to clearly show endogenous ErbB3 and Beta1 integrin endosomal co-localization. Including some insets with higher magnifications would also improve visual inspection of such interactions.

Response: As also mentioned in response to Reviewer 1 major comment #2. As requested, we have now performed immunofluorescence imaging of ErbB3, traced Integrin β 1 and the recycling endosome marker EHD1, showing triple colocalization in a subset of endosomes, as shown in the new Supplementary Figure S2D. Ectopically expressed ErbB3 in MCF10A cells

did not show convincing co-localisation (see example below). Since the focus of our work is on the recycling route rather than endocytosis, that does not seem influenced by absence of ErbB3, we believe that the new triple colocalization showing ErbB3 and Integrin $\beta 1$ in EHD1-positive endosomal compartments is encouraging and may satisfy this specific comment.

Supplementary Figure S1D

MCF10A (Confocal microscopy) – Revision only

2. Fig. 1H and 1I. Authors need to provide TIRF penetration depth to better evaluate the potential cytosolic contribution. Additionally, plasma membrane purification studies would help to validate their live imaging results.

Response: We now provide the TIRF depth in the Methods, which was 83 nm. To isolate plasma membrane could be a valuable validation of the recycling data. However, to the best of our knowledge, this has not previously been done to monitor recycling of Integrins, and we have opted to refrain from attempting to establish the membrane purification approach. Let us mention that we are confident in the imaging based approach that is well-established in the field.

3. Fig. 1J. Authors should explain better how they calculated normalized fluorescence.

Response: the normalized fluorescence is explained in the Fig. 2G (previous Figure 1J) legend and in the respective method section. Alexa488 intensity was normalized between 0-1, with the control as reference where $F_{norm} = (F_{max} - F_{min}) / (F - F_{min})$. All data points were corrected, followed by normalization to the pre-stimulatory level (F/F_0).

4. Fig. 2B. Authors should include some plasma membrane markers (such as WGA) to better localize cell surface after beta1 integrin tracing.

Response: we appreciate the reviewer's comment, and have attempted using WGA that did not give a clear membrane staining but a diffuse faint signal in MCF10A cells for reasons we do not understand, instead we visualised actin in earlier IF experiments, (e.g. Fig 1G) but deemed it unnecessary, as in our experience, low level integrin $\beta 1$ immunofluorescence always enabled identification of cell borders.

5. Fig. 1J, 1M-1L: beta1 integrin endocytic recycling should be compared across the same time-points to better evaluate kinetic differences.

Response: We believe the kinetics is directly comparable as the images are taken at the same 5 minutes intervals for all TIRF experiments and only differ in endpoints that range from 25 to 40 minutes, (Main Figures 2G, I-K) (previous Figures 1J, L-N).

6. Fig. 3. Author should consider adding additional experiments with Rab4 and Rab11 dominant negative forms to validate their results.

Response: the experiments proposed have been attempted, but the ectopic expression of dominant negative Rab4 and Rab11 had detrimental effects on the breast epithelial cells, with the formation of large endosomal blobs and rounding up of the MCF10A cells. Subsequently we do not feel confident drawing conclusions from these data based on potential artifacts. We ask the reviewer to understand this technical concern.

7. Fig. 4M. To validate authors' claim on the role of integrin Beta1-containing EVs on invasive behaviour, they should repeat the experiment using blocking beta1 antibodies prior to EV addition.

Response: we thank the reviewer for this comment. As requested, we performed the experiment using the Integrin $\beta 1$ blocking monoclonal antibody (mAb; clone P4C10). The new data show that P4C10A treatment alone or in combination with EVs derived from MCF10A cells transfected with siCtrl or siErbB3 significantly reduced invasiveness in comparison to IgG treatments, confirming the mechanistic role of Integrin $\beta 1$ promoting MCF10A invasive behaviour (new Figure 4M).

Figure 4M

8. While authors claim that their results could potentially clarify different aspects of tumour dissemination, most of their experiments are done in MCF10A, a non-tumorigenic epithelial cell line. To better support their conclusion, they should reproduce key experiments in MCF7 or other tumorigenic cell line.

Response: we thank the reviewer for the comment. As also mentioned in response to Editor's comment #1 and Reviewer 1 #1, the new data address the effect of ErbB3-depletion on recycling of integrin $\beta 1$ (new Supplementary Figure S3F). While we found that depletion of ErbB3 impaired recycling of Integrin $\beta 1$ in the non-malignant cells, including MCF10A and primary breast epithelial HMEC cells, Integrin $\beta 1$ internalisation was less effective in malignant MCF7 cells that overexpress ErbB3 (new supplementary Figures S1C and S3D, E), with limited detectable return of internalised Integrin $\beta 1$ to the cell surface of both control or ErbB3-depleted MCF7 cells complicating assessment of integrin recycling. In addition, we now provide a direct comparison of ErbB3 mRNA levels between the three cell lines used in our study (new Supplementary Figure S1C), and discuss the potential discrepancy between the normal/non-malignant cells and cancer cells, that raises the interesting question whether the mechanism and dynamics by which Integrin $\beta 1$ traffic, is affected during cancer progression,

meriting future investigation. Furthermore, we also included in this new version that GGA3 and Rab GTPase-binding effector protein 1 (Rabaptin5 or RABPT5) shared colocalisation with endogenous ErbB3 in MCF7 cells as the new Supplementary Figures 9A, B. We attempted to conduct the Arf6 IP in MCF7 cells, but as opposed to MCF10A cells, the yield of Arf6 in pull down experiments was much lower than in MCF10A cells (see figure below – reviewers only), and interacting proteins were not detectable. Finally, we also reproduced our findings about the impact of ErbB3, Rab4A and Rabaptin5 on EV secretion in MCF7 cells (New Supplementary Figures S4H-J and S5I-K).

New Supplementary Figures S3D, E

New Supplementary Figures S4G-I

New Supplementary Figures S5I-K

Supplementary Figures S5D, E

MCF7 protein extract: Arf6 IP (Reviewers only)

Minor comments:

1. Fig. 1D-1F: please explain better if beta1 integrin surface signal was quenched in these specific set of studies.

Response: Integrin β1 was quenched on ice with an antibody against Alexa488 as described by Arjonen et al. (Traffic, 2012; DOI: 10.1111/j.1600-0854.2012.01327.x), and outlined in the methods section and results section (page 6 and schematic Figure 4A).

2. Suppl. Fig. 3A: last WB lane should read "siErB2" instead of "siErbB3".

Response: we thank the reviewer and we apologize for this oversight. We corrected the siErbB2 lane in Supplementary Figure 2A (previous Supplementary Figure 3A), as requested.

September 9, 2025

RE: JCB Manuscript #202501255R

Dorival Rodrigues-Junior
Uppsala University

Dear Dr. Rodrigues-Junior:

Thank you for submitting your revised manuscript entitled "Loss of ErbB3 redirects Integrin β 1 from early endosomal recycling to secretion in extracellular vesicles". We would be happy to publish your paper in JCB pending final revisions necessary to meet our formatting guidelines as well as responding to the remaining reviewer concern (see details below).

A. MANUSCRIPT ORGANIZATION AND FORMATTING:

Please note that JCB now requires authors to submit Source Data used to generate figures containing gels and Western blots with all revised manuscripts. This Source Data consists of fully uncropped and unprocessed images for each gel/blot displayed in the main and supplemental figures. For assays performed using capillary electrophoresis and/or immunoassay-based detection, authors should instead provide the electropherogram graph(s) for each experiment, plotting fluorescence/chemiluminescence intensity vs. molecular weight/size. Please be sure to provide one Source Data file for each figure gels, blots, and/or capillary electrophoresis assays along with your revised manuscript files. File names for Source Data figures should be alphanumeric without any spaces or special characters (i.e., SourceDataF#, where F# refers to the associated main figure number or SourceDataFS# for those associated with Supplementary figures). For traditional gels and blots, the lanes of the gels/blots should be labeled as they are in the associated figure, the place where cropping was applied should be marked (with a box), and molecular weight/size standards should be labeled wherever possible. For capillary electrophoresis assays, each trace in the graph should be color-coded and labeled to indicate which protein, gene, or sample is being measured (please try to avoid red/green combinations to accommodate our color-blind readers).

1) Text limits: Character count for Articles is < 40,000, not including spaces. Count includes abstract, introduction, results, discussion, and acknowledgments. Count does not include title page, figure legends, materials and methods, references, tables, or supplemental legends.

2) Figures limits: Articles may have up to 10 main text figures.

3) **** Figure formatting:** Scale bars must be present on all microscopy images, including inset magnifications. Molecular weight or nucleic acid size markers must be included on all gel electrophoresis. Aspect ratios of images may not be altered. In order to accommodate readers with red-green color blindness, we ask that you please change all red/green color schemes. ******

4) Statistical analysis: Error bars on graphic representations of numerical data must be clearly described in the figure legend. The number of independent data points (n) represented in a graph must be indicated in the legend. Statistical methods should be explained in full in the materials and methods. For figures presenting pooled data the statistical measure should be defined in the figure legends. Please also be sure to indicate the statistical tests used in each of your experiments (either in the figure legend itself or in a separate methods section) as well as the parameters of the test (for example, if you ran a t-test, please indicate if it was one- or two-sided, etc.). Also, if you used parametric tests, please indicate if the data distribution was tested for normality (and if so, how). If not, you must state something to the effect that "Data distribution was assumed to be normal but this was not formally tested."

5) Abstract and title: The abstract should be no longer than 160 words and should communicate the significance of the paper for a general audience. The title should be less than 100 characters including spaces. Make the title concise but accessible to a general readership.

6) **** Materials and methods:** Should be comprehensive and not simply reference a previous publication for details on how an experiment was performed. Please provide full descriptions in the text for readers who may not have access to referenced manuscripts. ******

7) All antibodies, cell lines, animals, and tools used in the manuscript should be described in full, including accession numbers for materials available in a public repository such as the Resource Identification Portal. Please be sure to provide the sequences

for all of your primers/oligos and RNAi constructs in the materials and methods. You must also indicate in the methods the source, species, and catalog numbers (where appropriate) for all of your antibodies. Please also indicate the acquisition and quantification methods for immunoblotting/western blots.

8) Microscope image acquisition: The following information must be provided about the acquisition and processing of images:

- a. Make and model of microscope
- b. Type, magnification, and numerical aperture of the objective lenses
- c. Temperature
- d. Imaging medium
- e. Fluorochromes
- f. Camera make and model
- g. Acquisition software
- h. Any software used for image processing subsequent to data acquisition. Please include details and types of operations involved (e.g., type of deconvolution, 3D reconstitutions, surface or volume rendering, gamma adjustments, etc.).

9) * References: There is no limit to the number of references cited in a manuscript. References should be cited parenthetically in the text by author and year of publication. Abbreviate the names of journals according to PubMed.

10) Supplemental materials: There are strict limits on the allowable amount of supplemental data. Articles may have up to 5 supplemental figures. Please also note that tables, like figures, should be provided as individual, editable files. A summary of all supplemental material should appear at the end of the Materials and methods section.

13) ORCID IDs: ORCID IDs are unique identifiers allowing researchers to create a record of their various scholarly contributions in a single place. Please note that ORCID IDs are now **required** for all authors. At resubmission of your final files, please be sure to provide your ORCID ID and those of all co-authors.

Journal of Cell Biology now requires a data availability statement for all research article submissions. These statements will be published in the article directly above the Acknowledgments. The statement should address all data underlying the research presented in the manuscript. Please visit the JCB instructions for authors for guidelines and examples of statements at (<https://rupress.org/jcb/pages/editorial-policies#data-availability-statement>).

B. FINAL FILES:

****It is JCB policy that if requested, original data images must be made available to the editors. Failure to provide original images upon request will result in unavoidable delays in publication. Please ensure that you have access to all original data images prior to final submission.****

****The license to publish form must be signed before your manuscript can be sent to production. A link to the electronic license to publish form will be sent to the corresponding author only. Please take a moment to check your funder requirements before choosing the appropriate license.****

Thank you for your attention to these final processing requirements. Please revise and format the manuscript and upload materials within 7 days. If you need an extension for whatever reason, please let us know and we can work with you to determine a suitable revision period.

Thank you for this interesting contribution, we look forward to publishing your paper in Journal of Cell Biology.

Sincerely,

Johanna Ivaska
Monitoring Editor

Andrea L. Marat
Deputy Editor

Journal of Cell Biology

Reviewer #2 (Comments to the Authors (Required)):

The authors satisfactorily addressed or replied to all the issues I raised. However, in my opinion, the authors should still better contextualize their observations and conclusions in light of the role of ERBB3 in human cancers. First, at least one meta-analysis of different solid cancers indicate that ERBB3 overexpression correlates with poorer survival (Ocana et al., JNCI 2013 - 10.1093/jnci/djs501). Conversely, in a cohort of 177 ERBB2/HER2+ breast cancer cases, low total ERBB3/HER3 levels by immunohistochemistry predicted shorter recurrence-free survival independently, approximately a 2.3-fold higher risk of recurrence compared to high ERBB3/HER3 levels (Luhtala et al, BMC Cancer, 2018 - 10.1186/s12885-018-4917-1). Luhtala et al.'s article seems to be particularly relevant since Mendes Rodrigues's study focuses on normal and malignant breast epithelial cell lines.

Reviewer #3 (Comments to the Authors (Required)):

The authors have satisfactorily addressed all my comments

Dear Dr. Ivaska and Dr. Marat,

We are pleased with your consideration to publish our paper: "Loss of ErbB3 redirects integrin β 1 from early endosomal recycling to secretion in extracellular vesicles" in the *Journal of Cell Biology*. We hereby submit a revised version of the manuscript, incorporating the final revisions requested by Reviewer #02 and formatting the document according to JCB guidelines.

Summary of modifications made to meet JCB requirements:

1. *Figure formatting: Scale bars have been added to all microscopy images, and red-green color schemes have been changed to magenta-green to accommodate readers with color blindness.*
2. *Materials and methods: All antibodies, cell lines, animals, and tools used in the study are now properly described.*
3. *References: All citations are now presented parenthetically in the text by author and year of publication.*
4. *eTOC summary: An eTOC summary has been included as requested.*
5. *All remaining formatting and submission requirements were previously addressed.*

Reviewer #2 (Comments to the Authors (Required)):

The authors satisfactorily addressed or replied to all the issues I raised. However, in my opinion, the authors should still better contextualize their observations and conclusions in light of the role of ERBB3 in human cancers. First, at least one meta-analysis of different solid cancers indicate that ERBB3 overexpression correlates with poorer survival (Ocana et al., JNCI 2013 - 10.1093/jnci/djs501). Conversely, in a cohort of 177 ERBB2/HER2+ breast cancer cases, low total ERBB3/HER3 levels by immunohistochemistry predicted shorter recurrence-free survival independently, approximately a 2.3-fold higher risk of recurrence compared to high ERBB3/HER3 levels (Luhtala et al, BMC Cancer, 2018 - 10.1186/s12885-018-4917-1). Luhtala et al.'s article seems to be particularly relevant since Mendes Rodrigues's study focuses on normal and malignant breast epithelial cell lines.

Response: We thank the reviewer for this insightful comment. To better contextualize our observations and conclusions regarding the role of ErbB3 in human cancers, we have now included the references of Ocana et al. (2013) – last paragraph of page 8, and Luhtala et al. (2018) – page 22.

We hope you will consider this submission favourably and look forward to hearing from you.

With best regards,

Ingvar Ferby, PhD

Dorival Mendes Rodrigues-Junior, PhD